# Why Has Predicting Downstream Capabilities of Frontier AI Models with Scale Remained Elusive?

**Rylan Schaeffer** [1]  **Hailey Schoelkopf** [2]  **Brando Miranda** [1]  **Gabriel Mukobi** [1]  **Varun Madan** [1]  **Adam Ibrahim** [3]
**Herbie Bradley** [4]  **Stella Biderman** [2]  **Sanmi Koyejo** [1]

## Abstract

Predicting changes from scaling advanced AI systems is a desirable property for engineers, economists, governments and industry alike, and, while a well-established literature exists on how pretraining performance scales, predictable scaling behavior on downstream capabilities remains elusive. While many factors are certainly responsible, this paper identifies a significant factor that makes predicting scaling behavior on widely used multiple-choice question answering benchmarks challenging and illuminates a path towards making such downstream evaluations predictable with scale. Using five model families and twelve well-established multiple-choice benchmarks, we demonstrate that downstream performance is computed from negative log likelihoods via a sequence of transformations that progressively degrades the statistical relationship between performance and scale. We then pinpoint the mechanism causing this degradation: downstream metrics require comparing the correct choice against a small number of specific incorrect choices, meaning accurately predicting downstream capabilities requires predicting not just how probability mass concentrates on the correct choice with scale, but also how probability mass fluctuates on the alternative incorrect choices with scale. We empirically study how probability mass on the correct choice co-varies with probability mass on incorrect choices with increasing compute, suggesting that scaling laws for *incorrect* choices might be achievable. Our work also explains why pretraining scaling laws are commonly regarded as more predictable than downstream capabilities and con-

tributes towards establishing scaling-predictable evaluations of frontier AI models.

## 1. Introduction

Predictable scaling behavior of frontier AI systems such as Gemini (Team et al., 2023; Reid et al., 2024) and GPT (OpenAI, 2024; OpenAI et al., 2024) is crucial for anticipating capabilities and informing key decisions regarding model development and deployment (Anthropic, 2023; OpenAI, 2023; Dragan et al., 2024). Predictable scaling behaviors enable engineers to make informed decisions about optimal model design choices and to de-risk investment in exceedingly expensive pretraining runs by determining the payoff from scaling up compute; for example, OpenAI publicly noted that "A large focus of the GPT-4 project was building a deep learning stack that scales predictably" and that "[OpenAI] developed infrastructure and optimization methods that have very predictable behavior across multiple scales" (Achiam et al., 2023); OpenAI noted that this ideally goes beyond predicting loss values, and that "Having a sense of the capabilities of a model before training can improve decisions around alignment, safety, and deployment". Additionally, predicting capabilities is of interest beyond AI practitioners: economists and governments also have a significant interest in predicting the capabilities of current and future frontier AI systems for better decision-making (regulation, taxation, and safety) and forecasting of economic impacts (Council of Economic Advisers, 2024). Downstream capabilities are especially of interest because quantities like pretraining loss are difficult to translate into quantities more meaningful to society, such as the impact on economic labor or societal harms.

However, while scaling laws for pretraining loss are well-established (Hestness et al., 2017; Rosenfeld et al., 2019; Henighan et al., 2020; Kaplan et al., 2020; Gordon et al., 2021; Hernandez et al., 2021; Jones, 2021; Zhai et al., 2022; Hoffmann et al., 2022; Clark et al., 2022; Neumann & Gros, 2022; Hernandez et al., 2022; Maloney et al., 2022; Sardana & Frankle, 2023; Muennighoff et al., 2024; Besiroglu et al., 2024), the literature is less conclusive regarding predicting specific downstream capabilities with scale. Prior work has

[1]Stanford Computer Science  [2]EleutherAI  [3]MILA  [4]University of Cambridge.  Correspondence to: Rylan Schaeffer <rschaef@cs.stanford.edu>, Sanmi Koyejo <sanmi@cs.stanford.edu>.

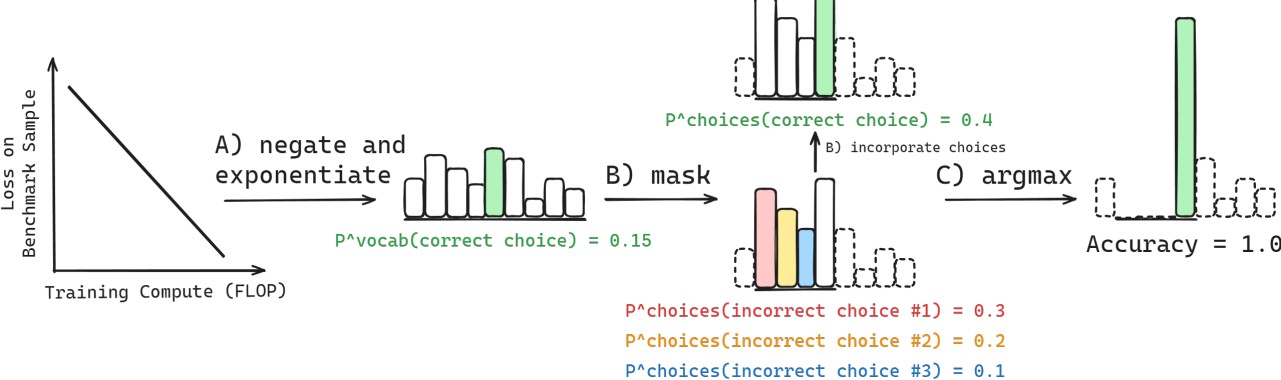

*Figure 1.* **Multiple-choice benchmark accuracy is computed from negative log likelihoods via a sequence of transformations that degrades predictability.** Computing `Accuracy` begins with computing the negative log likelihoods of each choice, then negating and exponentiating each to obtain the probability of each choice (**A**). Choices are then restricted to a set of available choices by *masking* invalid continuations, and renormalizing to obtain relative probability mass on each choice (**B**). Lastly, the model's choice is defined as $\arg\max_i\{p^{\text{Choices}}(\text{Available Choice}_i)\}$, and `Accuracy` is 1 if and only if the model's choice is the correct choice (**C**).

observed that performance on standard natural language processing (NLP) benchmarks can exhibit *emergent abilities* (Brown et al., 2020; Ganguli et al., 2022; Srivastava et al., 2022; Wei et al., 2022) where performance changes unpredictably with scale, but further work demonstrated that such unpredictable changes can be artifacts of researchers' analyses, i.e., choices of metrics and lack of sufficient resolution from too few samples (Srivastava et al., 2022; Schaeffer et al., 2023; Hu et al., 2024). More recently, Du et al. (2024) claim that downstream capabilities *can* be predicted, but *only* after the pretraining cross-entropy loss falls below a certain threshold, and Gadre et al. (2024) claim that while performance on individual tasks can be difficult to predict, aggregating results across dozens of diverse benchmarks yields clearer scaling trends. In this work, we ask: in contrast with strongly-predictable pretraining losses, *why has predicting specific downstream capabilities with scale remained elusive?*

## 2. Contribution: Explaining Why Predicting Downstream Capabilities With Scale Has Remained Elusive

Our goal is to understand what breaks down between the predictability of pretraining losses and the unpredictability of downstream evaluations. To do this, we investigated the comparative predictability of different evaluation methodologies and setups, focusing on popular and comparatively simple (yet still highly difficult to predict) multiple-choice question answering benchmarks. We began with scaling-predictable pretraining log likelihoods and tracked how these log likelihoods are transformed in the process of calculating downstream evaluation metrics that are notoriously

difficult to predict, such as `Accuracy` or `Brier Score` (See Fig. 1 and Sec. 4 for further details):

$$\underbrace{\log p_\theta^{\text{Vocab}}(\text{Correct Choice})}_{\text{Scaling-Predictable}}$$
$$\rightarrow p_\theta^{\text{Vocab}}(\text{Correct Choice})$$
$$\rightarrow p_\theta^{\text{Choices}}(\text{Correct Choice})$$
$$\rightarrow \underbrace{\texttt{Accuracy}}_{\text{Scaling-Unpredictable}}$$

This paper demonstrates the following findings:

1. **Calculating downstream metrics requires a sequence of transformations applied to the original scaling-predictable quantities. These transformations progressively deteriorate the statistical relationship between those metrics and the scaling parameters (parameters, data, and compute).** This formalizes an intuition that "more complex" metrics might be less easily predictable.

2. **Accurately predicting downstream multiple-choice performance requires modeling not only the probability mass assigned to the correct choice with scale, but also the probability mass assigned to the incorrect alternatives.** This explains the cause of comparative unpredictability of multiple-choice benchmarks; it also suggests a potential path forward for successful predictive models of downstream performance in the area of multiple-choice question answering, and in general the need to model *external information* not related to scaling-predictable log likelihoods needed for downstream metric computation.

3. **Continuous metrics such as `Brier Score` are insufficient for recovering predictability.** We observe that, contrary to prior work showing that metrics such as `Brier Score` can hide *emergent behavior* at times, `Brier Score` is insufficient to improve the statistical relationship degraded by incorporating incorrect choices' probability mass.

## 3. Methodology: Data for Studying Scaling of Downstream Capabilities

To study how downstream capabilities on specific tasks change with scale for different model families, we generated per-sample scores from a large number of model families and multiple-choice NLP benchmarks. To ensure the computed scores were consistent with prior work, we used EleutherAI's Language Model Evaluation Harness (Gao et al., 2023).

**Model Families**  Because our goal is to explore the scaling behavior of evaluations with increasing compute, we chose to evaluate model families with dense combinations of parameter counts and token counts. The following families were evaluated (additional details in App. E):

1. **Pythia** (Biderman et al., 2023b): The Pythia family contains 8 models from 70M to 12B parameters trained on the Pile (Gao et al., 2020) for 300B tokens. We used 8 checkpoints per size of the non-deduplicated variants.

2. **Cerebras-GPT** (Dey et al., 2023): The Cerebras-GPT family contains 7 models ranging from 111M to 13B parameters. The models were trained on the Pile (Gao et al., 2020) for different durations as part of a scaling study with a ratio of $\sim 20\times$ tokens to parameters in a "Chinchilla"-optimal manner (Hoffmann et al., 2022).

3. **OLMo** (Groeneveld et al., 2024): The OLMo family contains a 1B parameter model trained for 3T tokens and two 7B parameter models trained for 2T-2.5T tokens. We selected 7 checkpoints for 1B (spanning 84B[1] to 3T tokens) and 7 checkpoints for 7B (spanning 4B to 2.4T tokens).

4. **INCITE** (AI, 2023): The INCITE family contains 3B and 7B parameter models, trained on 0.8T and 1T tokens of RedPajama-v1(Computer, 2023). The 3B model has only a single checkpoint, so we excluded it. We found this family to be a slight outlier from other families, which we speculate is because its pretraining data were contaminated by benchmarks (Elazar et al., 2023).

5. **LLM360** (Liu et al., 2023): LLM360 includes two 7B parameter LLMs trained on 1.3T and 1.4T tokens. We selected 13 checkpoints of Amber spaced approximately logarithmically.

**NLP Benchmarks**  We evaluated the above model families on widely-used multiple-choice benchmarks for assessing comprehension, reasoning, and world knowledge: AI2 Reasoning Challenge (ARC) Easy and Hard (Clark et al., 2018), HellaSwag (Zellers et al., 2019), MathQA (Amini et al., 2019), MCTACO (Zhou et al., 2019), MMLU (Hendrycks et al., 2020), OpenbookQA (Mihaylov et al., 2018), PIQA (Bisk et al., 2020), RACE (Lai et al., 2017), SciQ (Welbl et al., 2017), SIQA (Sap et al., 2019a), WinoGrande (Keisuke et al., 2019) and XWinoGrad En (Muennighoff et al., 2023). For MMLU, we analyzed each of the 57 subjects (e.g., Abstract Algebra) independently. For each benchmark, we used default evaluation settings from the LM Evaluation Harness (Gao et al., 2023).

**Performance Metrics**  We used three common multiple-choice metrics (Srivastava et al., 2022; Schaeffer et al., 2023; Du et al., 2024): `Accuracy`, `Brier Score` (Brier, 1950), and probability mass on the correct choice relative to the available choices $p_\theta^{\text{Choices}}(\text{Correct Choice})$.

**Compute Budget Calculations**  Following prior work (Kaplan et al., 2020), we approximated[2] the pretraining compute $C$ (in terms of training FLOP) of a given model checkpoint as a function of the parameter count (excluding embeddings) $N$ and the amount of training data seen in tokens $D$: $C = C(N, D) \approx 6\,N\,D$.

## 4. What Makes Predicting Downstream Performance Difficult?

Performance on multiple choice benchmarks is commonly published as `Accuracy`, `Brier Score`, or probability mass on the correct choice out of the available choices $p_\theta^{\text{Choices}}(\text{Correct Choice})$. These quantities are computed via a sequence of transformations that begins with the negative log likelihood of the correct choice on this particular benchmark sample as some function $f(\cdot, \cdot)$ of compute:

$$\mathcal{L}_\theta^{\text{Vocab}}(\text{Correct Choice}) = f(\text{Compute}, \text{Benchmark Datum})$$

Two details are critical. Firstly, this negative log likelihood is not computed in expectation over a corpus; it is specific to this particular singular datum in the benchmark. *All the*

---

[1]OLMo 1B checkpoints below 84B tokens were unfortunately accidentally lost by their creators.

[2]This approximation neglects FLOP costs associated with attention calculations over sequence length; however, such operations are negligible so long as $d_{model} >> n_{ctx}/12$, and this approximation is therefore standard in most language model scaling law analyses.

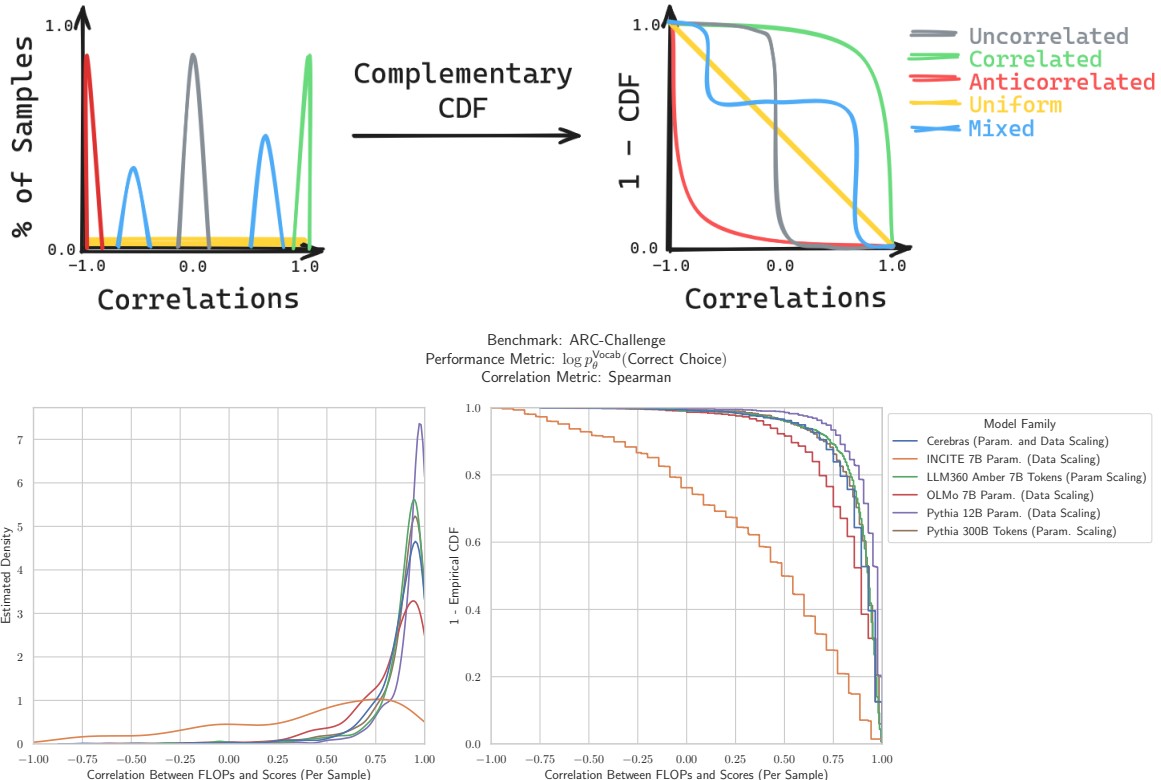

*Figure 2.* **Distributions of score-compute correlations and their corresponding complementary cumulative distribution functions. Left:** For each benchmark, model family, performance metric, and correlation metric, we computed how scores correlate with compute. This yields a distribution (over samples) of score-compute correlations. Note: the uniform distribution is small but non-zero everywhere. **Right:** To easily extract what fraction of samples in a benchmark has score-compute correlations above any given threshold, we converted the probability distributions to *complementary cumulative distribution functions*, i.e., 1 minus the empirical cumulative distribution function (CDF). **Top:** Idealized distributions. **Bottom:** Actual data on ARC Challenge.

*scores we discuss are per-datum.* Secondly, this negative log likelihood is computed over the vocabulary of the model. One can then compute the probability mass of the correct choice, again with respect to the vocabulary:

$$p_\theta^{\text{Vocab}}(\text{Correct Choice}) = \exp\left(-\mathcal{L}_\theta^{\text{Vocab}}(\text{Correct Choice})\right)$$

Next, probabilities are restricted to the set of available choices by masking invalid continuations and normalizing again with respect to this set:

$$p_\theta^{\text{Choices}}(\text{Correct Choice}) \stackrel{\text{def}}{=} \frac{p_\theta^{\text{Vocab}}(\text{Correct Choice})}{\sum_i p_\theta^{\text{Vocab}}(\text{Available Choice}_i)}$$

We distinguish the support over the token space of the model versus over the set of available choices in the benchmark's question because, as we will show, the support crucially affects predictability. Finally, the choices-normalized probability masses become standard downstream metrics:

$$\text{Accuracy}_\theta \stackrel{\text{def}}{=}$$

$$\mathbb{1}\left(\text{Correct Choice} = \arg\max_i \left\{p_\theta^{\text{Choices}}(\text{Avail. Choice}_i)\right\}\right)$$

and Brier score:

$$\text{Brier Score}_\theta \stackrel{\text{def}}{=} \sum_i \Big(\mathbb{1}(\text{Avail. Choice}_i =$$

$$\text{Correct Choice}) - p_\theta^{\text{Choices}}(\text{Avail. Choice}_i)\Big)^2$$

where $\mathbb{1}(\cdot)$ is an indicator variable. To quantify how this sequence of transformations affects predictability of performance, we measured how per-sample scores correlate with pretraining compute, and then studied how the distribution (over samples) of correlation values shifts from log likelihoods to $p_\theta^{\text{Vocab}}(\text{Correct Choice})$ to $p_\theta^{\text{Choices}}(\text{Correct Choice})$ to Accuracy or Brier Score. Specifically, for each combination of *model family, benchmark, performance metric, correlation metric*, we computed a correlation value for each sample in the benchmark between pretraining compute and scores. This yielded a distribution (over samples) of correlation values for the combination (Fig. 2 Left). Visualizing the distribution of correlations for the combination told us what fraction of samples in the benchmark yielded scores that are correlated, uncorrelated or anticorrelated

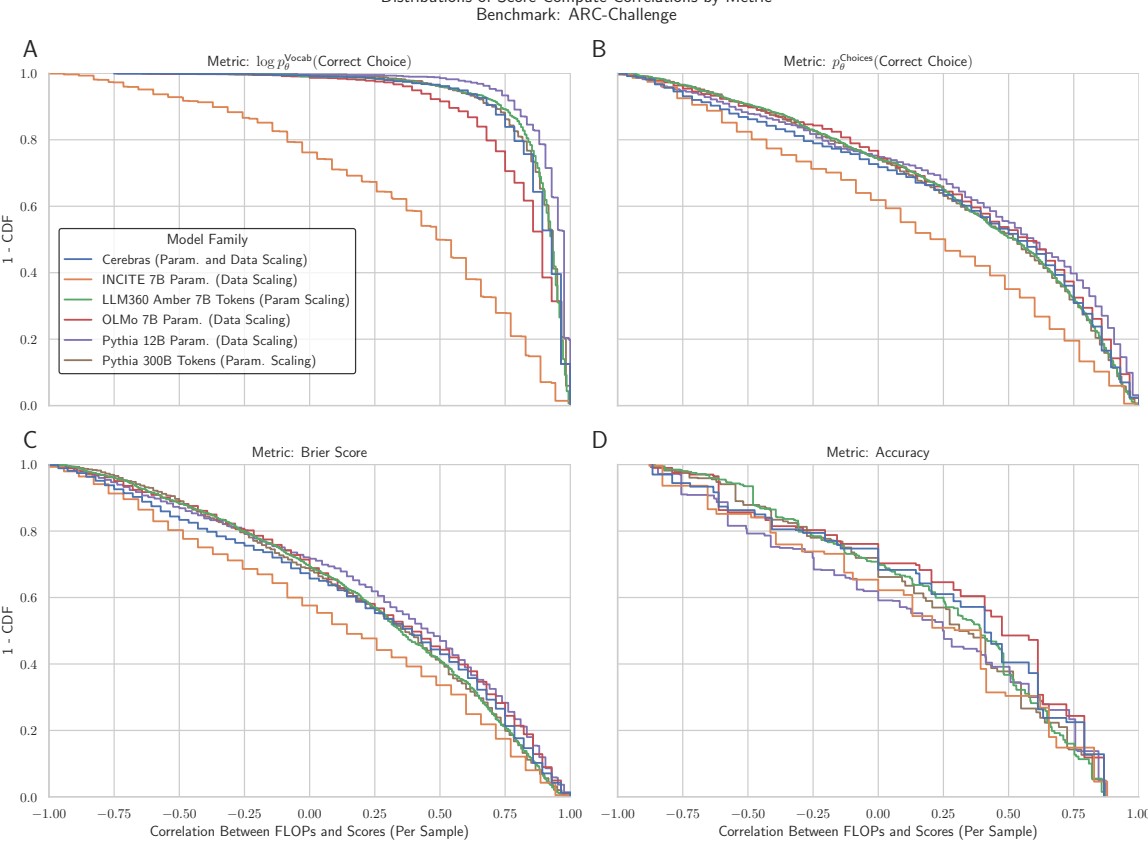

*Figure 3.* **Multiple-choice metrics like `Accuracy` and `Brier Score` are computed via a sequence of transformations that degrades correlations between performance scores and pretraining compute. (A)** Initially, scores under $\log p_\theta^{\text{Vocab}}(\text{Correct Choice})$ and compute are highly correlated. Transforming $\log p_\theta^{\text{Vocab}}(\text{Correct Choice}) \rightarrow p_\theta^{\text{Vocab}}(\text{Correct Choice})$ has no effect for rank correlations. **(B)** Transforming $p_\theta^{\text{Vocab}}(\text{Correct Choice}) \rightarrow p_\theta^{\text{Choices}}(\text{Correct Choice})$ decorrelates scores from compute. **(C)** Transforming $p_\theta^{\text{Choices}}(\text{Correct Choice}) \rightarrow$ `Brier Score` minorly decreases score-compute correlations. **(D)** Transforming $p_\theta^{\text{Choices}}(\text{Correct Choice}) \rightarrow$ `Accuracy` more substantially decorrelates scores from compute. Correlation: Spearman. Results are consistent across benchmarks and all three correlation metrics (App. H).

with compute (Fig. 2 Right). We found consistent results using all three standard correlation metrics: Pearson (1895), Kendall (1938) and Spearman (1961).

We demonstrate how the sequence of transformations affects the distribution of score-compute correlations using ARC Challenge (Clark et al., 2018) as an illustrative benchmark; we note that all other benchmarks exhibited similar patterns as well (App. H). We visualized the distributions via their complementary (empirical) cumulative distribution functions (complementary CDFs) (App. C):

$$\hat{S}(c) \stackrel{\text{def}}{=} \frac{1}{S} \sum_{s=1}^{S} \mathbb{1}\{C_s > c\}, \tag{1}$$

where $S$ is the number of samples in the benchmark and $C_s$ is the correlation (over the models in the model family) between compute and scores on the $s$-th sample in the benchmark. For a given threshold $c$, the complementary CDF $\hat{S}(c)$

returns the fraction of the benchmark's samples with score-compute correlations greater than the threshold $c$ (Fig. 3A). Beginning with log likelihoods, approximately 90% of samples exhibit score-compute correlations $> 0.75$, regardless of the model family (Fig. 3A). Transforming negative log likelihoods into probability masses $p_\theta^{\text{Vocab}}(\text{Correct Choice})$ does not affect the distribution of score-compute correlations for Spearman and Kendall. However, transforming $p_\theta^{\text{Vocab}}(\text{Correct Choice})$ into $p_\theta^{\text{Choices}}(\text{Correct Choice})$ decreases the distribution of score-compute correlations (Fig. 3B), with only 40% of samples having score-compute correlations $> 0.75$. Transforming $p_\theta^{\text{Choices}}(\text{Correct Choice})$ into `Brier Score` has little-to-no effect (Fig. 3C), but transforming into `Accuracy` (Fig. 3D) furthers decreases score-compute correlations. To quantitatively test whether these transformations indeed decrease the correlation between scores and compute, we measured four statistics of these score-compute correlation distributions: (1) the mean,

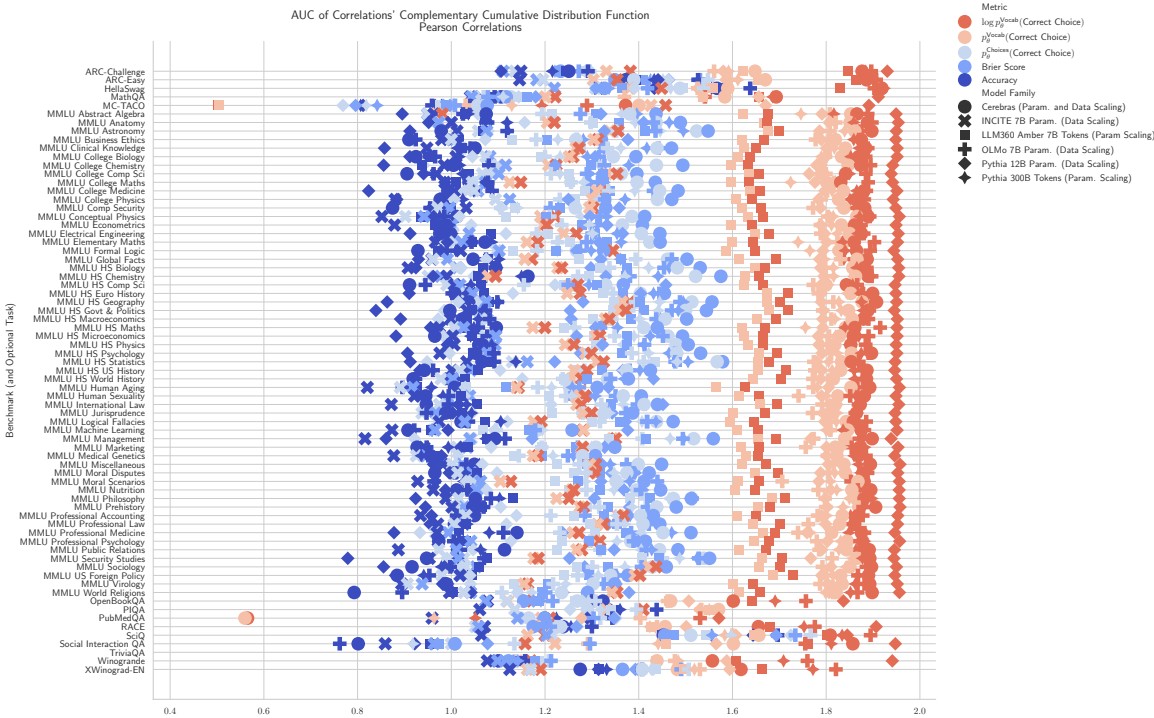

*Figure 4.* **All four statistics of score-compute correlation distributions demonstrate that transforming** $\log p_\theta^{\text{Vocab}}(\textbf{Correct Choice})$ **into** `Accuracy` **causes score-compute correlations to deteriorate.** We find a consistent trend that the sequence of transformations degrades score-compute correlations, as shown by the right-to-left $\log p_\theta^{\text{Vocab}}(\textbf{Correct Choice})$-to-$p_\theta^{\text{Vocab}}(\textbf{Correct Choice})$-to-$p_\theta^{\text{Choices}}(\textbf{Correct Choice})$-or-`Brier Score`-to-`Accuracy` vertical stripes. This trend holds across benchmarks and model families for three correlation metrics (Spearman, Pearson and Kendall) and for four statistics of correlation distributions (mean, median, the area under the survival function, and negative Wasserstein distance from perfect correlation or perfect anti-correlation). See App. Figs. 7, 8, 9 for other correlation metrics and other score-compute correlation distribution statistics.

(2) the median, (3) the area under the complementary CDF and (4) the negative[3] of the minimum of two Wasserstein distances: between the empirical correlation distribution and an ideal distribution of all correlations = 1, and between the empirical distribution and an ideal distribution of all correlations = −1. Across the four summary statistics, for most benchmarks and for most model families, we discovered a consistent ordering of metrics of the score-compute correlation distributions (Fig. 4):

$$\text{Corr}\big(\text{Compute}, \log p_\theta^{\text{Vocab}}(\textbf{Correct Choice})\big)$$
$$\geq \text{Corr}\big(\text{Compute}, p_\theta^{\text{Vocab}}(\textbf{Correct Choice})\big)$$
$$> \text{Corr}\big(\text{Compute}, p_\theta^{\text{Choices}}(\textbf{Correct Choice})\big)$$
$$\geq \text{Corr}\big(\text{Compute}, \texttt{Brier Score}\big)$$
$$> \text{Corr}\big(\text{Compute}, \texttt{Accuracy}\big)$$

To quantitatively confirm that the correlation scores indeed

follow this ordering, we computed what fraction of (benchmark, correlation metric, model family, correlation distribution statistic) tuples obey the ordering. To be maximally conservative, we checked for strict inequalities only. We found that across benchmarks, model families, and the 4 correlation distribution statistics, the claimed ordering of metrics held at least 82.4% of the time for Pearson, 85.6% for Spearman and 90.4% for Kendall.

## 5. Probability Mass on Incorrect Choices Causes Unpredictability

What is the mechanism that degrades how scores correlate with compute? All three metrics with degraded correlations - $p_\theta^{\text{Choices}}(\text{Correct Choice})$, `Accuracy`, and `Brier Score` - depend not just on how the model's probability mass $p_\theta^{\text{Vocab}}(\text{Correct Choice})$ concentrates on the correct choice as compute increases, but also depend on how the model's probability mass fluctuates on *incorrect* available choices $\{p_\theta^{\text{Vocab}}(\text{Incorrect Choice})\}_{\text{Incorrect Choices}}$ as compute increases. As an example, suppose

---

[3]We chose the *negative* Wasserstein distance for consistency with the other statistics: higher values correspond to higher correlations between scores and compute.

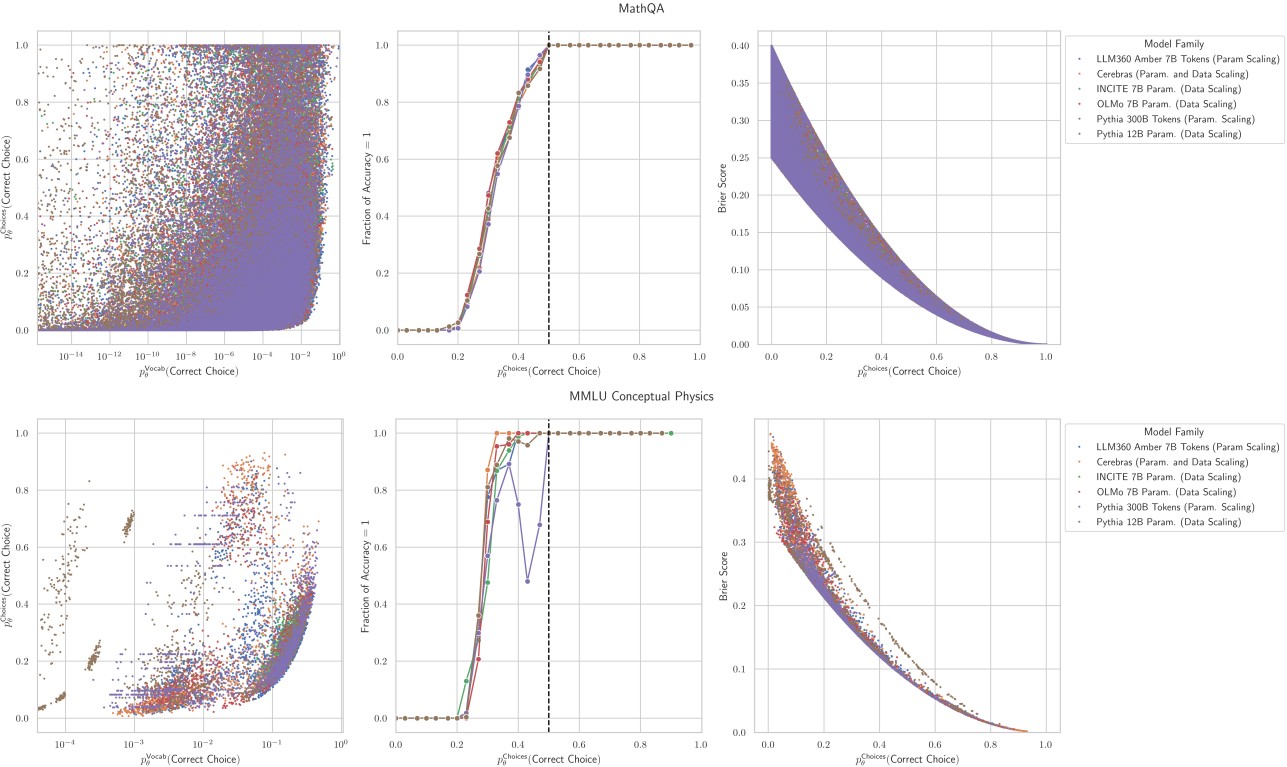

*Figure 5.* **Predictability deteriorates because of probability mass fluctuating on alternative incorrect choices with scale. Left:** Transitioning from $p_\theta^{\text{Vocab}}(\text{Correct Choice})$ to $p_\theta^{\text{Choices}}(\text{Correct Choice})$ demonstrates that $p_\theta^{\text{Vocab}}(\text{Correct Choice})$ contains little information about $p_\theta^{\text{Choices}}(\text{Correct Choice})$ and vice versa; loosely speaking, any value of one can map to any value of the other. **Center:** While $p_\theta^{\text{Choices}}(\text{Correct Choice}) > 0.5$ must yield `Accuracy = 1`, for any $p_\theta^{\text{Choices}}(\text{Correct Choice}) < 0.5$, knowing $p_\theta^{\text{Choices}}(\text{Correct Choice})$ contains little information about `Accuracy` and vice versa. **Right**: `Brier Score` is more predictable from $p_\theta^{\text{Choices}}(\text{Correct Choice})$ than `Accuracy`, but still quite variable. Two benchmarks shown: MathQA & MMLU Conceptual Physics.

$p_\theta^{Vocab}(\text{Correct Choice}) = 0.4$ on a 4-way multiple-choice question; what is the accuracy? Spreading the remaining mass uniformly on the incorrect choices will make `Accuracy = 1`, whereas concentrating mass on a single incorrect choice will make `Accuracy = 0`.

To demonstrate how drastically the probability mass placed on incorrect choices can alter performance, we visualized the relationships between pairs of metrics immediately preceding and following a given transformation (Fig. 5). For negative log likelihood of the correct choice and $p_\theta^{Vocab}(\text{Correct Choice})$ (not pictured), we observed a clean correspondence between performance under the metric and compute: one can reliably map a given value of these metrics to compute, and vice versa. In contrast, once performance is evaluated using a metric that is a function of the incorrect choices - $p_\theta^{\text{Choices}}(\text{Correct Choice})$, `Accuracy` or `Brier Score` - nearly any value of a score under one metric can map to any value of $p_\theta^{\text{Vocab}}(\text{Correct Choice})$ or $p_\theta^{\text{Choices}}(\text{Correct Choice})$ respectively (Fig. 5), thereby breaking the chain along which one can cleanly infer com-

pute from an observed metric. We can see that `Brier Score`, a metric meant to produce more continuous scores (Schaeffer et al., 2023), is less variable than `Accuracy`, provided a known $p_\theta^{\text{Choices}}(\text{Correct Choice})$, but it cannot recover information about $p_\theta^{\text{Vocab}}(\text{Correct Choice})$ that is lost when shifting to $p_\theta^{\text{Choices}}(\text{Correct Choice})$. We next show that this is because of the additional information regarding the underdetermined values of $p_\theta^{\text{Choices}}(\text{Incorrect Choice})$ for each incorrect choice.

## 6. Scaling Behavior of Probability Mass on Incorrect Choices

In general, aggregate performance over a distribution is often of interest. Such a focus on aggregate performance leads to an important insight: in MCQA, probability mass fluctuations on incorrect choices do not "average out". Unlike estimating the mean of a random variable, where positive and negative deviations cancel, the nonlinear nature of metrics like Accuracy and Brier Score means that probability

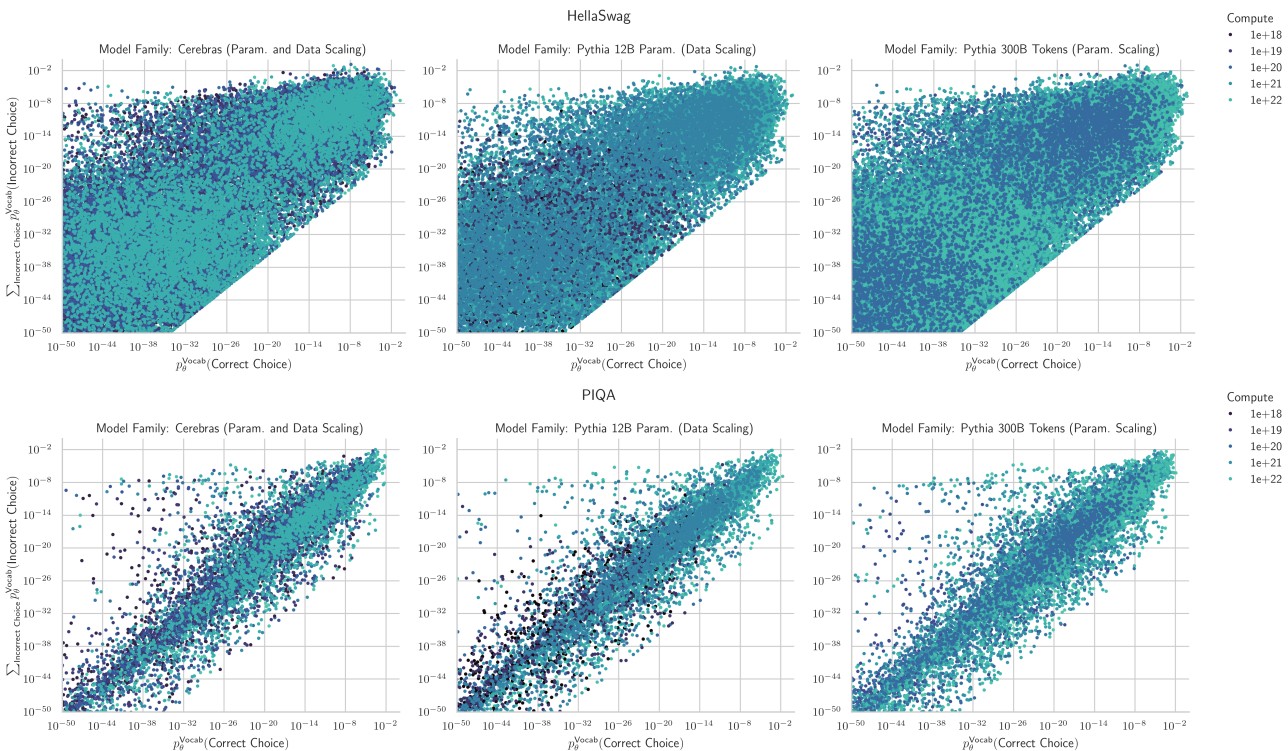

*Figure 6.* **Probability mass on the correct choices and the incorrect choices are correlated, but can fluctuate substantially.** Probability mass on correct choices and incorrect choices positively covaries and typically increases with compute. However, the spread is large: for any given value of $p_\theta^{\text{Vocab}}(\text{Correct Choice})$, the mass on incorrect choices can vary by many orders of magnitude.

mass shifts between incorrect options affect scores in complex ways that persist under averaging. For example, if probability mass shifts from incorrect option A to incorrect option B, the impact on accuracy depends on whether either option had enough mass to compete with the correct answer - there's no natural cancellation. This perhaps counter-intuitive behavior partially explains why predicting aggregate performance via averaging has remained elusive.

This analysis suggests that modeling probability mass fluctuations on incorrect choices could improve predictions of metrics like Accuracy and Brier Score, though the magnitude of improvement remains an open question. For metrics like `Accuracy`, such predictions should be made for each sample because knowing the average mass (across many data) placed on incorrect choices says little about how much mass is placed on any single incorrect choice for a single sample. We conclude by providing preliminary evidence that achieving such a feat might be possible. Specifically, we test how probability masses on correct choices and probability masses on incorrect choices covary with increasing compute (Fig. 6). Multiple benchmarks display strong positive relationships between mass on correct choices and mass on incorrect choices, suggesting that fitting *per-sample scal-*

*ing trends for each incorrect choice* might be possible; doing so might enable better predicting changepoints in metrics like `Accuracy` or `Brier Score`. However, whether per-benchmark per-sample per-choice scaling trends can be fit and accurately extrapolated is unclear since the spread varies by several orders of magnitude. We leave this challenge to future work.

## 7. Discussion

This work identifies why multiple-choice assessments of frontier AI models are unpredictable, as well as the underlying mechanism: probability mass on incorrect choices. Our results have implications for the design of future evaluations of frontier AI models that are reliably predictable with scaling. We hope that our work will be extended to further the science of scaling-predictable evaluation of AI systems, especially for complex and important model capabilities. We note several future directions for extension of our work, and we hope that the community also adopts our framing to further improve scaling-predictable evaluations. For **Related Work**, please see Appendix A. For **Future Directions**, please see Appendix B.

## Acknowledgements

RS acknowledges support from Stanford Data Science and the OpenAI Superalignment Fast Grant. SK acknowledges support by NSF 2046795 and 2205329, IES R305C240046, ARPA-H, the MacArthur Foundation, Schmidt Sciences, OpenAI, and Stanford HAI.

## Impact Statement

This work contributes to the crucial task of making AI system capabilities more predictable as they scale. Our research has several important implications for the development and deployment of frontier AI systems:

First, by identifying why downstream capability prediction has remained challenging, our work can help AI developers and organizations make more informed decisions about model development and resource allocation. This could lead to more efficient and targeted development of AI systems, potentially reducing the environmental and computational costs associated with training large models.

Second, our findings have implications for AI governance and safety. The ability to better predict model capabilities is essential for proactive policy-making and risk assessment. Our work highlights fundamental challenges in capability prediction that should inform discussions about AI development timelines and safety measures. This is particularly relevant as policymakers and organizations work to establish frameworks for responsible AI development.

Third, our research advances the scientific understanding of how AI capabilities emerge and scale. This improved understanding could help bridge the gap between theoretical scaling laws and practical applications, potentially enabling more reliable forecasting of AI progress.

However, we acknowledge that improving capability prediction could potentially be misused to optimize for harmful applications or accelerate unsafe development practices. To mitigate these risks, we have focused our research on understanding fundamental mechanisms rather than providing specific optimization techniques, and we emphasize the importance of using these insights within a framework of responsible AI development.

We encourage future work in this area to carefully consider both the benefits of improved predictability and the potential risks of capability optimization, ensuring that advances in understanding AI scaling behavior contribute to the safe and beneficial development of artificial intelligence.

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

## A. Related Work

**Language Model Evaluation**    The capabilities of AI models are typically evaluated using constructed datasets to assess performance on a specific task, acting as a proxy for some real-world usage scenario. However, performing robust and reliable evaluations is a challenge, with many potential pitfalls and unsolved problems (Biderman et al., 2024). For example, we might prefer to ask models open-ended questions and evaluate their answers in natural language, but it then often becomes difficult to robustly score the resulting model outputs, especially for partial correctness. For this reason, it is common practice for evaluation benchmarks to simplify their scoring via approximations, such as extracting a sub-string from free-form outputs heuristically (Joshi et al., 2017; Kwiatkowski et al., 2019; Hendrycks et al., 2021) and checking that it matches a specific gold target string, or casting a task to a *multiple-choice* format, in which a closed set of correct and incorrect answers is known, and the model's answer is determined by selecting the most likely option among these strings. For more details on the precise procedures typically used for multiple choice elsewhere in the literature, see Biderman et al. (2024). We believe that the multiple-choice format is valuable, due to its flexibility, popularity and relevance (Brown et al., 2020; Beeching et al., 2023; Biderman et al., 2024), but we discuss its limitations in Section 7.

**Scaling Laws**    Many neural networks exhibit power-law scaling of the pretraining loss as a function of the amount of compute, data, or parameters used for training (Hestness et al., 2017; Brown et al., 2020; Hoffmann et al., 2022). These neural scaling laws demonstrate that the pretraining loss can be highly predictable as a function of these fundamental inputs, which has a number of practical applications: Scaling laws fit to smaller training runs can be used to predict the pretraining loss of a much larger training run, and can be used to determine effective hyperparameters (McCandlish et al., 2018; DeepSeek-AI et al., 2024), or the optimal allocation of dataset and model size for a given compute budget (Hoffmann et al., 2022; Muennighoff et al., 2024; Dey et al., 2023; Sardana & Frankle, 2023; Besiroglu et al., 2024). In some cases, such laws can be used to predict performance of a larger model in a particular domain, such as coding (Achiam et al., 2023). The existence of scaling laws turns deep learning into a predictable science at the macro level by providing a simple recipe for improving model quality and de-risking returns on increasing investment into scale (Ganguli et al., 2022; Bowman, 2023).

**Emergent Abilities**    Language models have been observed to exhibit apparent *emergent abilities*—behaviors on downstream task performance that cannot be predicted from smaller scales (Wei et al., 2022; Srivastava et al., 2022). Emergence appears not to be simply a product of training compute or model size, but is also dependent on other factors such as dataset composition (Muckatira et al., 2024; Wei et al., 2022). Schaeffer et al. (2023) find that some emergent phenomena can be a "mirage" arising due to choices made by researchers such as the use of discontinuous metrics and insufficient resolution. However, Du et al. (2024) note that for many tasks, emergence remains despite the use of continuous metrics. Additionally, discontinuous metrics have been argued to often be the most reflective of real-world usefulness, so emergence in these hard metrics is important. Hu et al. (2024) found that for generative evaluations, infinite resolution can be achieved but requires significant compute and that generated answer be verifiable.

**Predicting Downstream Task Performance**    Although predicting macroscopic pretraining loss is useful, a far more useful goal is to predict the scaling of model performance on particular downstream tasks or domains. If this was possible, then model developers could tune their datasets and training procedures in a more fine-grained way before launching computationally intensive training runs. Model performance on a particular downstream task is typically correlated with compute, albeit with a few exceptions (McKenzie et al., 2022; Huang et al., 2024). However, despite attempts to fit scaling laws to values other than loss, including benchmark scores (Gadre et al., 2024; Zhang et al., 2024), model memorization (Biderman et al., 2023a), or reward (Gao et al., 2022), these downstream performance metrics are usually more noisy or require more compute to fit accurately. Owen (2024) and Gadre et al. (2024) both find that while *aggregate* benchmark performance with more compute can be predicted, the scaling behaviour of individual tasks can be noisy. Additionally, Owen (2024), Du et al. (2024) and Gadre et al. (2024) claim that predicting scaling behavior on a task without access to models exhibiting better-than-random performance (i.e., "before emergence occurs") cannot be done reliably. Concurrently to our work, Ruan et al. (2024) propose Observational Scaling Laws by mapping model capabilities from compute to a shared low-dimensional space of capabilities across model families before predicting performance on novel tasks. Our goal in this work is to investigate the comparative unpredictability of individual downstream performance scores, and advise how to create more scaling-predictable evaluations that are closely coupled with real-world use-cases.

## B. Future Directions

**Direction 1: Beyond Multiple Choice Benchmarks**    Our study is restricted to benchmarks evaluated via log likelihood-based multiple-choice formats. While we believe this is inherently valuable due to the usefulness and prevalence of such tasks, this limits the application of our findings. We hope that our discoveries and proposed mechanisms may be used to inform the study of predictable and reliable evaluation writ large, and that future work should explore the extent to which our findings can be generalized to more complex capabilities. Our findings corroborate those of Lyu et al. (2024), who find that multiple-choice answer scores often diverge from generative evaluations. Consequently, a particularly important direction for further study is to investigate generative evaluations, which may contain similar transformations distancing performance from the observed loss.

**Direction 2: Predicting Benchmark Performance A Priori**    Our work provides an explanation why multiple-choice benchmark performance is not easily predictable for metrics such as `Accuracy` and `Brier Score`, as observed in the literature (Du et al., 2024). However, our analyses assume access to entire model families' scores across several orders of magnitude of pretraining FLOPs, and do not employ backtesting, as sensibly recommended by Alabdulmohsin et al. (2022) and Owen (2024). A predictive model should be able to identify change points well in advance on standard metrics like `Accuracy` or `Brier Score`.

## C. Definition of Survival Function

The survival function $S_X(x)$ – also known as the reliability function, the tail distribution, or the complementary cumulative distribution function – gives the probability that a random variable $X$ exceeds a certain value $x$ (Kleinbaum & Klein, 2012; contributors, 2023):

$$S_X(x) \overset{\text{def}}{=} Pr[X > x] = \int_x^\infty f_X(x')\,dx' = 1 - F_X(x) \tag{2}$$

where $F_X(x) = Pr[X \leq x]$ is the cumulative distribution function (CDF) and $f_X(x)$ is the probability density function (pdf) or probability mass function (pmf) of the random variable $X$. The CDF $F_X(x)$ gives the probability that the random variable $X$ is at most $x$, while the survival function $S_X(x)$ gives the probability that $X$ exceeds $x$.

When the true distribution of $X$ is unknown, we can use the empirical CDF (ECDF) $\hat{F}_X(x)$ and the empirical survival function (ESF) $\hat{S}_X(x)$:

$$\hat{S}_X(x) \overset{\text{def}}{=} \frac{1}{S}\sum_{s=1}^s 1\{X_s > x\} = 1 - \hat{F}_X(x) \tag{3}$$

where $n$ is the number of observations, $x_i$ is the realized value of the random variable $X$ for observation $i$, and $1\{x_i > x\}$ is the indicator function. The empirical survival function $\hat{S}_X(x)$ specifies the fraction of observations for which the sampled random variable $X$ exceeds $x$.

## D. Compute Resources for Experiments

Experiments were done across a wide family of model families and sizes. The GPUs we used for medium-sized models (7B parameters and above) used a single A100s with 80GB of vRAM. For smaller models (≤8B) we used A100s with 80GB of vRAM, Quadro RTX 8000 with 48GB of vRAM, or RTX A4000 with 16GB of vRAM. For 70B parameter models, we used at least 2 A100 GPUs with 80GB of vRAM.

## E. Additional Model Family Details

Here we provide further experimental details regarding our selection of model families.

1. **Pythia** (Biderman et al., 2023b): We consider two "families" for Pythia in our experiments. **Pythia (Parameter Scaling)** refers to the use of fully-trained checkpoints from 9 different model sizes (all model sizes documented in Biderman et al. (2023), as well as a 14M parameter model trained later by the authors). **Pythia-12B (Data Scaling)** refers to the use of 8 checkpoints across training for the Pythia-12B model, namely having seen 2M, 64M, 2B, 6B, 20B, 60B, 200B, and 300B tokens in training.

2. **Cerebras-GPT** (Dey et al., 2023): **Cerebras (Parameter and Data Scaling)** refers to our use of 1 checkpoint per model in the Cerebras-GPT family, each fully trained for differing quantities of data as documented by the model creators, for 7 checkpoints in total.

3. **OLMo** (Groeneveld et al., 2024): **OLMo (7B Data Scaling)** refers to the use of 7 checkpoints for OLMo-7B across training, namely, checkpoints having seen 4B, 44B, 133B, 442B, 885B, 1.5T, and 2.4T tokens.

4. **INCITE** (AI, 2023): **INCITE-7B (Data Scaling)** considers 6 checkpoints over training for the 7B parameter model, having seen 240B, 280B, 400B, 500B, 700B, and 1T tokens.

5. **LLM360** (Liu et al., 2023): **LLM360 Amber (Data Scaling)** considers 13 checkpoints of the Amber model, having seen 0B, 3.5B, 7B, 10.5B, 17.5B, 31.5B, 49B, 87.5B, 147B, 252B, 430B, 738B, and 1.26T tokens.

## F. Broader Impact

This paper contributes to a better understanding of the predictability of large language models (LLMs), which can have both positive and negative societal impacts. On the positive side, by making LLM benchmarks more predictable, this research can help society anticipate and plan for potential challenges associated with their development and deployment. This increased predictability can facilitate proactive measures to mitigate risks and ensure the responsible use of AI technologies.

However, the increased predictability of LLMs could theoretically be exploited by malicious actors to accelerate the development of AI systems designed for malicious purposes. We also stress the importance of proactive risk assessment and the implementation of safeguards to prevent the misuse of AI technologies.

# G. Score-Compute Correlation Distributions' Statistics

## G.1. Pearson Correlations

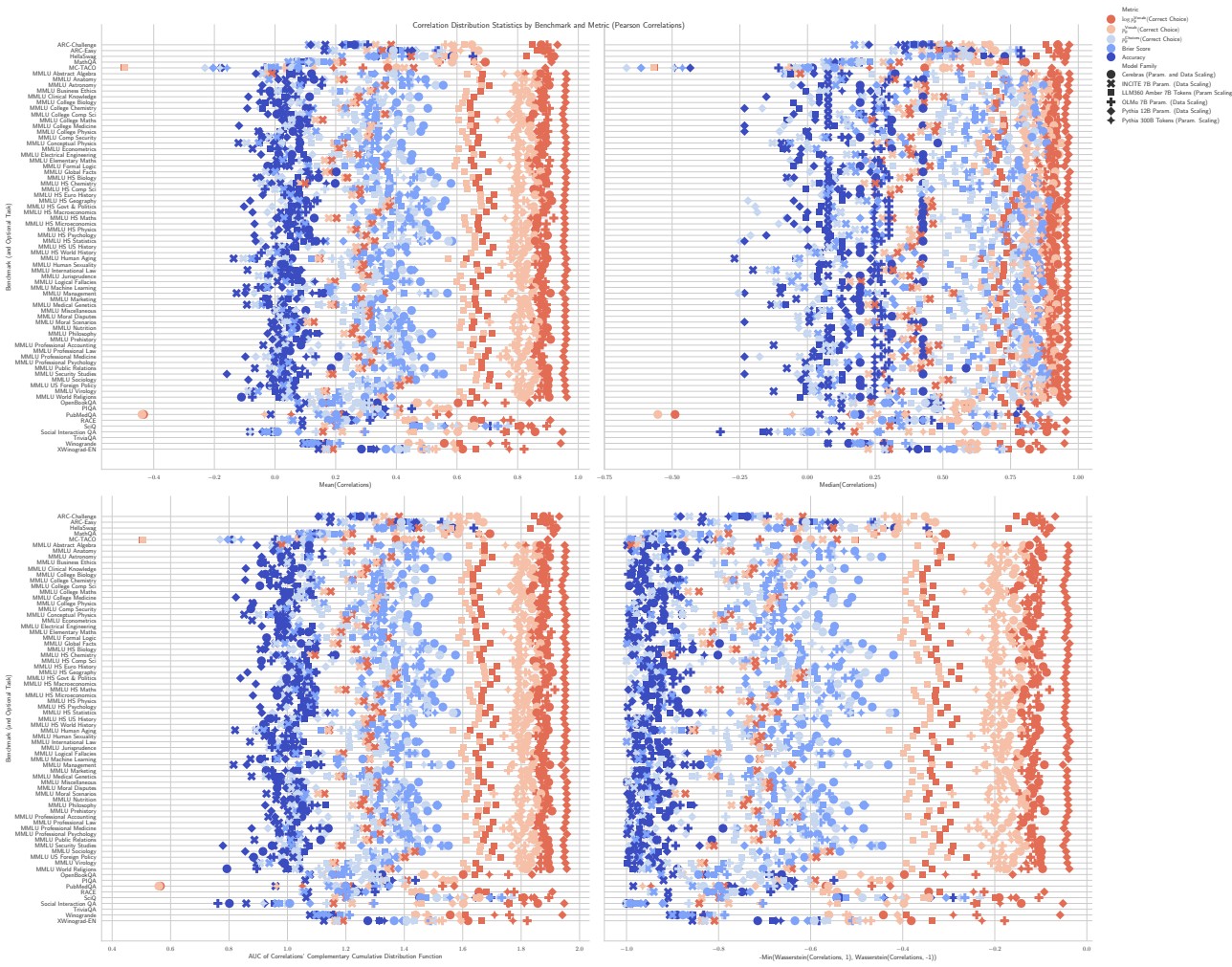

*Figure 7.* **Statistics for empirical distributions of correlations between scores and compute for all benchmarks and model families.** These correlation values were computed with Pearson correlation and are consistent with the main text's results computed with Spearman correlation (Fig. 4): The sequence of transformations from $\log p_\theta^{\text{Vocab}}(\text{Correct Choice}) \rightarrow p_\theta^{\text{Vocab}}(\text{Correct Choice}) \rightarrow p_\theta^{\text{Choices}}(\text{Correct Choice}) \rightarrow$ Accuracy degrades predictability.

## G.2. Spearman Correlations

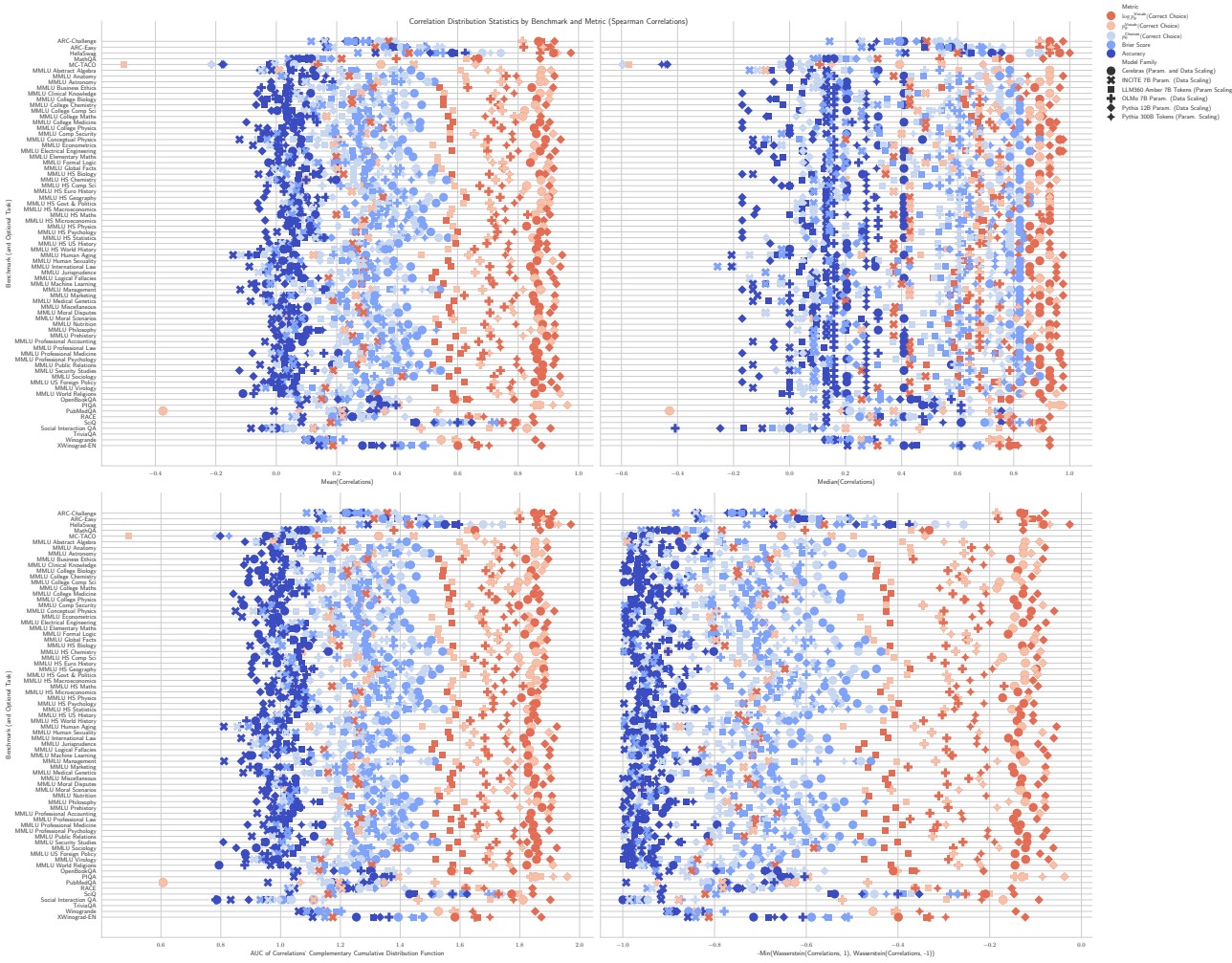

*Figure 8.* **Statistics for empirical distributions of correlations between scores and compute for all benchmarks and model families.** These correlation values were computed with Spearman correlation. The sequence of transformations from $\log p_\theta^{\text{Vocab}}(\text{Correct Choice}) \rightarrow p_\theta^{\text{Vocab}}(\text{Correct Choice}) \rightarrow p_\theta^{\text{Choices}}(\text{Correct Choice}) \rightarrow \text{Accuracy}$ degrades predictability.

## G.3. Kendall Correlations

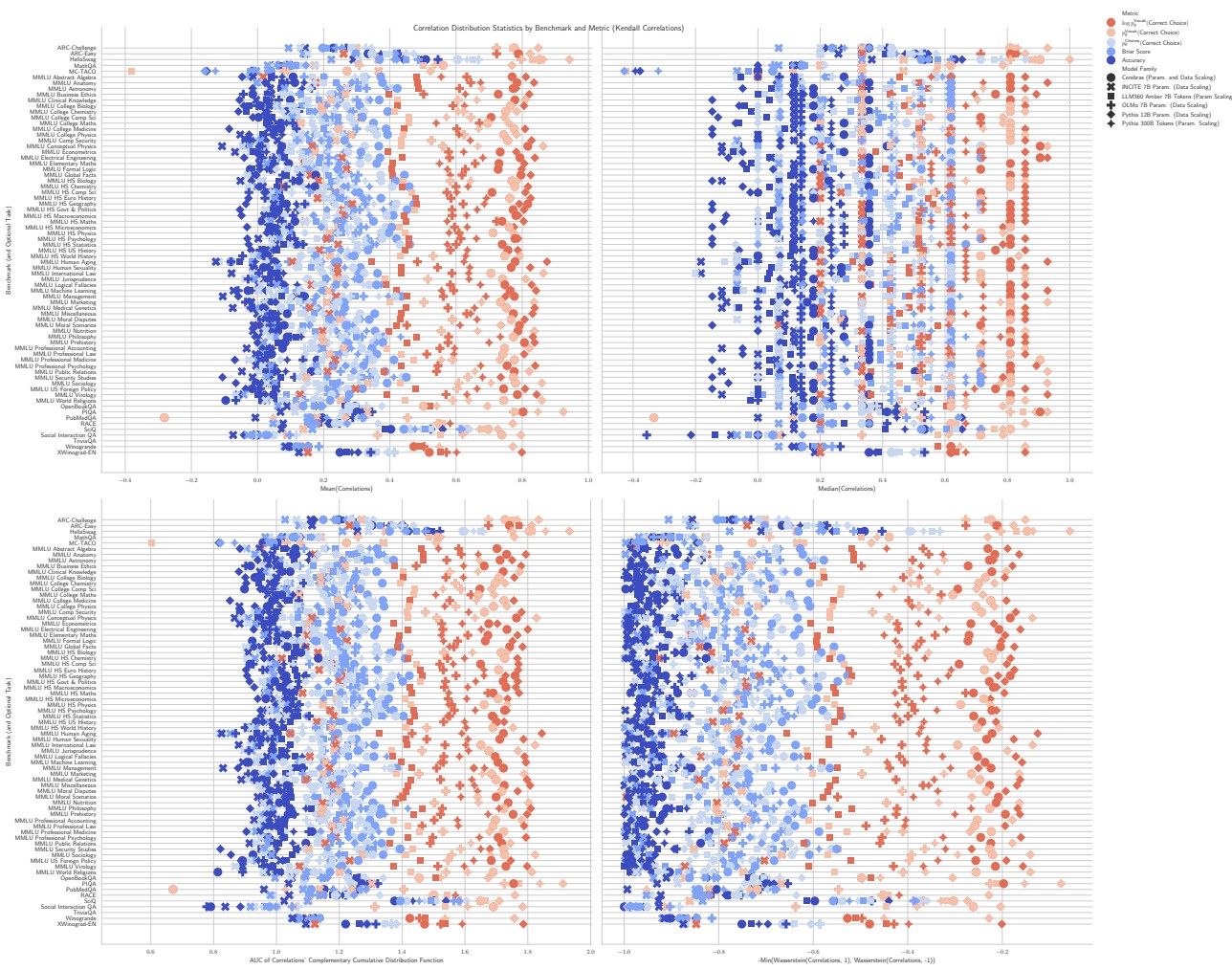

*Figure 9.* **Statistics for empirical distributions of correlations between scores and compute for all benchmarks and model families.** These correlation values were computed with Kendall correlation and are consistent with the main text's results computed with Spearman correlation (Fig. 4): The sequence of transformations from $\log p_\theta^{\text{Vocab}}(\text{Correct Choice}) \rightarrow p_\theta^{\text{Vocab}}(\text{Correct Choice}) \rightarrow p_\theta^{\text{Choices}}(\text{Correct Choice}) \rightarrow \text{Accuracy}$ degrades predictability.

# H. Per-Benchmark Score-Compute Correlation Distributions

## H.1. NLP Benchmark: ARC Challenge (Clark et al., 2018)

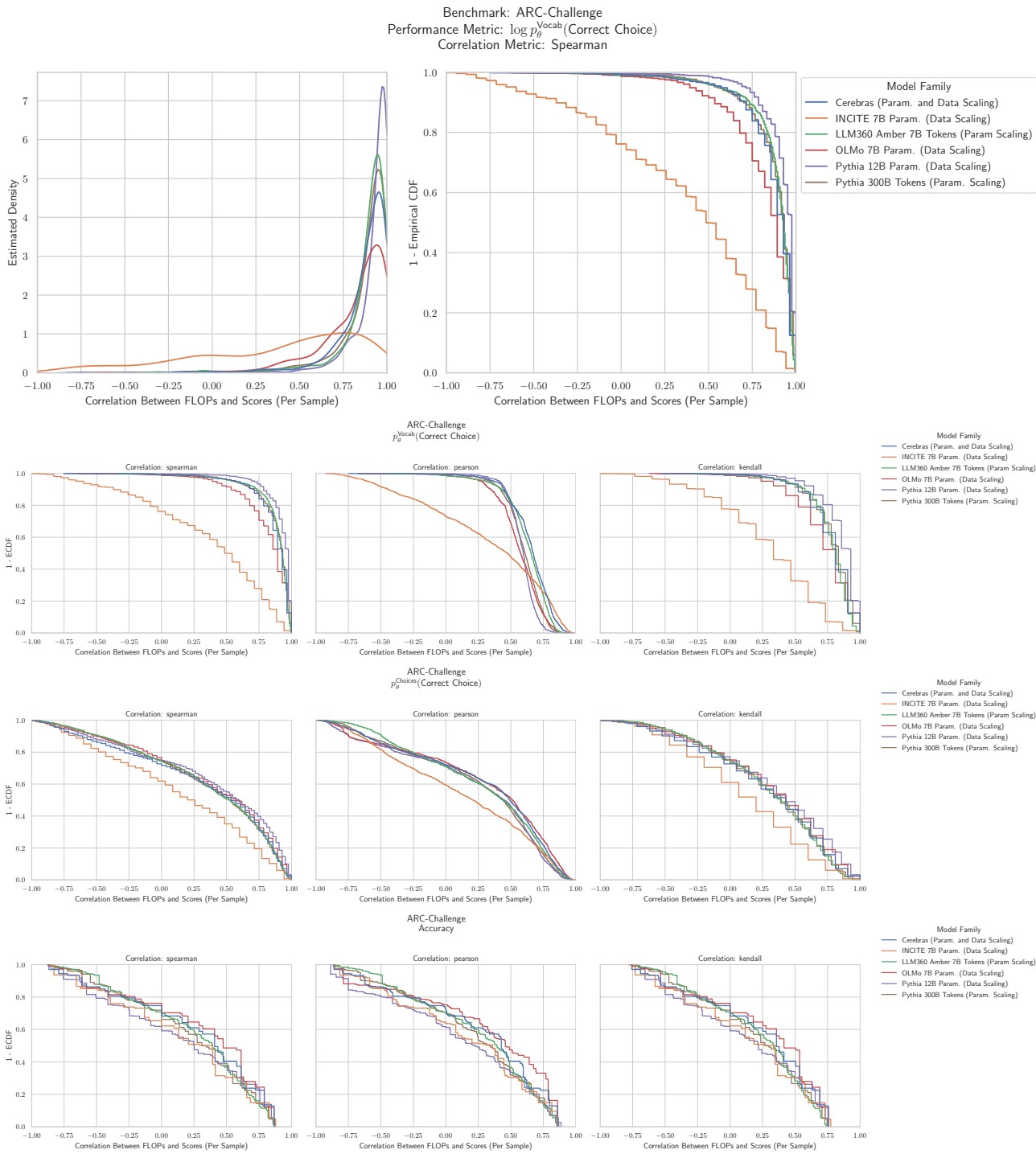

*Figure 10.* **ARC Challenge: Downstream performance is computed via a sequence of transformations that deteriorate correlations between scores and pretraining compute.**

## H.2. NLP Benchmark: ARC Easy (Clark et al., 2018)

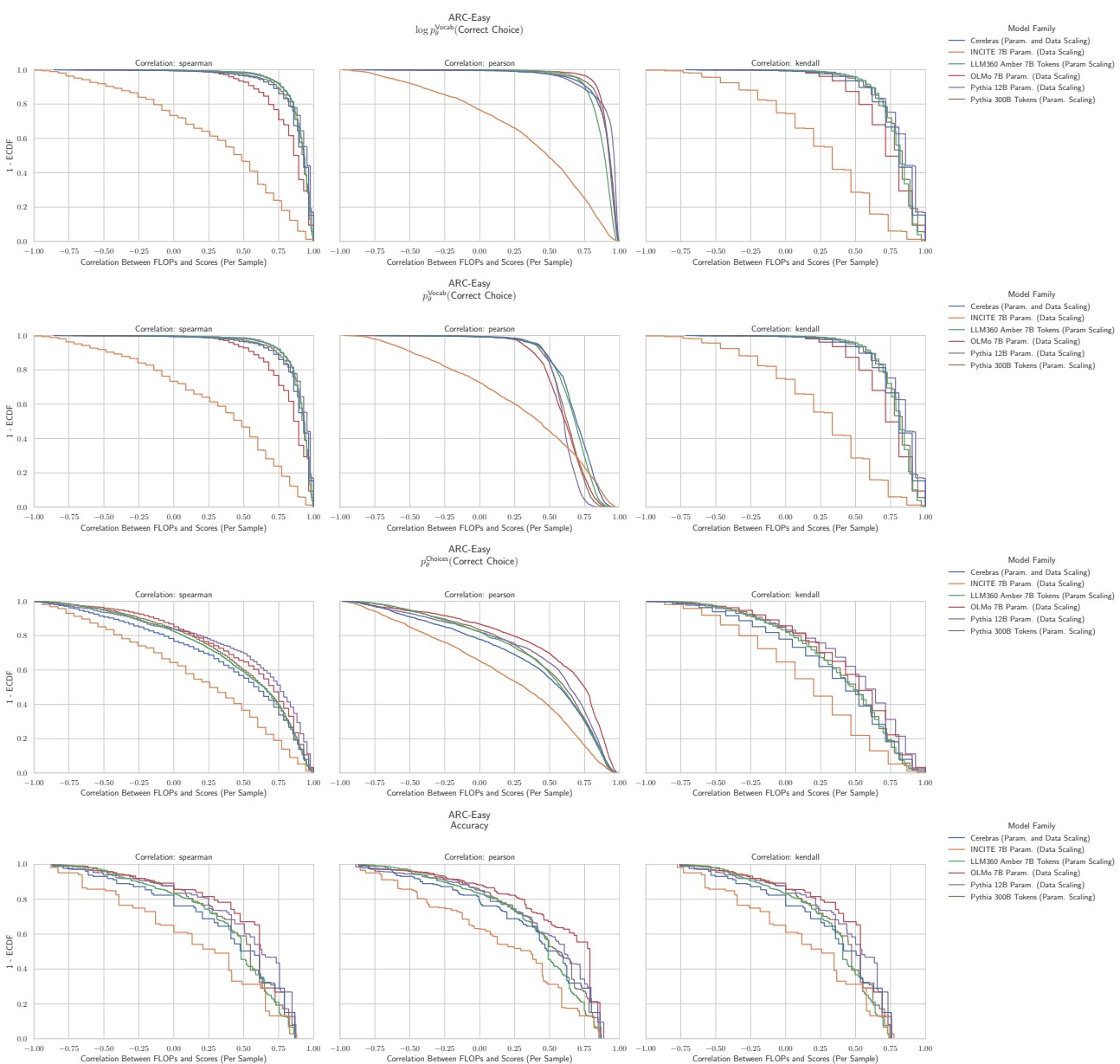

*Figure 11.* **ARC Easy: Downstream performance is computed via a sequence of transformations that deteriorate correlations between scores and pretraining compute.**

## H.3. NLP Benchmark: HellaSwag (Zellers et al., 2019)

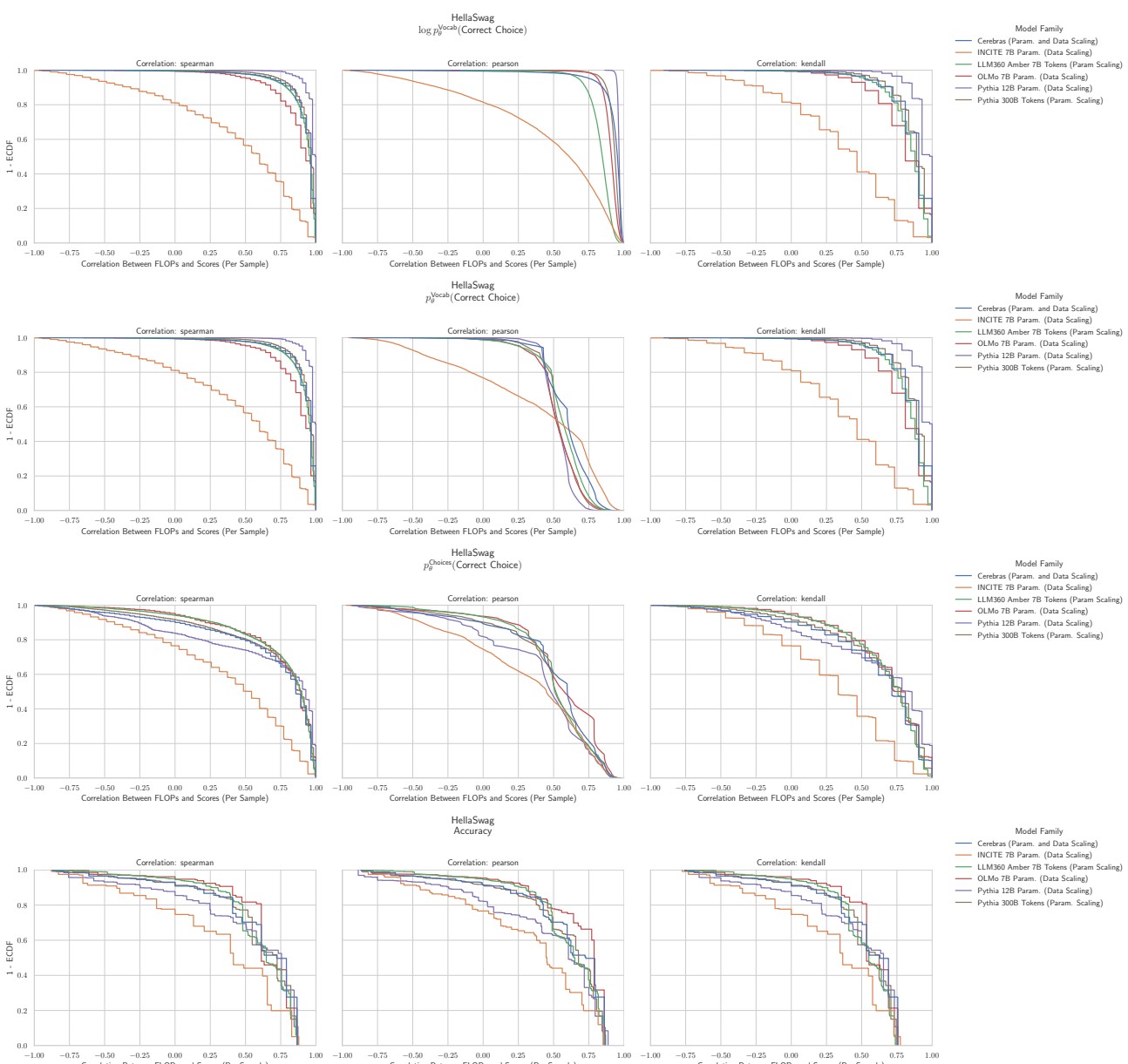

*Figure 12.* **HellaSwag: Downstream performance is computed via a sequence of transformations that deteriorate correlations between scores and pretraining compute.**

## H.4. NLP Benchmark: MathQA (Amini et al., 2019)

*Figure 13.* **HellaSwag: Downstream performance is computed via a sequence of transformations that deteriorate correlations between scores and pretraining compute.**

## H.5. NLP Benchmark: MC TACO (Zhou et al., 2019)

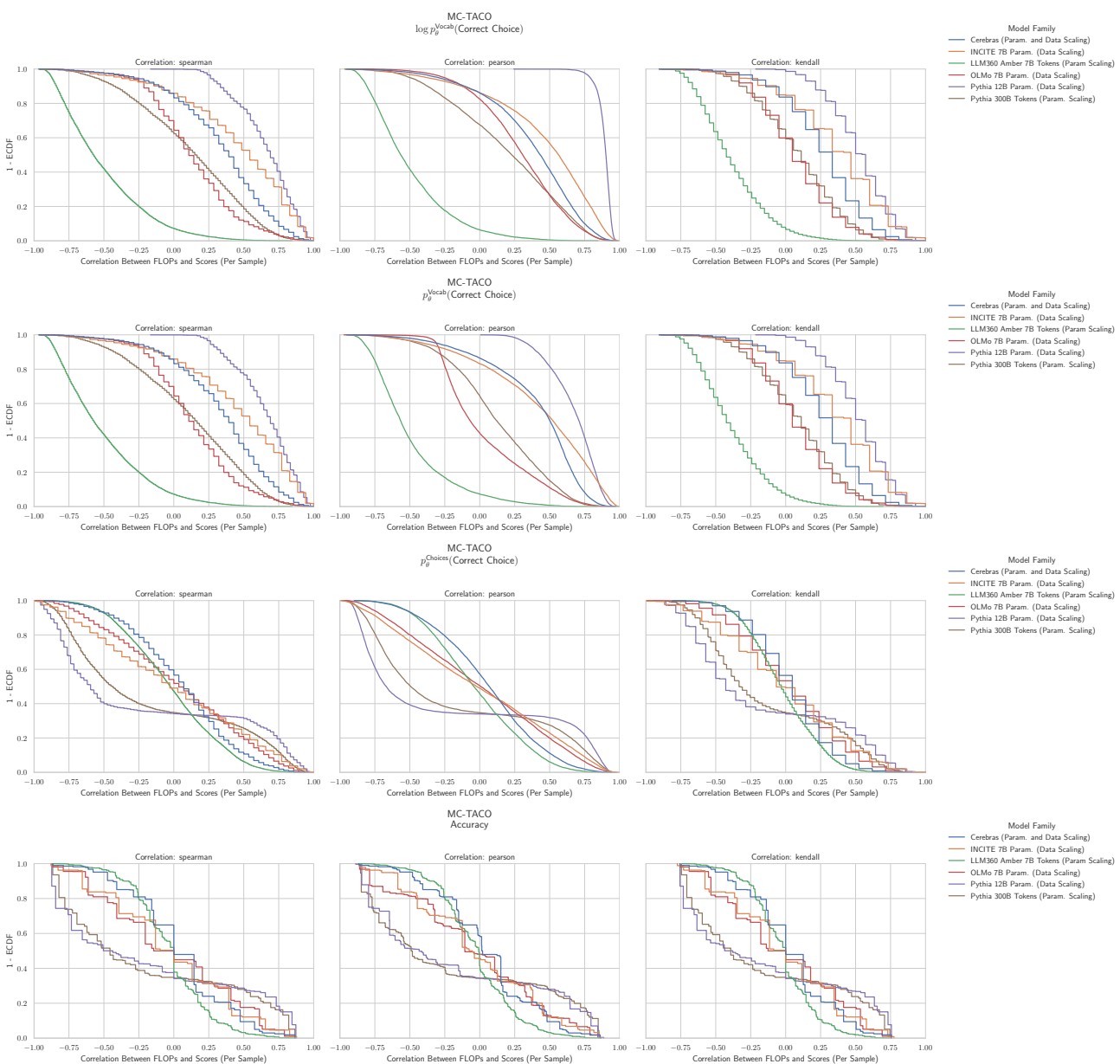

*Figure 14.* **MC TACO: Downstream performance is computed via a sequence of transformations that deteriorate correlations between scores and pretraining compute.**

## H.6. NLP Benchmark: MMLU Abstract Algebra (Hendrycks et al., 2020)

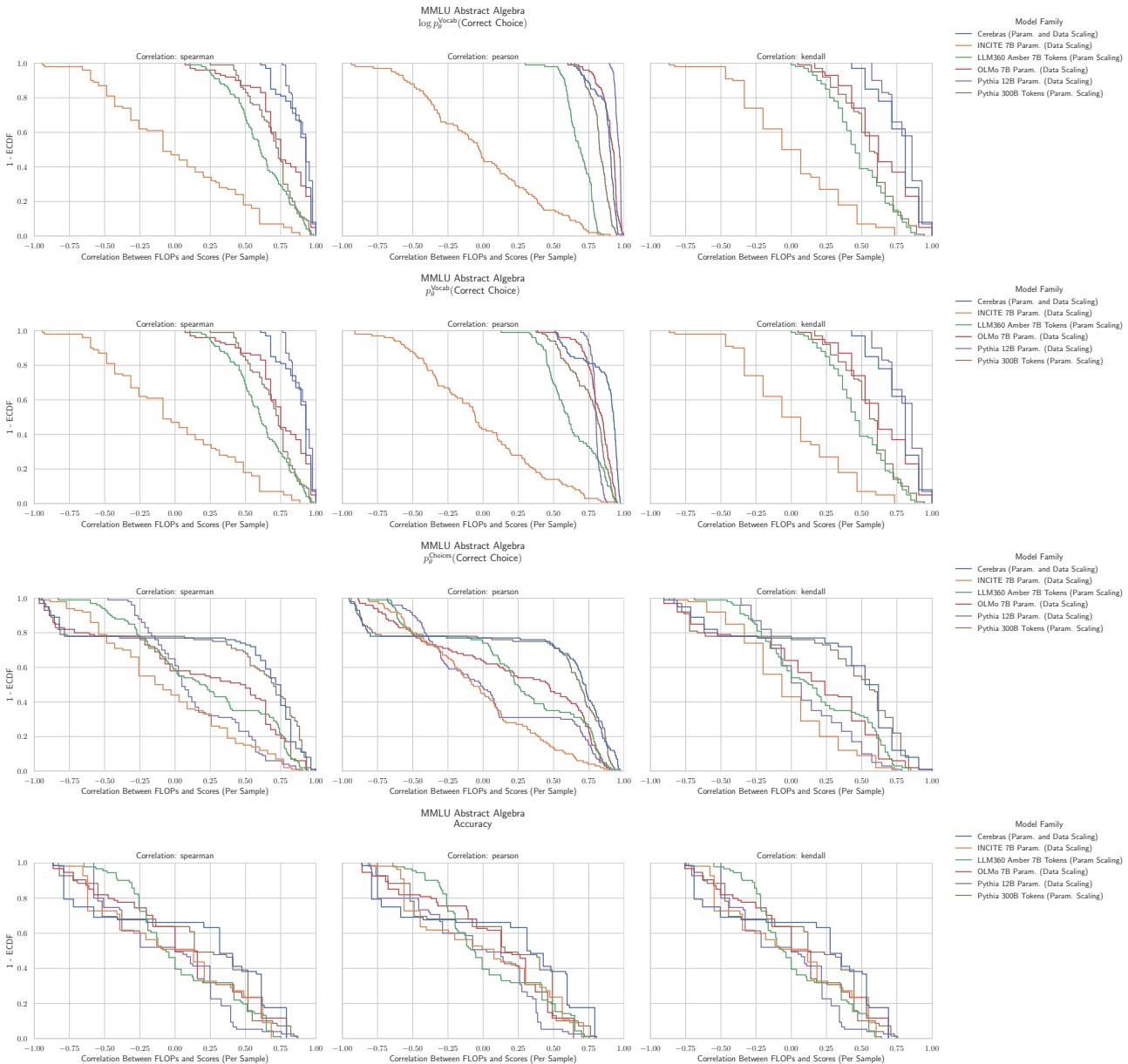

*Figure 15.* **MMLU Abstract Algebra: Downstream performance is computed via a sequence of transformations that deteriorate correlations between scores and pretraining compute.**

## H.7. NLP Benchmark: MMLU Anatomy (Hendrycks et al., 2020)

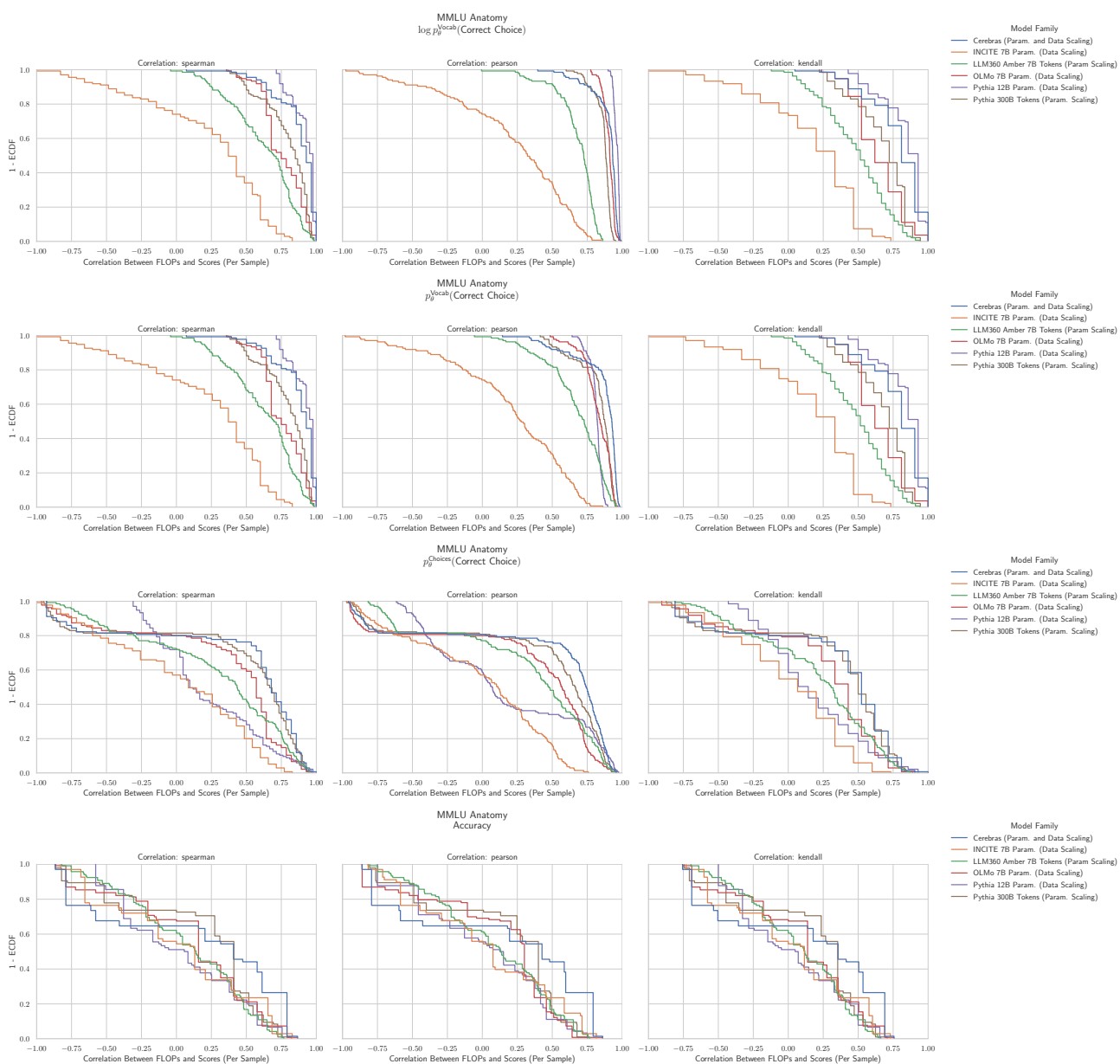

*Figure 16.* **MMLU Anatomy: Downstream performance is computed via a sequence of transformations that deteriorate correlations between scores and pretraining compute.**

## H.8. NLP Benchmark: MMLU Astronomy (Hendrycks et al., 2020)

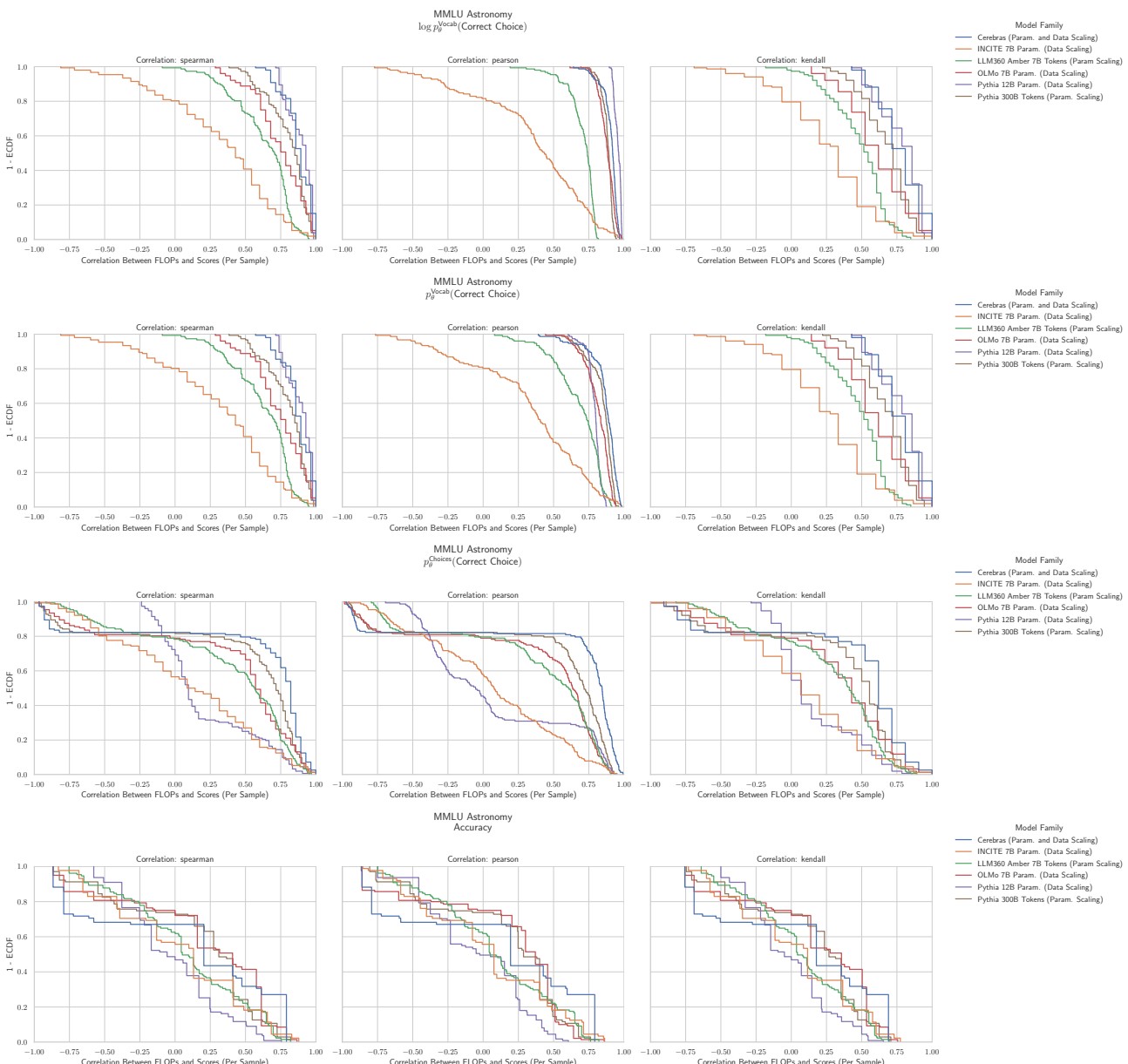

*Figure 17.* **MMLU Astronomy: Downstream performance is computed via a sequence of transformations that deteriorate correlations between scores and pretraining compute.**

## H.9. NLP Benchmark: MMLU Business Ethics (Hendrycks et al., 2020)

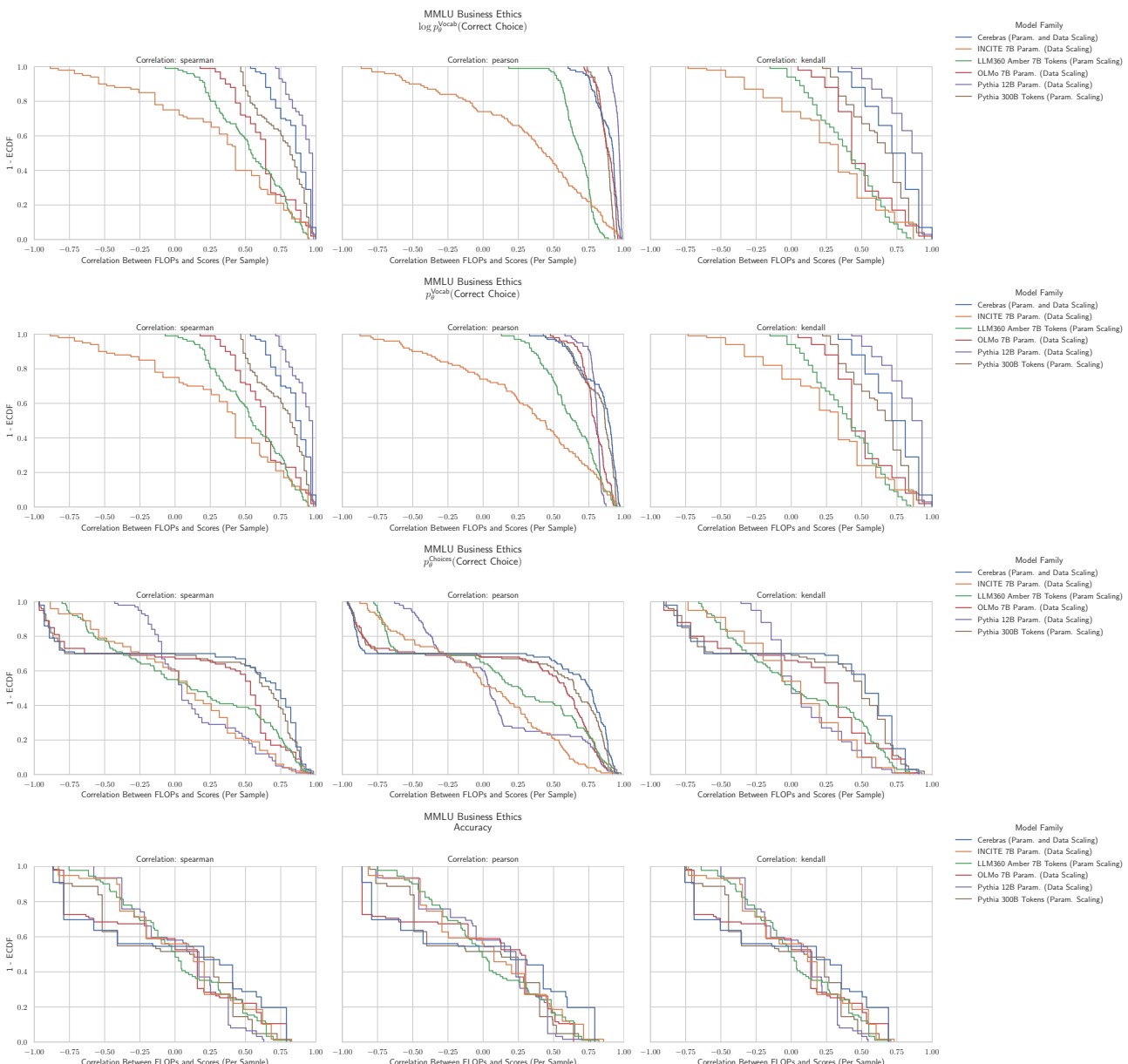

*Figure 18.* **MMLU Business Ethics: Downstream performance is computed via a sequence of transformations that deteriorate correlations between scores and pretraining compute.**

## H.10. NLP Benchmark: MMLU Clinical Knowledge (Hendrycks et al., 2020)

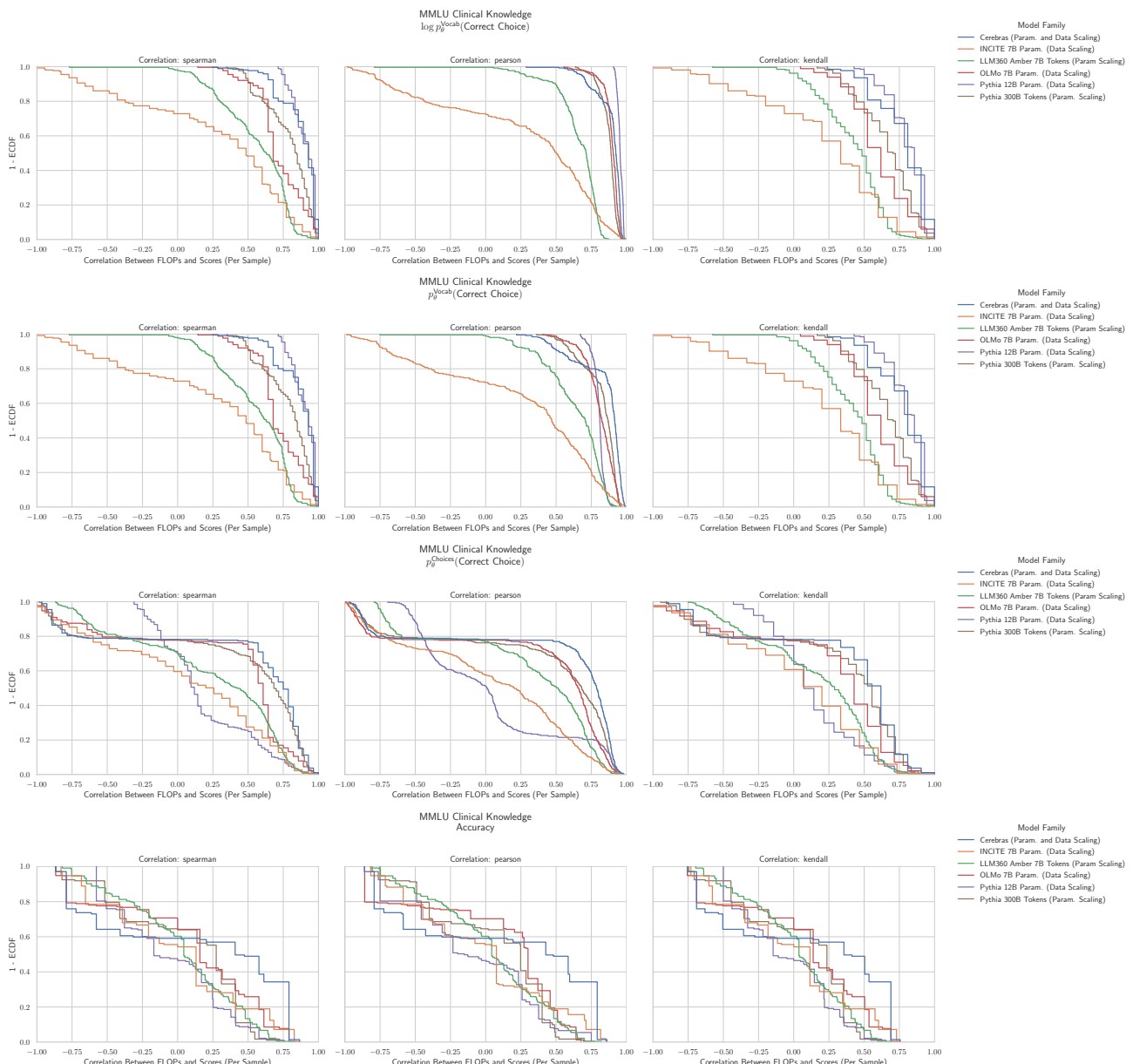

*Figure 19.* **MMLU Clinical Knowledge: Downstream performance is computed via a sequence of transformations that deteriorate correlations between scores and pretraining compute.**

## H.11. NLP Benchmark: MMLU College Biology (Hendrycks et al., 2020)

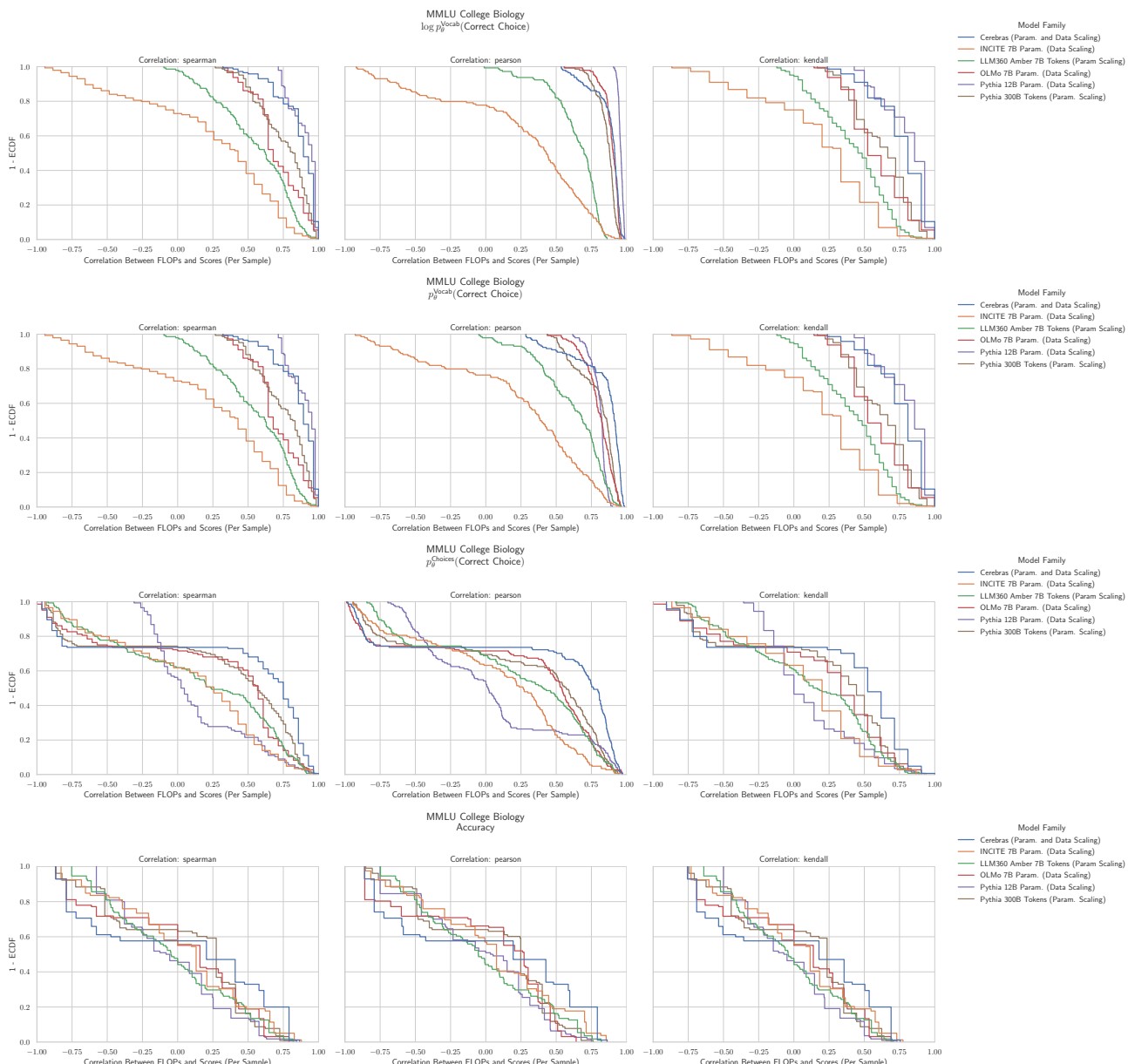

*Figure 20.* **MMLU College Biology: Downstream performance is computed via a sequence of transformations that deteriorate correlations between scores and pretraining compute.**

## H.12. NLP Benchmark: MMLU College Chemistry (Hendrycks et al., 2020)

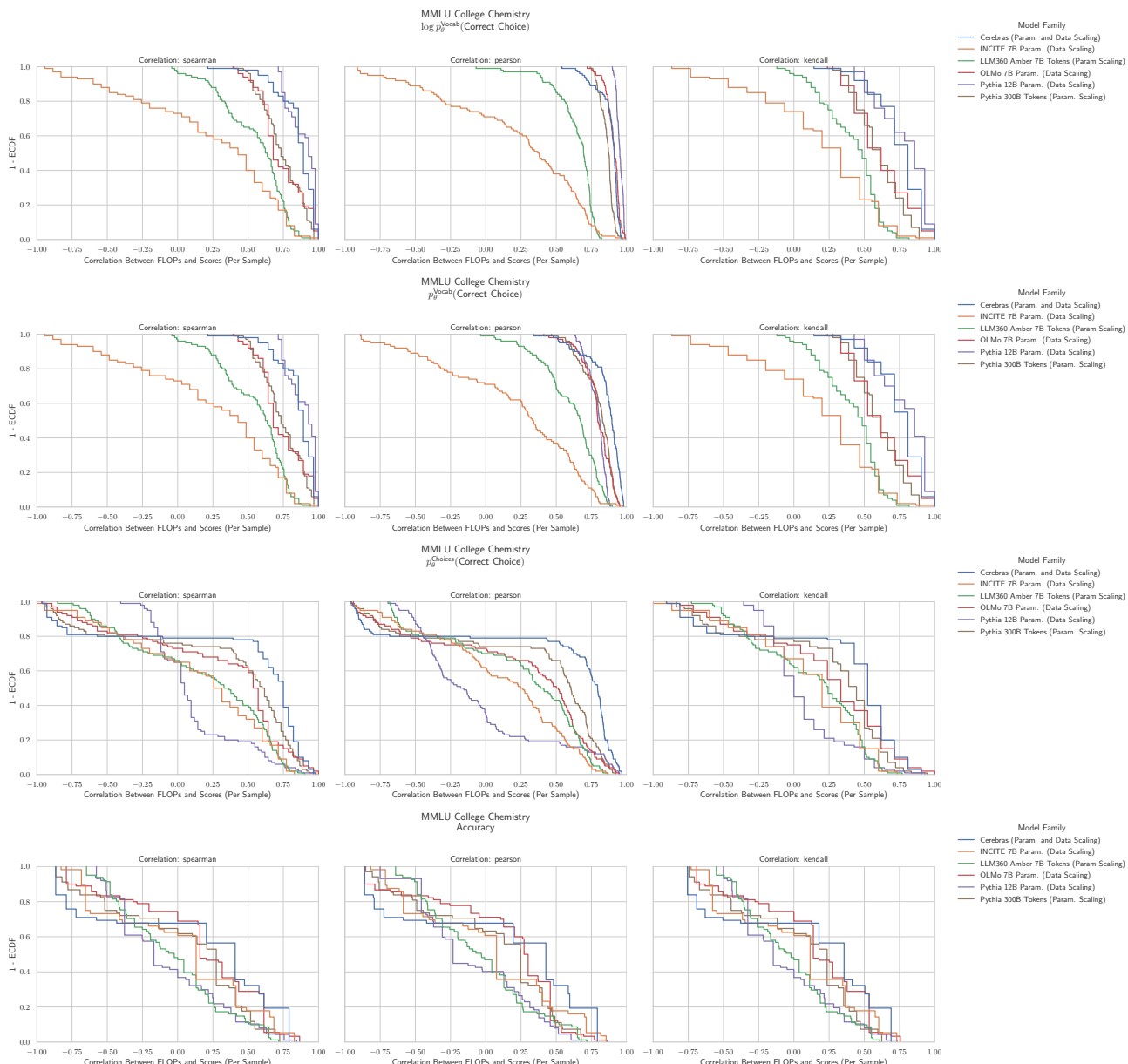

*Figure 21.* **MMLU College Chemistry: Downstream performance is computed via a sequence of transformations that deteriorate correlations between scores and pretraining compute.**

## H.13. NLP Benchmark: MMLU College Computer Science (Hendrycks et al., 2020)

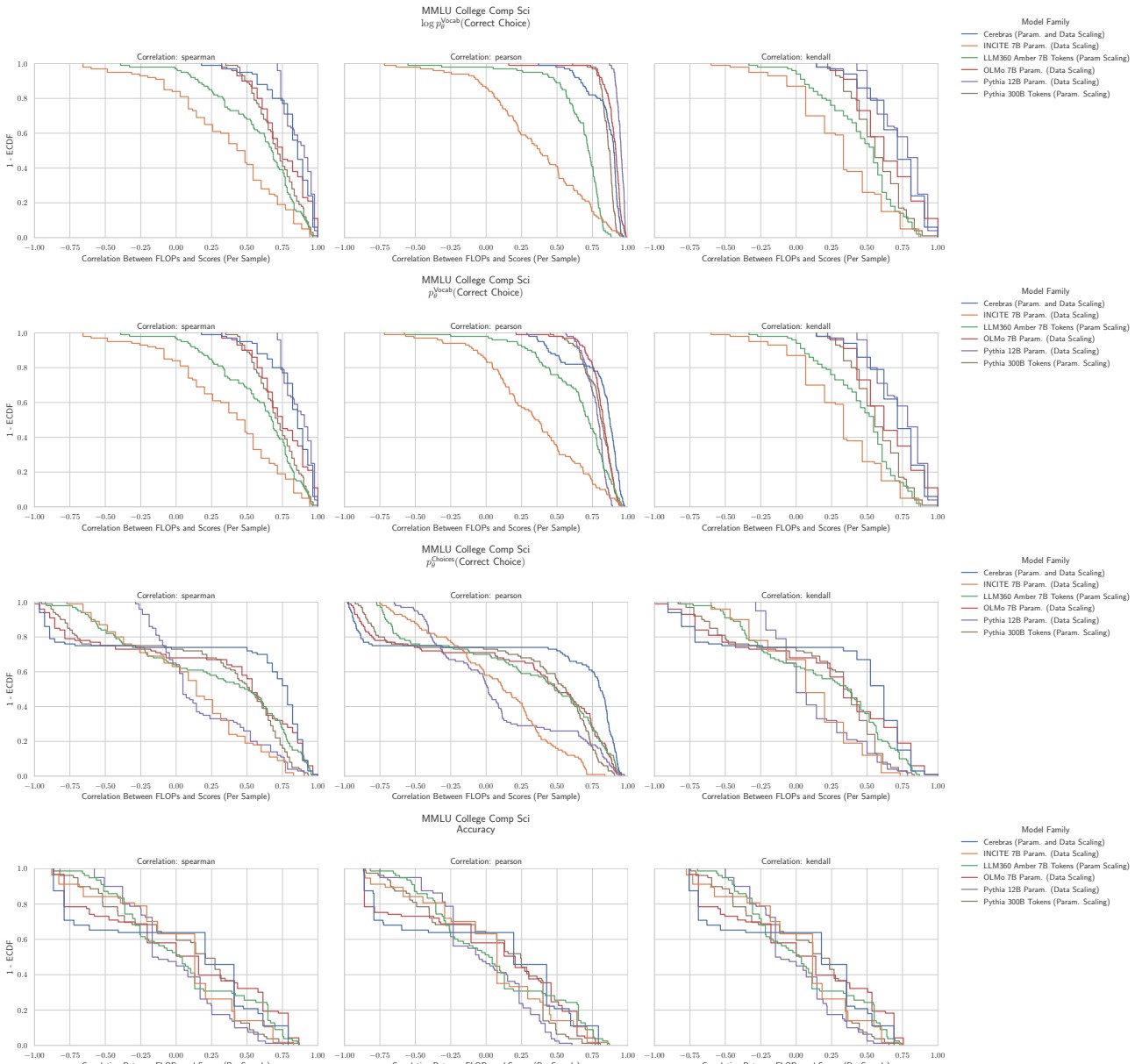

*Figure 22.* **MMLU College Computer Science: Downstream performance is computed via a sequence of transformations that deteriorate correlations between scores and pretraining compute.**

## H.14. NLP Benchmark: MMLU College Mathematics (Hendrycks et al., 2020)

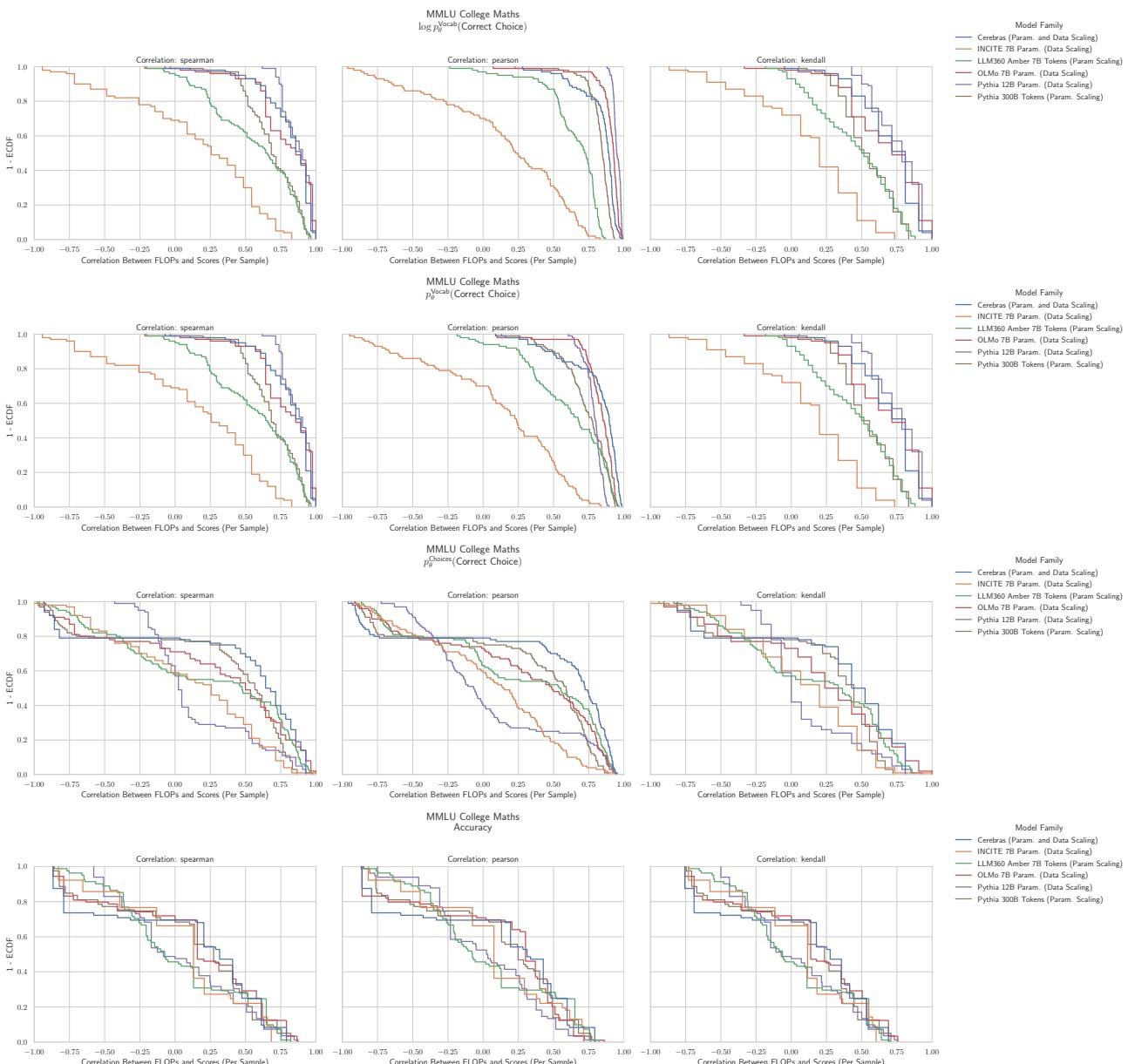

*Figure 23.* **MMLU College Mathematics: Downstream performance is computed via a sequence of transformations that deteriorate correlations between scores and pretraining compute.**

## H.15. NLP Benchmark: MMLU College Medicine (Hendrycks et al., 2020)

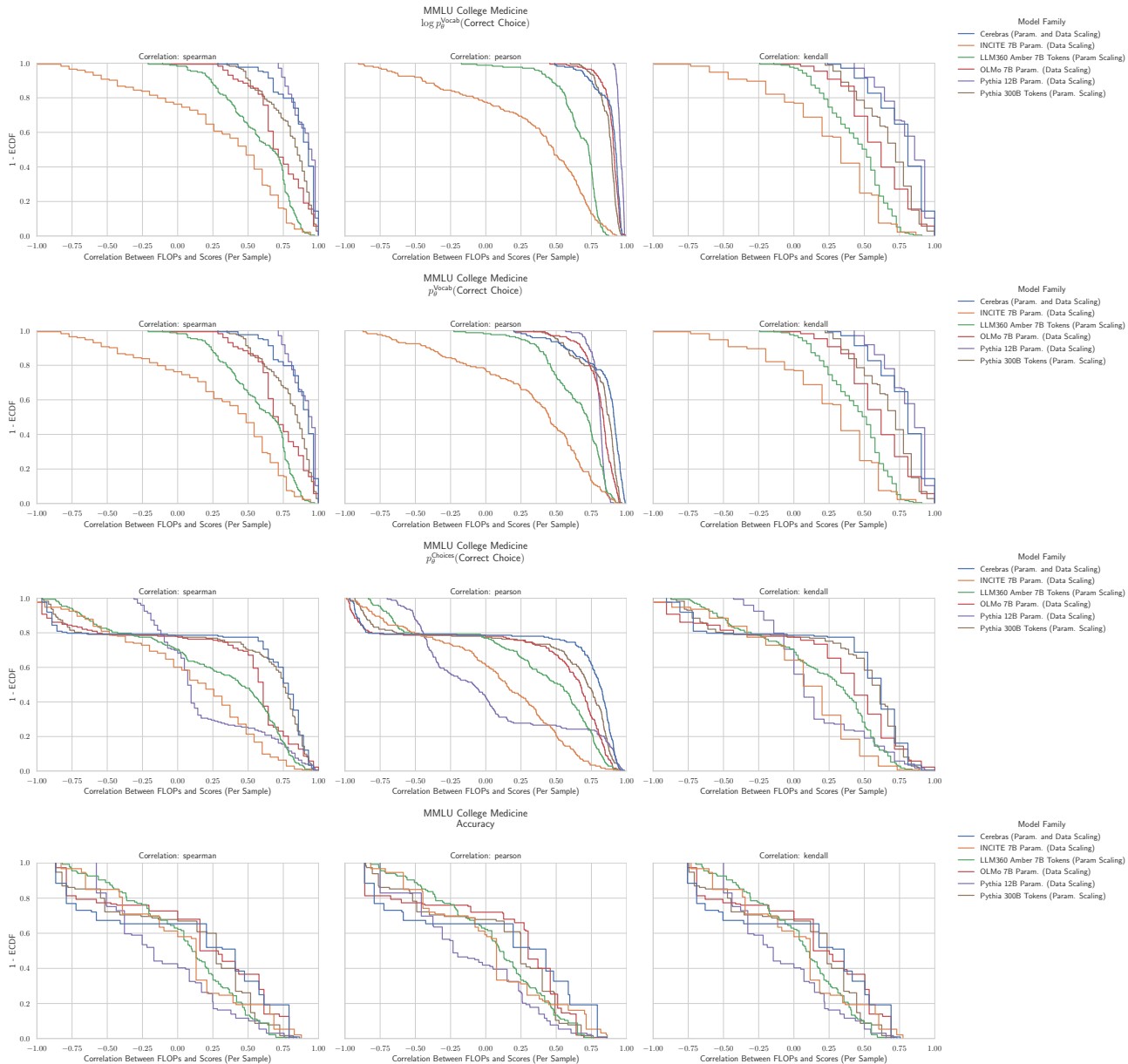

*Figure 24.* **MMLU College Medicine: Downstream performance is computed via a sequence of transformations that deteriorate correlations between scores and pretraining compute.**

## H.16. NLP Benchmark: MMLU College Physics (Hendrycks et al., 2020)

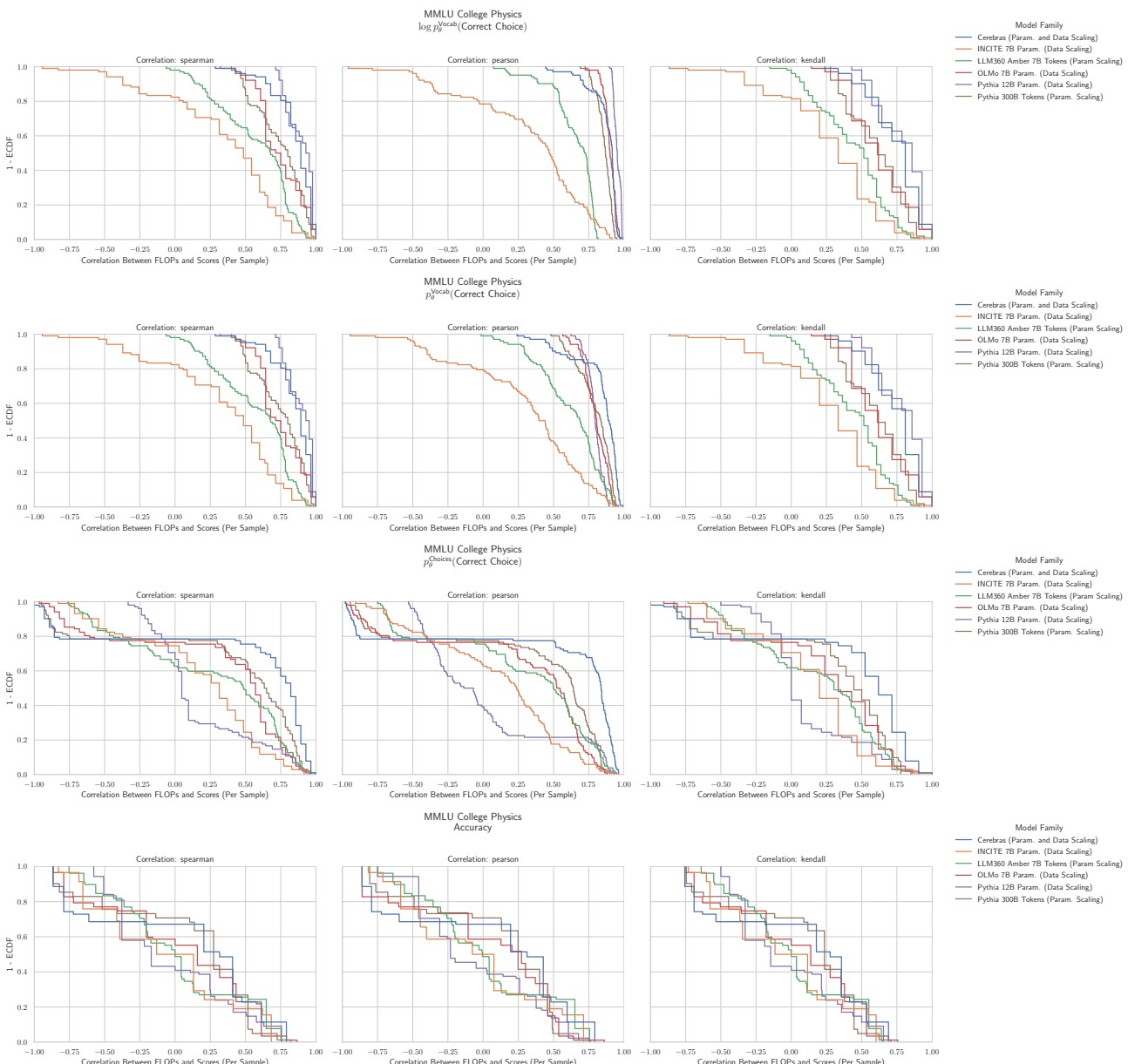

*Figure 25.* **MMLU College Physics: Downstream performance is computed via a sequence of transformations that deteriorate correlations between scores and pretraining compute.**

## H.17. NLP Benchmark: MMLU Computer Security (Hendrycks et al., 2020)

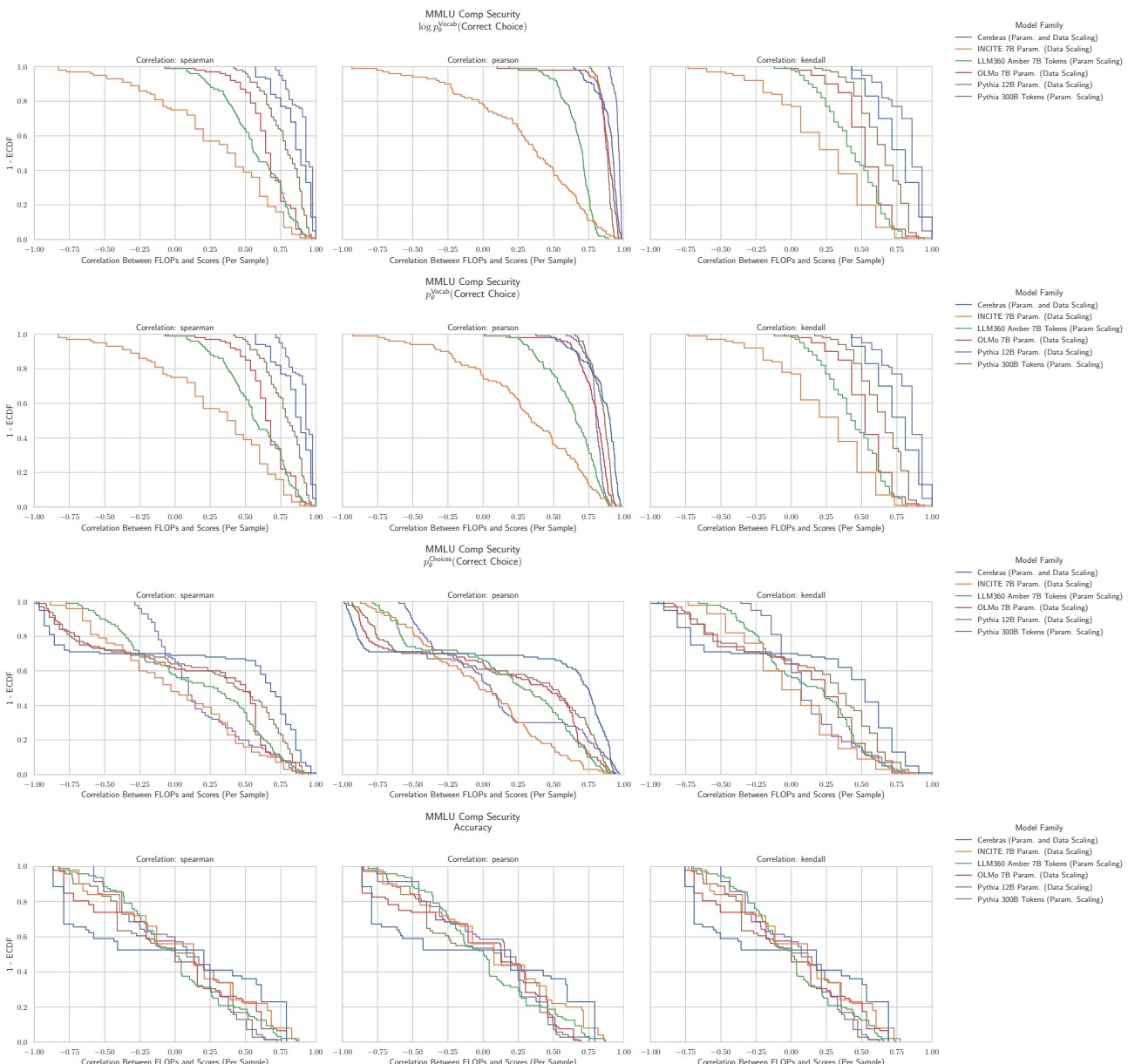

*Figure 26.* **MMLU Computer Security: Downstream performance is computed via a sequence of transformations that deteriorate correlations between scores and pretraining compute.**

## H.18. NLP Benchmark: MMLU Conceptual Physics (Hendrycks et al., 2020)

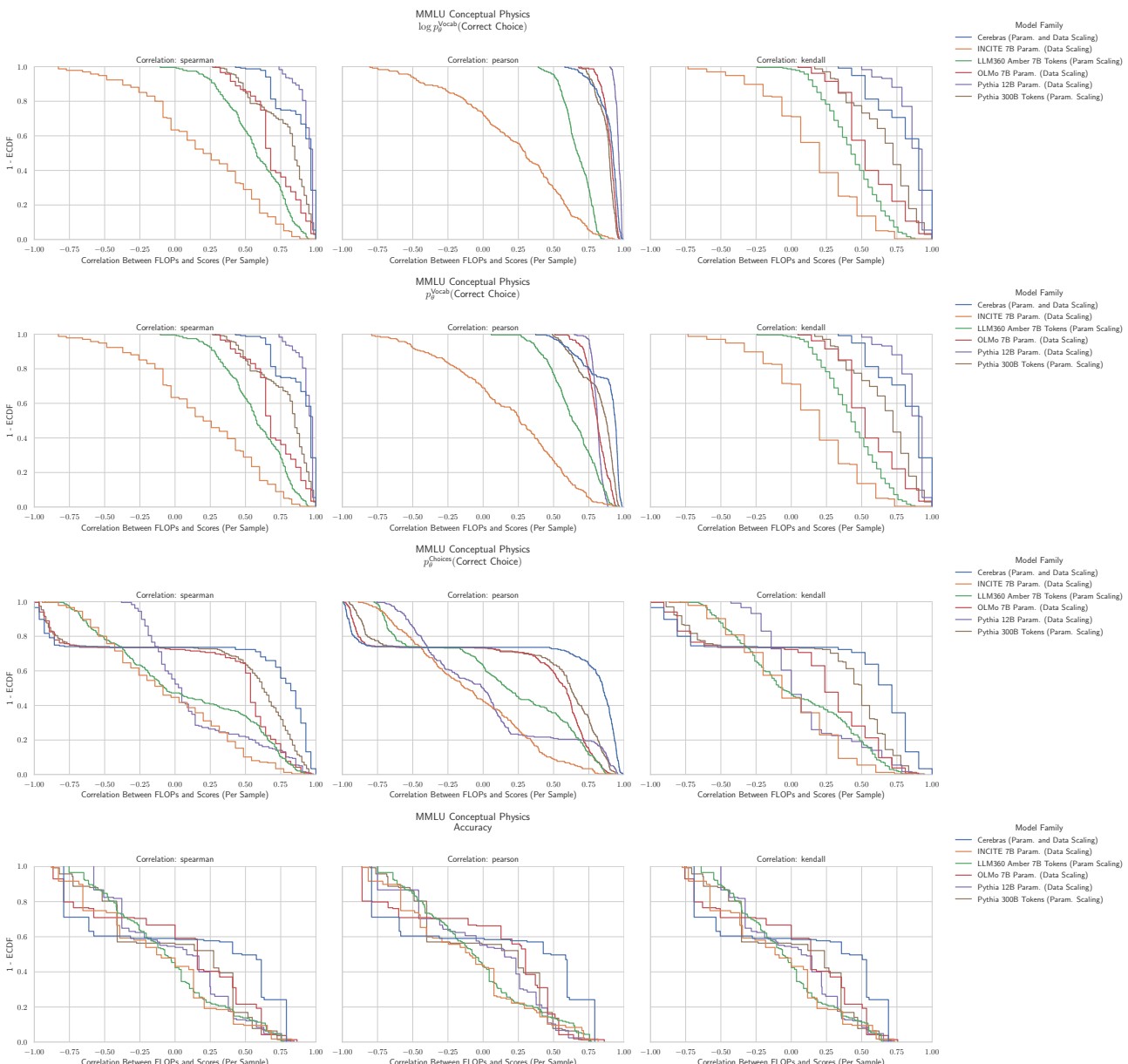

*Figure 27.* **MMLU Conceptual Physics: Downstream performance is computed via a sequence of transformations that deteriorate correlations between scores and pretraining compute.**

## H.19. NLP Benchmark: MMLU Econometrics (Hendrycks et al., 2020)

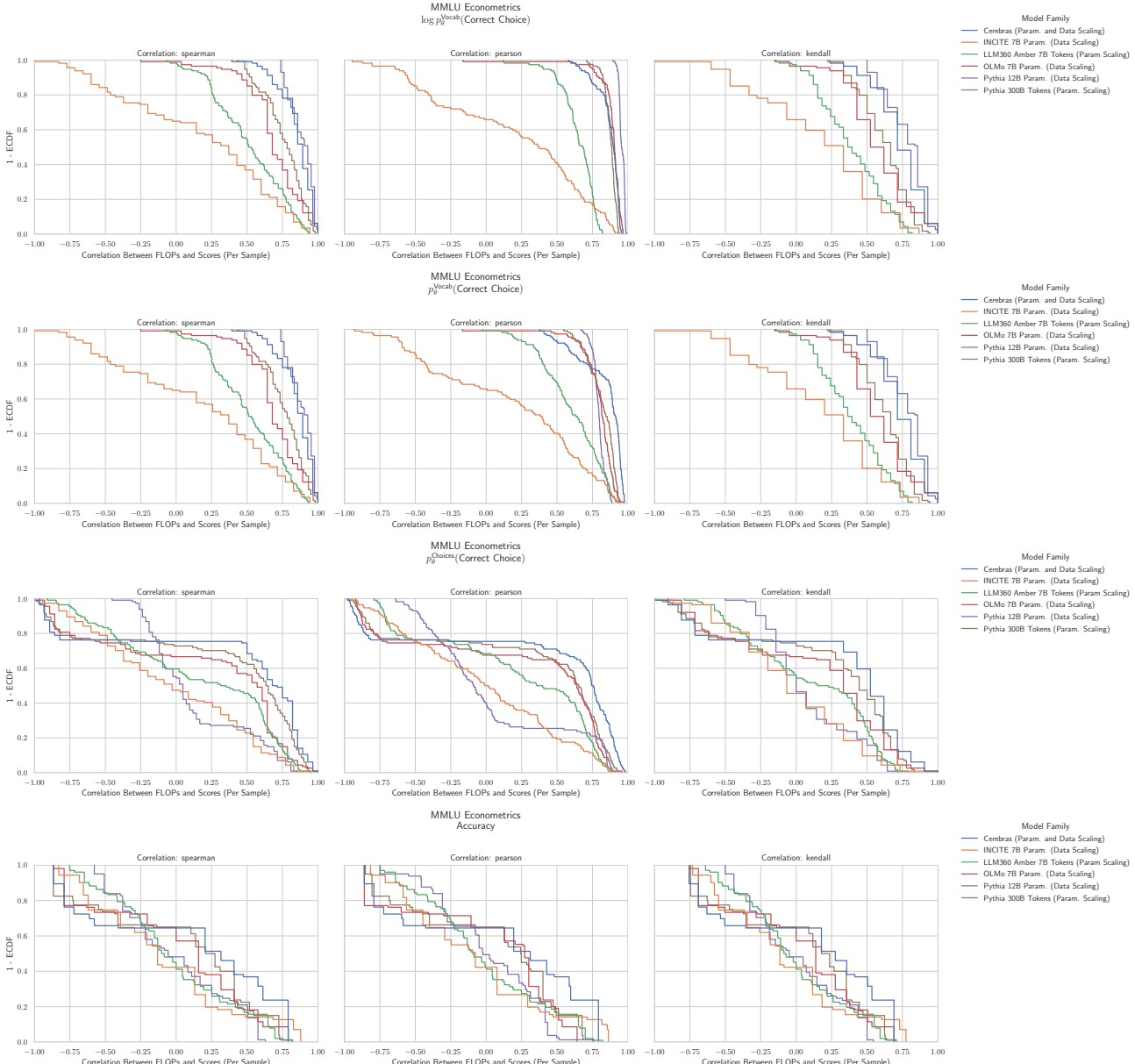

*Figure 28.* **MMLU Econometrics: Downstream performance is computed via a sequence of transformations that deteriorate correlations between scores and pretraining compute.**

## H.20. NLP Benchmark: MMLU Electrical Engineering (Hendrycks et al., 2020)

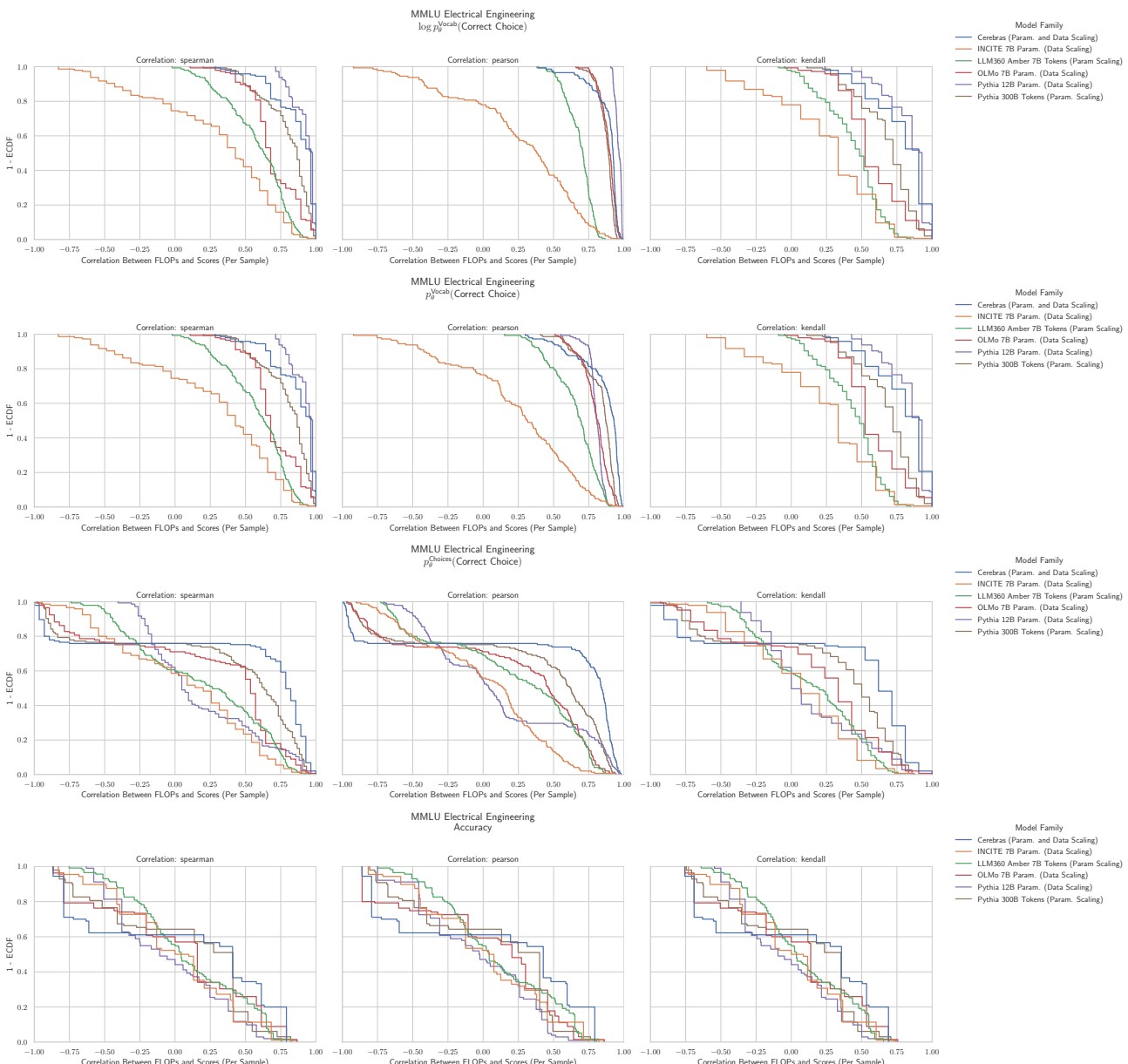

*Figure 29.* **MMLU Electrical Engineering: Downstream performance is computed via a sequence of transformations that deteriorate correlations between scores and pretraining compute.**

## H.21. NLP Benchmark: MMLU Elementary Mathematics (Hendrycks et al., 2020)

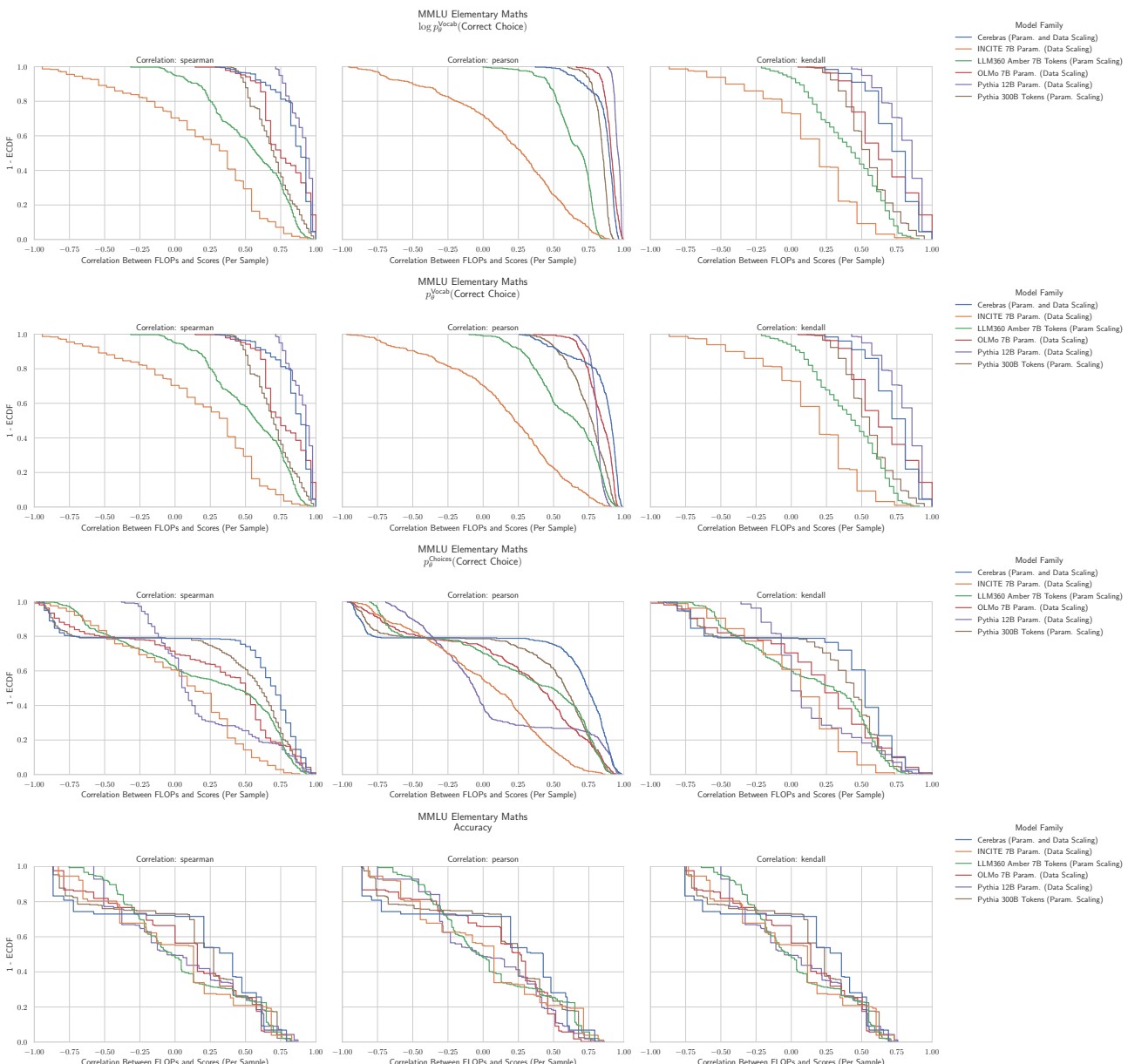

*Figure 30.* **MMLU Elementary Mathematics: Downstream performance is computed via a sequence of transformations that deteriorate correlations between scores and pretraining compute.**

## H.22. NLP Benchmark: MMLU Formal Logic (Hendrycks et al., 2020)

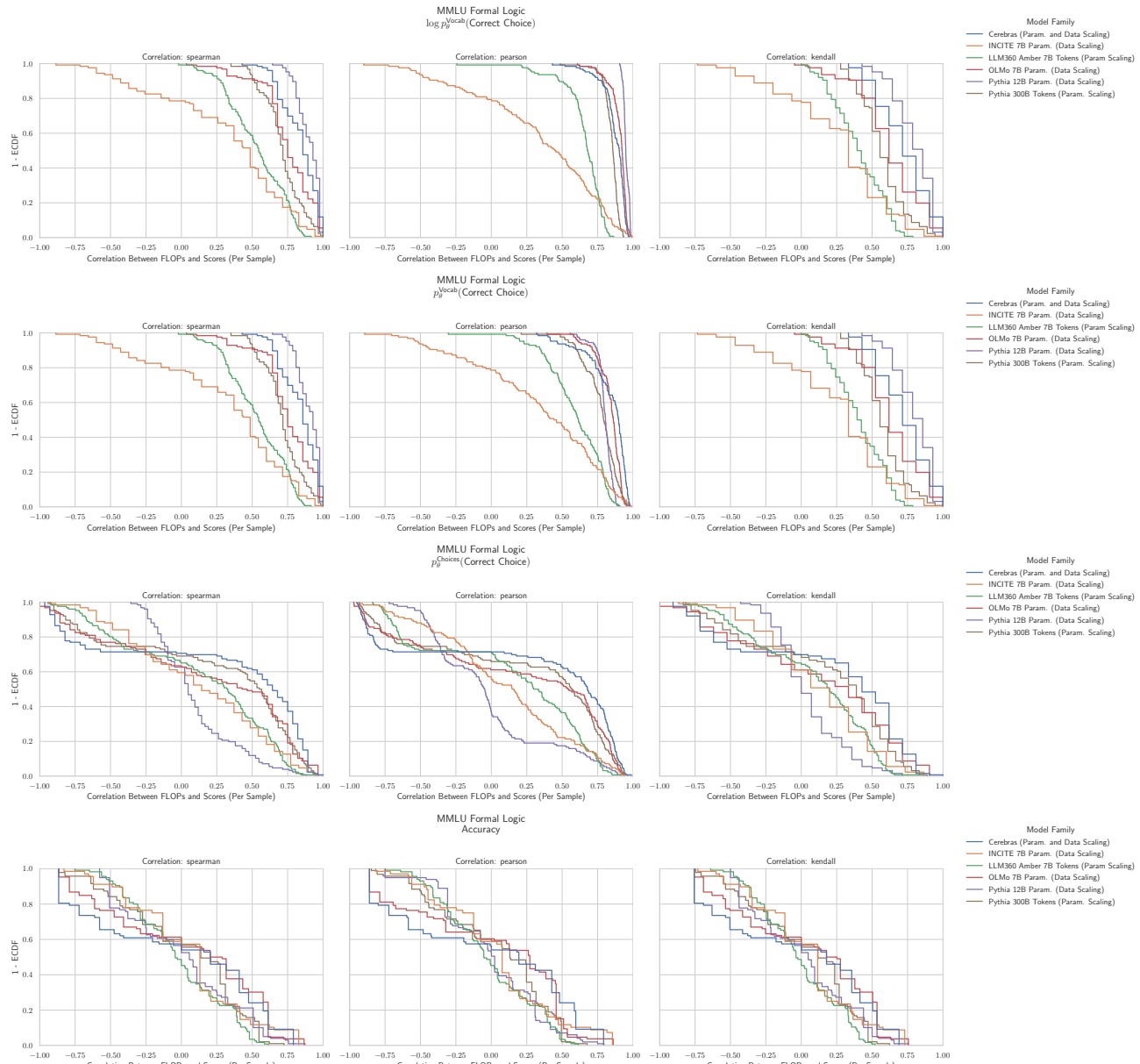

*Figure 31.* **MMLU Formal Logic: Downstream performance is computed via a sequence of transformations that deteriorate correlations between scores and pretraining compute.**

## H.23. NLP Benchmark: MMLU Global Facts (Hendrycks et al., 2020)

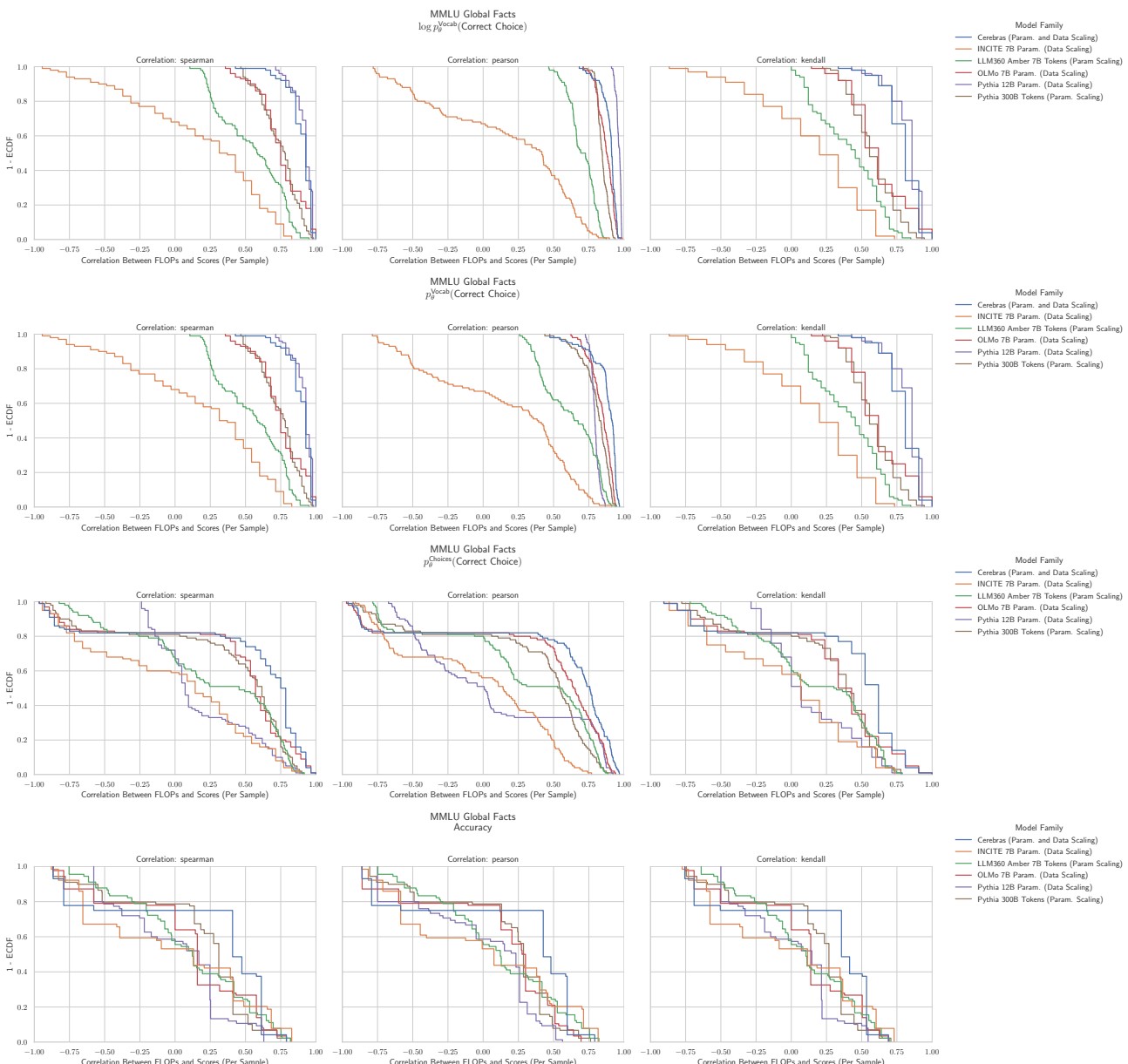

*Figure 32.* **MMLU Global Facts: Downstream performance is computed via a sequence of transformations that deteriorate correlations between scores and pretraining compute.**

## H.24. NLP Benchmark: MMLU High School Biology (Hendrycks et al., 2020)

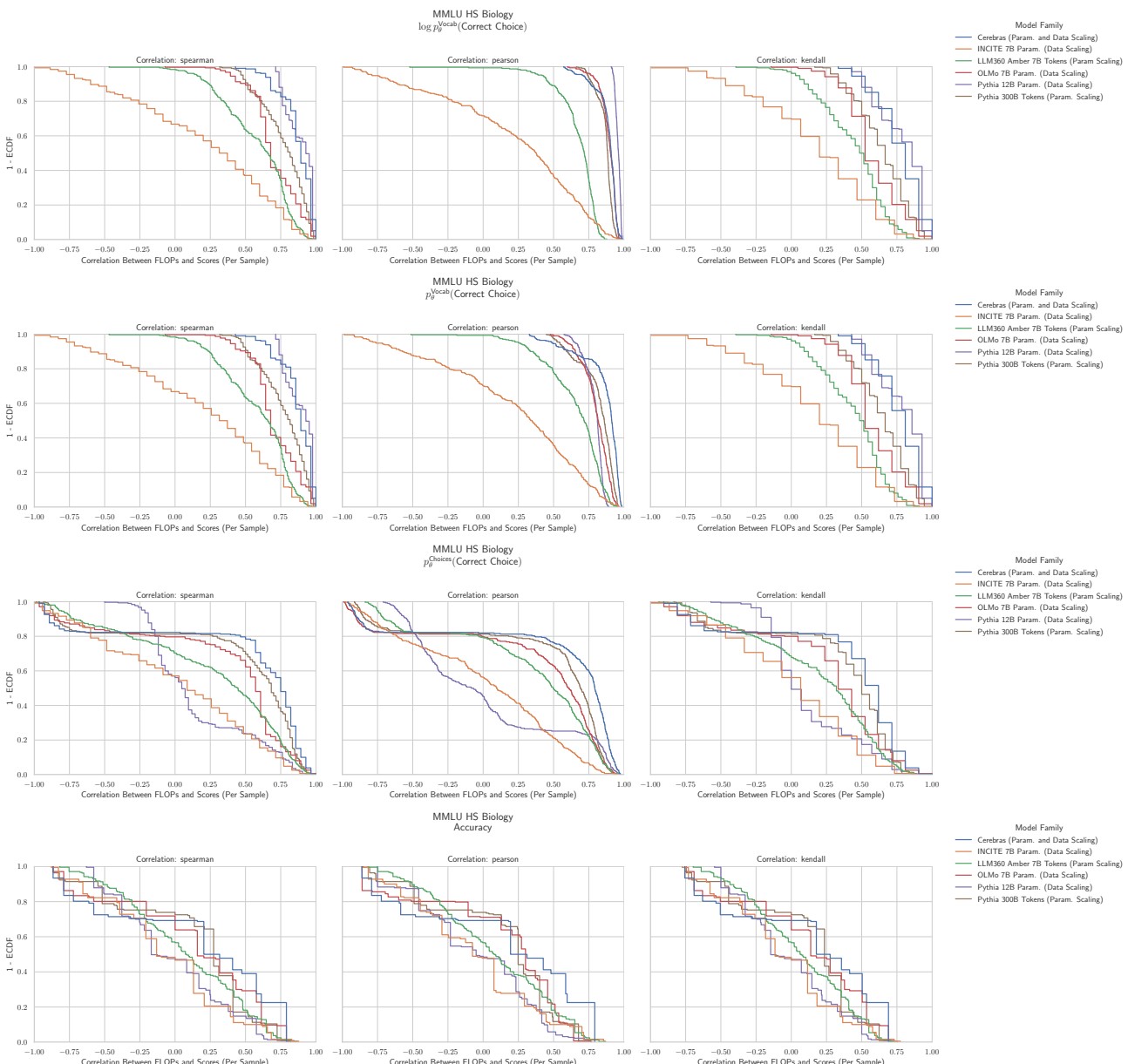

*Figure 33.* **MMLU High School Biology: Downstream performance is computed via a sequence of transformations that deteriorate correlations between scores and pretraining compute.**

## H.25. NLP Benchmark: MMLU High School Chemistry ([Hendrycks et al., 2020](#))

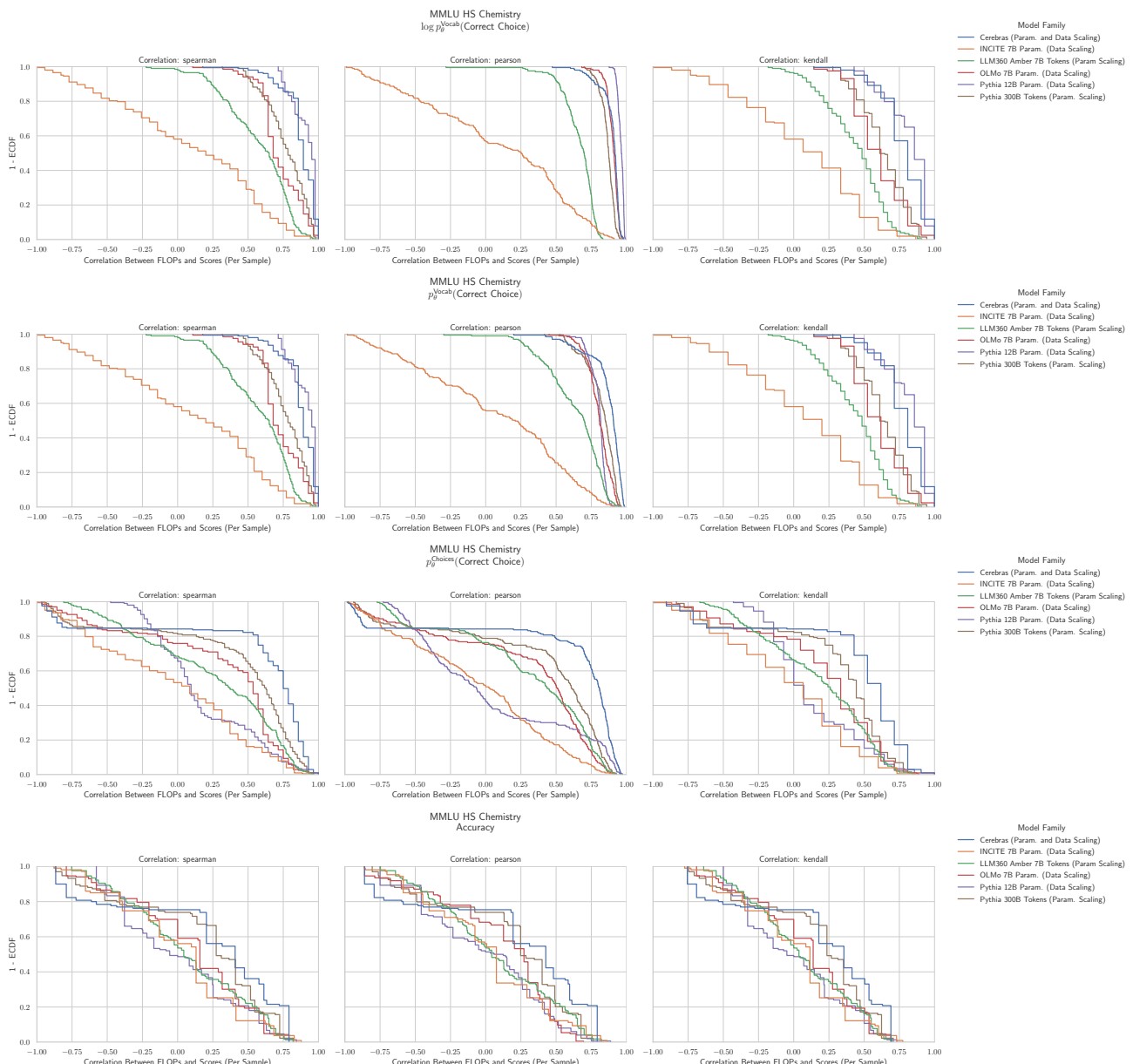

*Figure 34.* **MMLU High School Chemistry: Downstream performance is computed via a sequence of transformations that deteriorate correlations between scores and pretraining compute.**

## H.26. NLP Benchmark: MMLU High School Computer Science (Hendrycks et al., 2020)

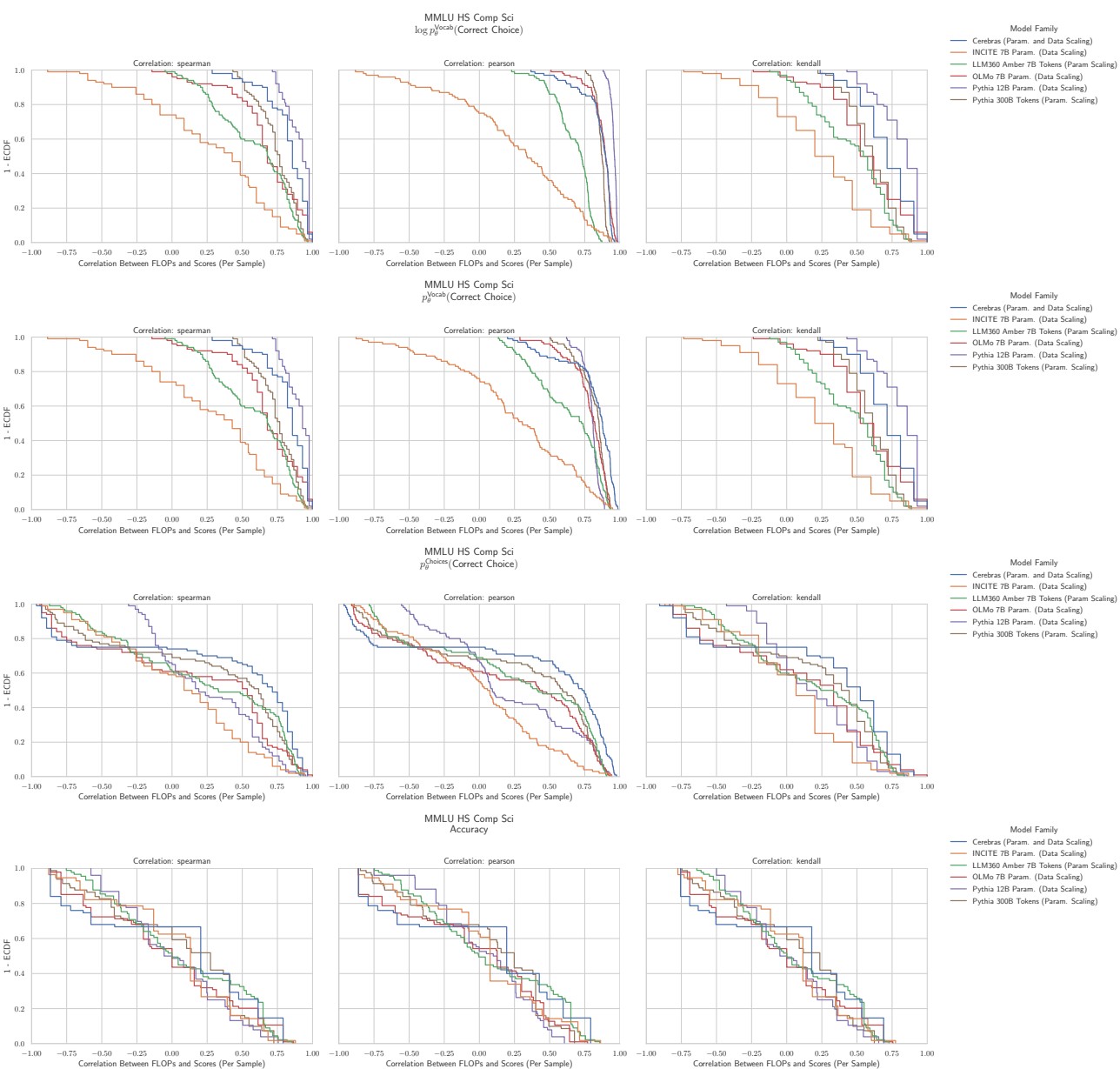

*Figure 35.* **MMLU High School Computer Science: Downstream performance is computed via a sequence of transformations that deteriorate correlations between scores and pretraining compute.**

## H.27. NLP Benchmark: MMLU High School Chemistry (Hendrycks et al., 2020)

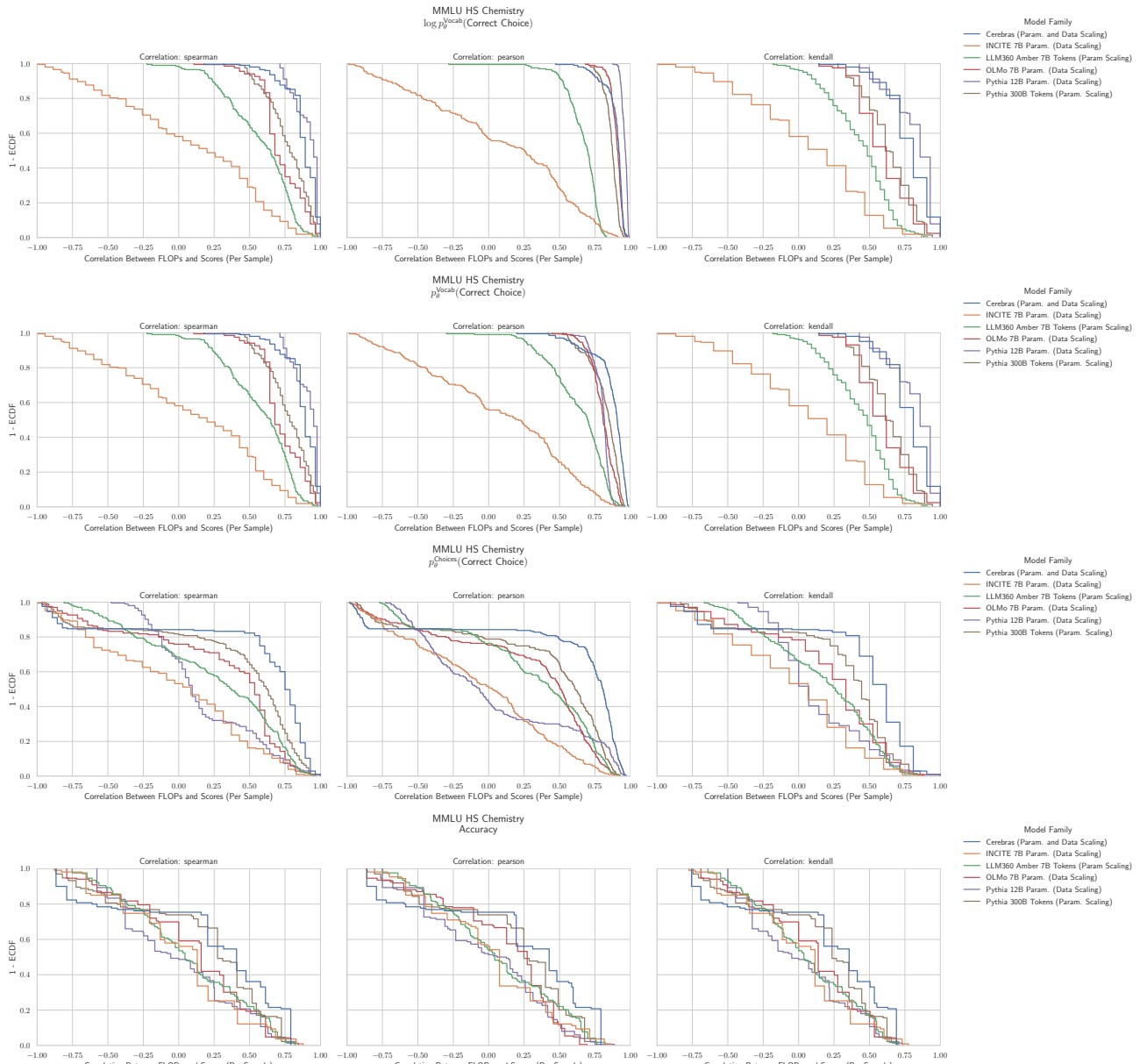

*Figure 36.* **MMLU High School Chemistry: Downstream performance is computed via a sequence of transformations that deteriorate correlations between scores and pretraining compute.**

## H.28. NLP Benchmark: MMLU High School European History ([Hendrycks et al., 2020](#))

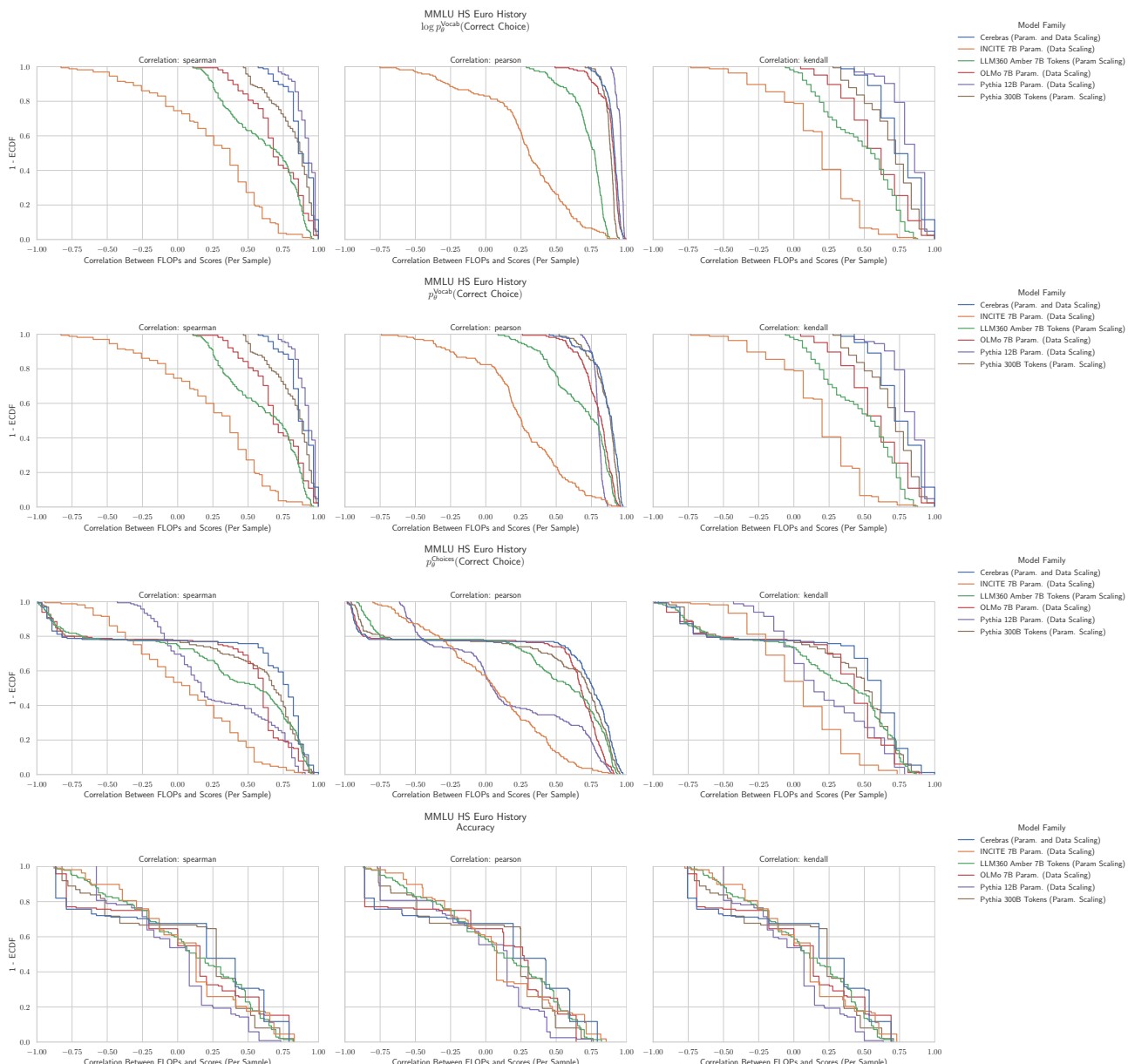

*Figure 37.* **MMLU High School European History: Downstream performance is computed via a sequence of transformations that deteriorate correlations between scores and pretraining compute.**

## H.29. NLP Benchmark: MMLU High School Geography (Hendrycks et al., 2020)

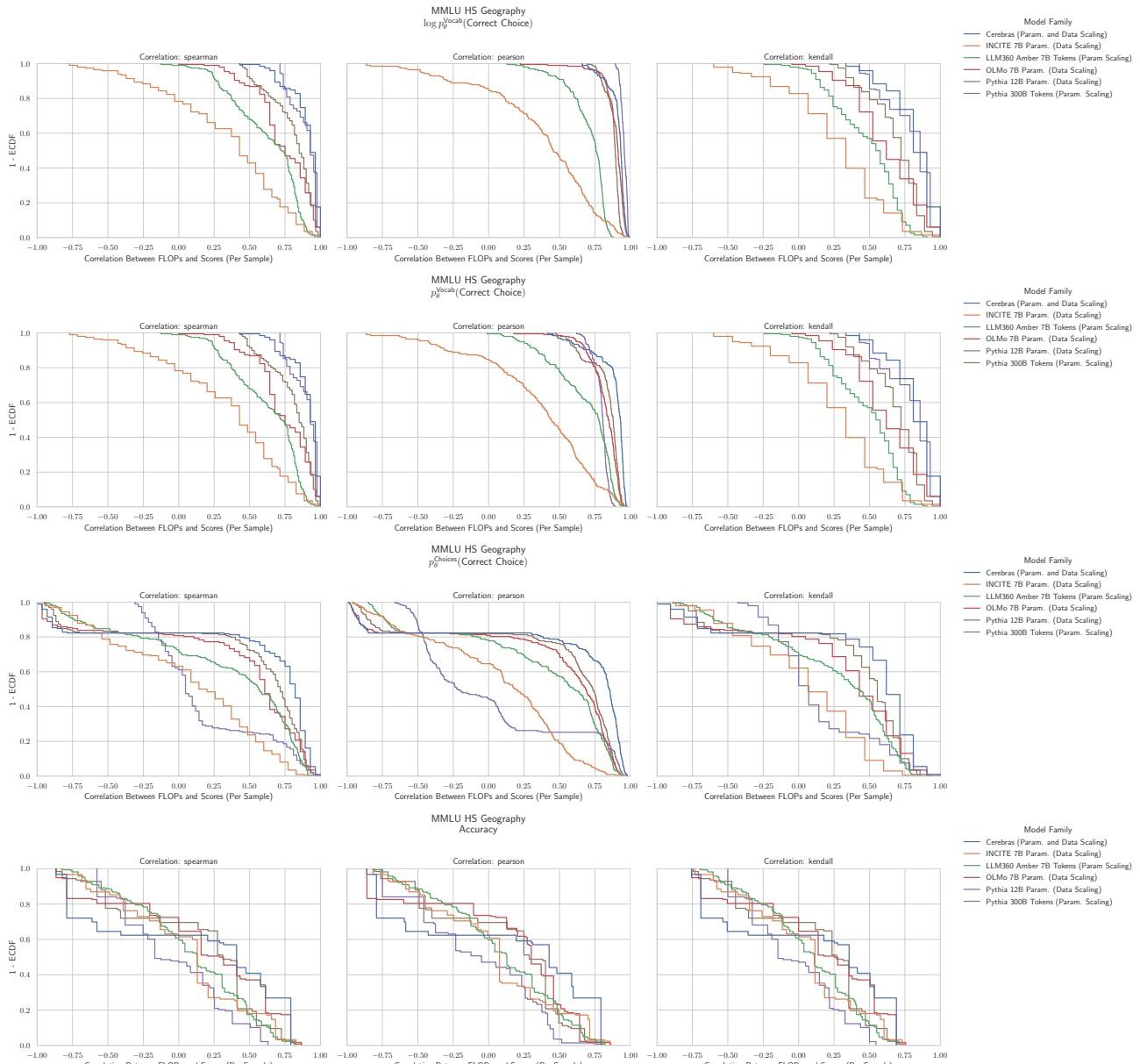

*Figure 38.* **MMLU High School Geography: Downstream performance is computed via a sequence of transformations that deteriorate correlations between scores and pretraining compute.**

## H.30. NLP Benchmark: MMLU High School Government & Politics ([Hendrycks et al., 2020](#))

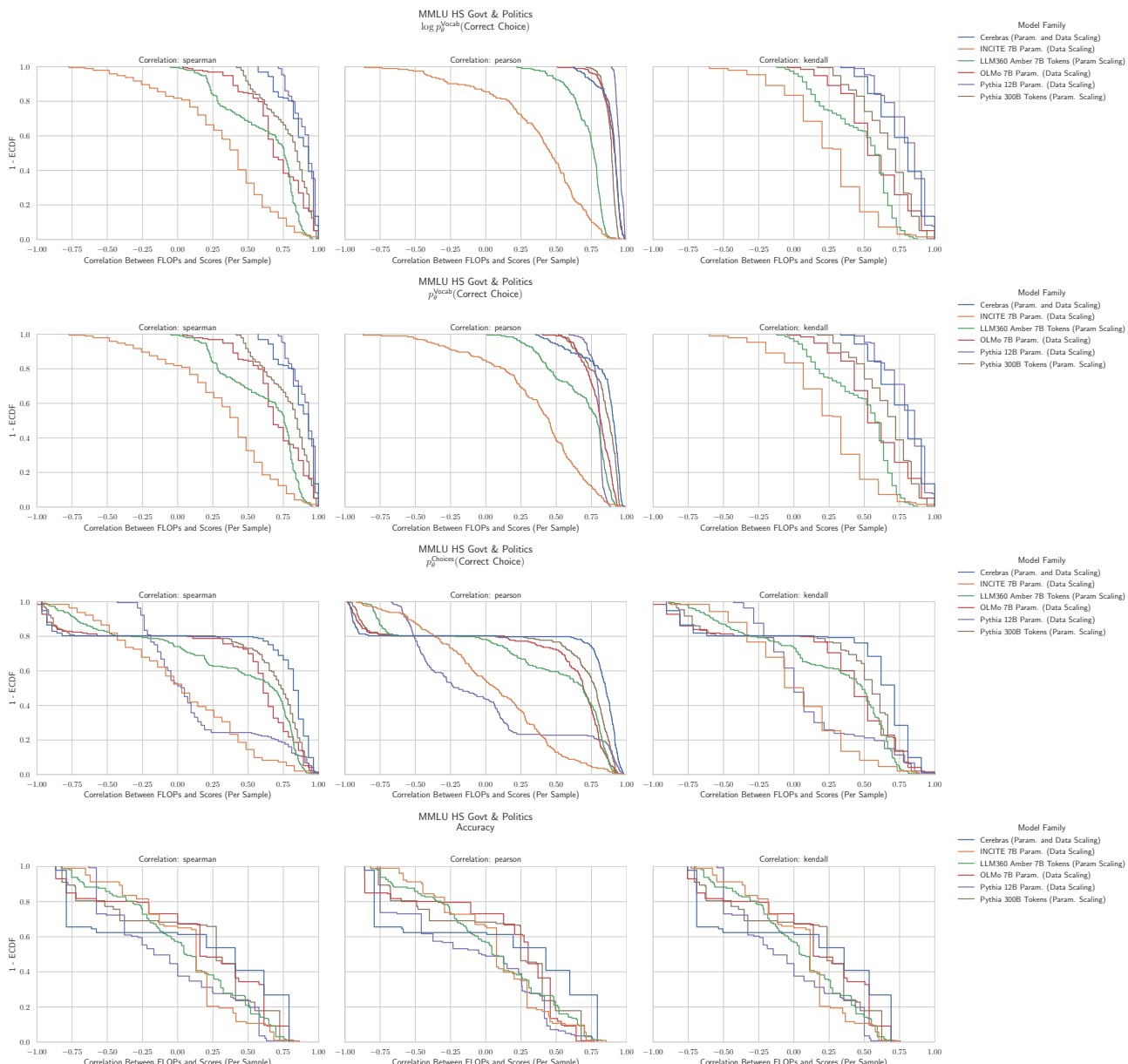

*Figure 39.* **MMLU High School Government & Politics: Downstream performance is computed via a sequence of transformations that deteriorate correlations between scores and pretraining compute.**

## H.31. NLP Benchmark: MMLU High School Macroeconomics (Hendrycks et al., 2020)

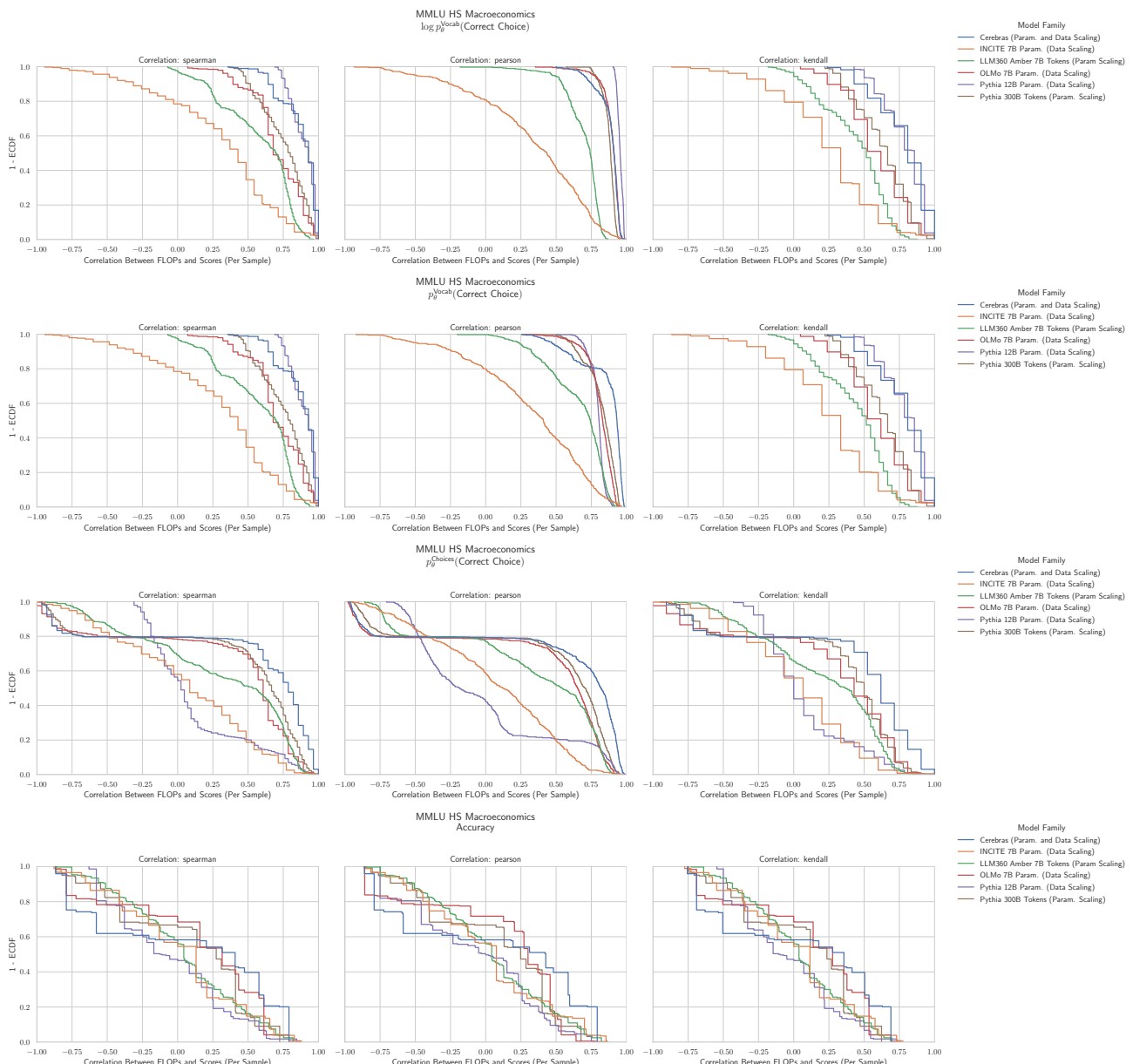

*Figure 40.* **MMLU High School Macroeconomics: Downstream performance is computed via a sequence of transformations that deteriorate correlations between scores and pretraining compute.**

## H.32. NLP Benchmark: MMLU High School Mathematics ([Hendrycks et al., 2020](#))

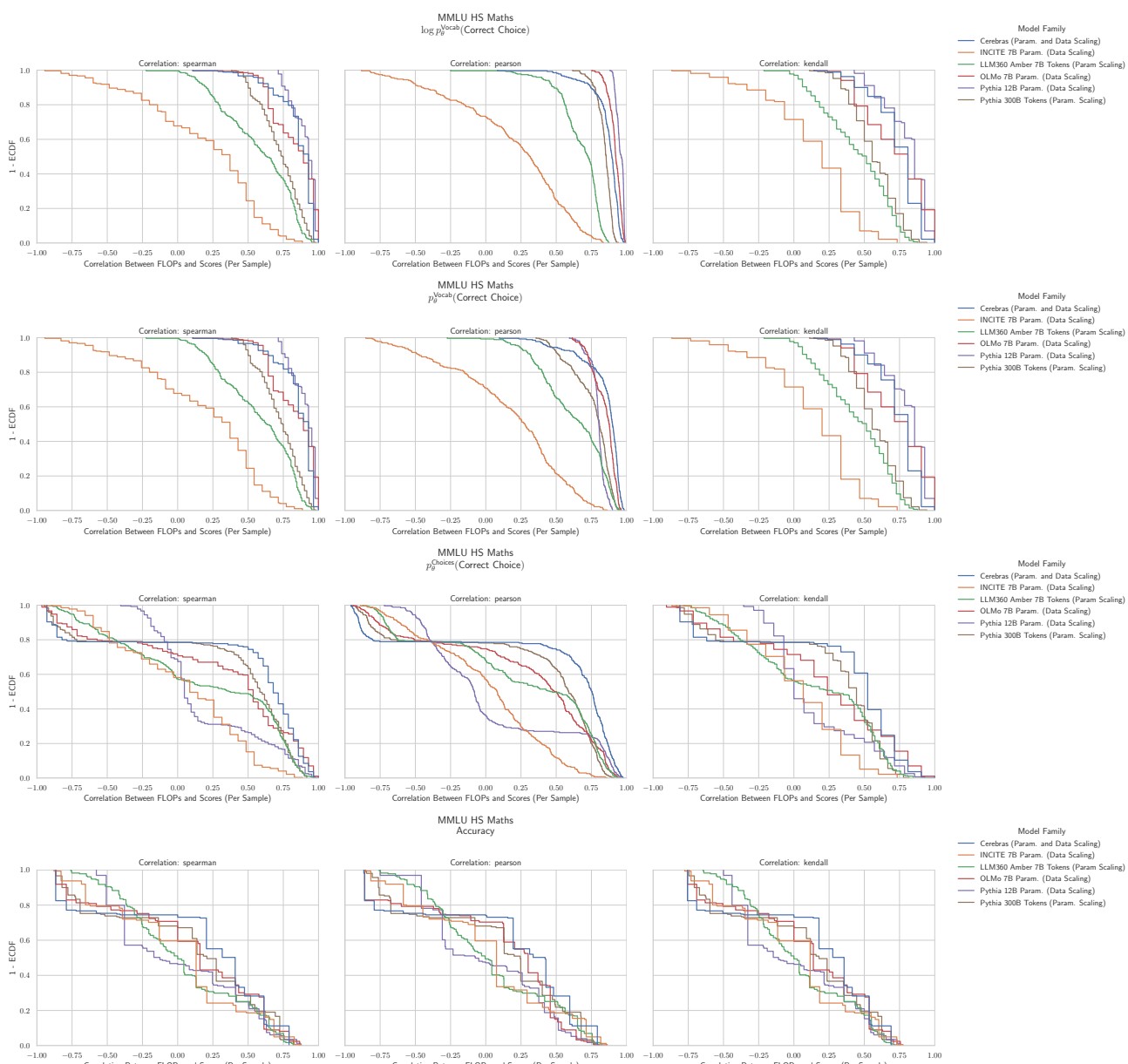

*Figure 41.* **MMLU High School Mathematics: Downstream performance is computed via a sequence of transformations that deteriorate correlations between scores and pretraining compute.**

## H.33. NLP Benchmark: MMLU High School Microeconomics (Hendrycks et al., 2020)

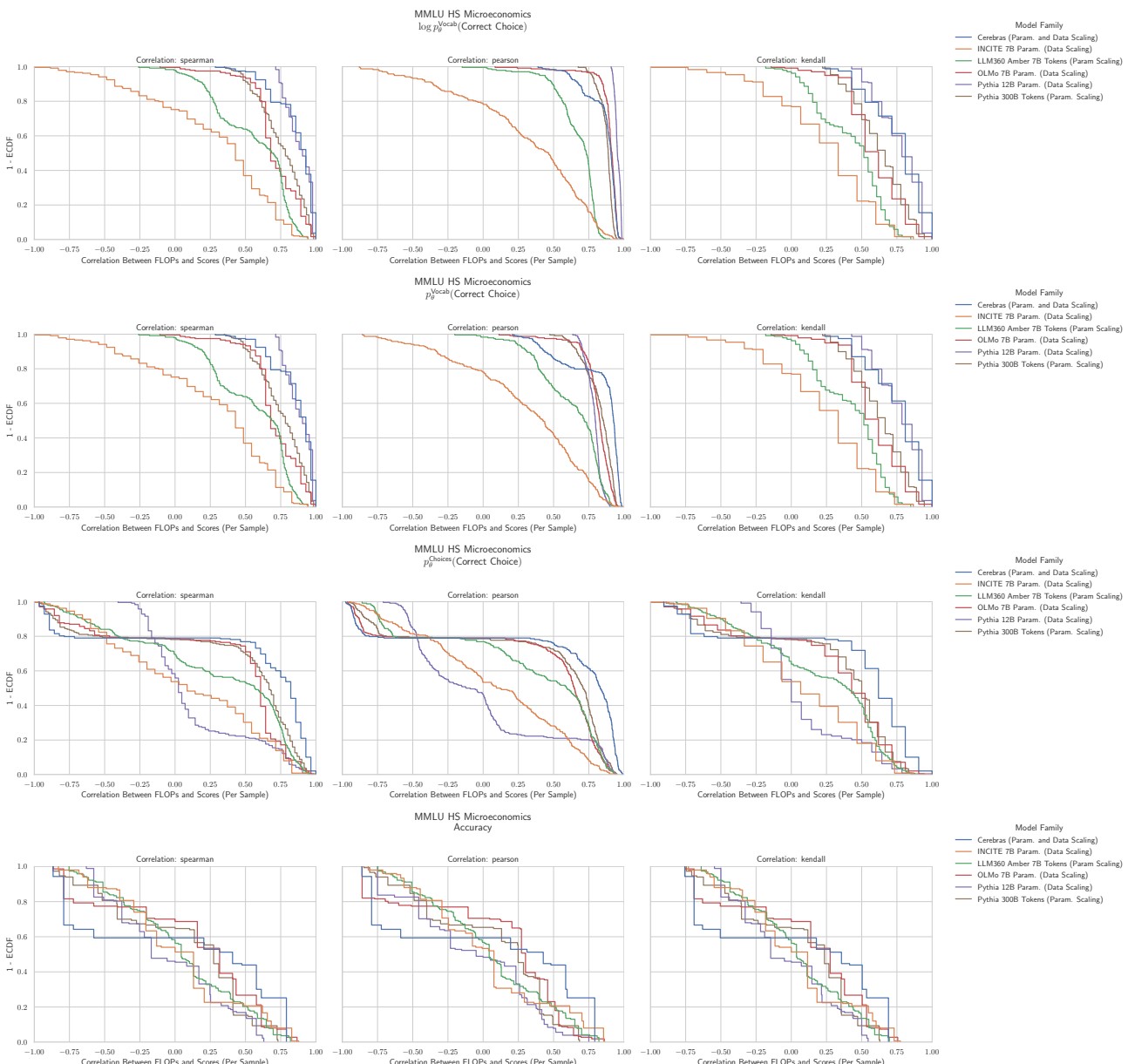

*Figure 42.* **MMLU High School Microeconomics: Downstream performance is computed via a sequence of transformations that deteriorate correlations between scores and pretraining compute.**

## H.34. NLP Benchmark: MMLU High School Physics (Hendrycks et al., 2020)

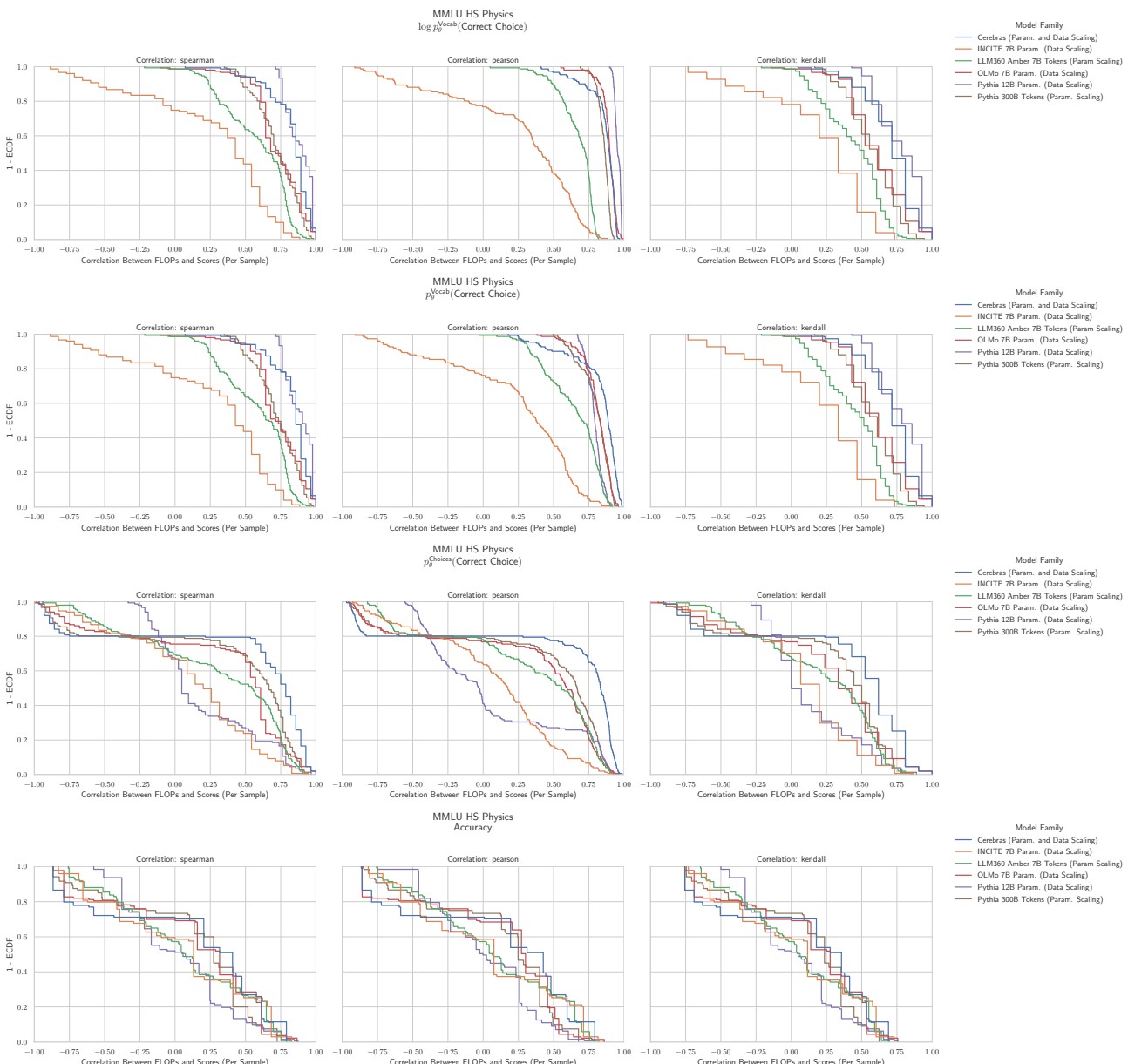

*Figure 43.* **MMLU High School Physics: Downstream performance is computed via a sequence of transformations that deteriorate correlations between scores and pretraining compute.**

## H.35. NLP Benchmark: MMLU High School Psychology (Hendrycks et al., 2020)

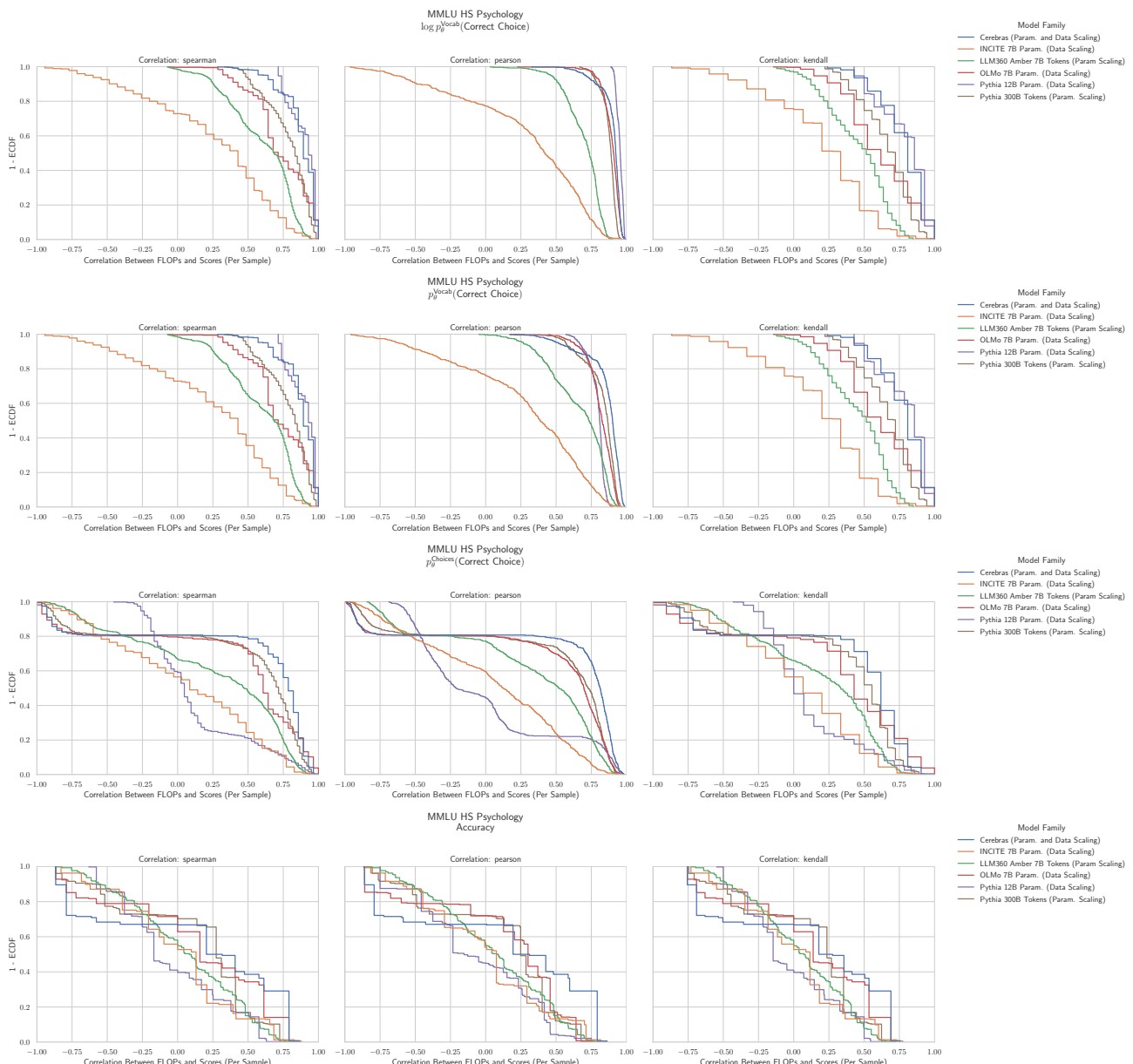

*Figure 44.* **MMLU High School Psychology: Downstream performance is computed via a sequence of transformations that deteriorate correlations between scores and pretraining compute.**

## H.36. NLP Benchmark: MMLU High School Statistics (Hendrycks et al., 2020)

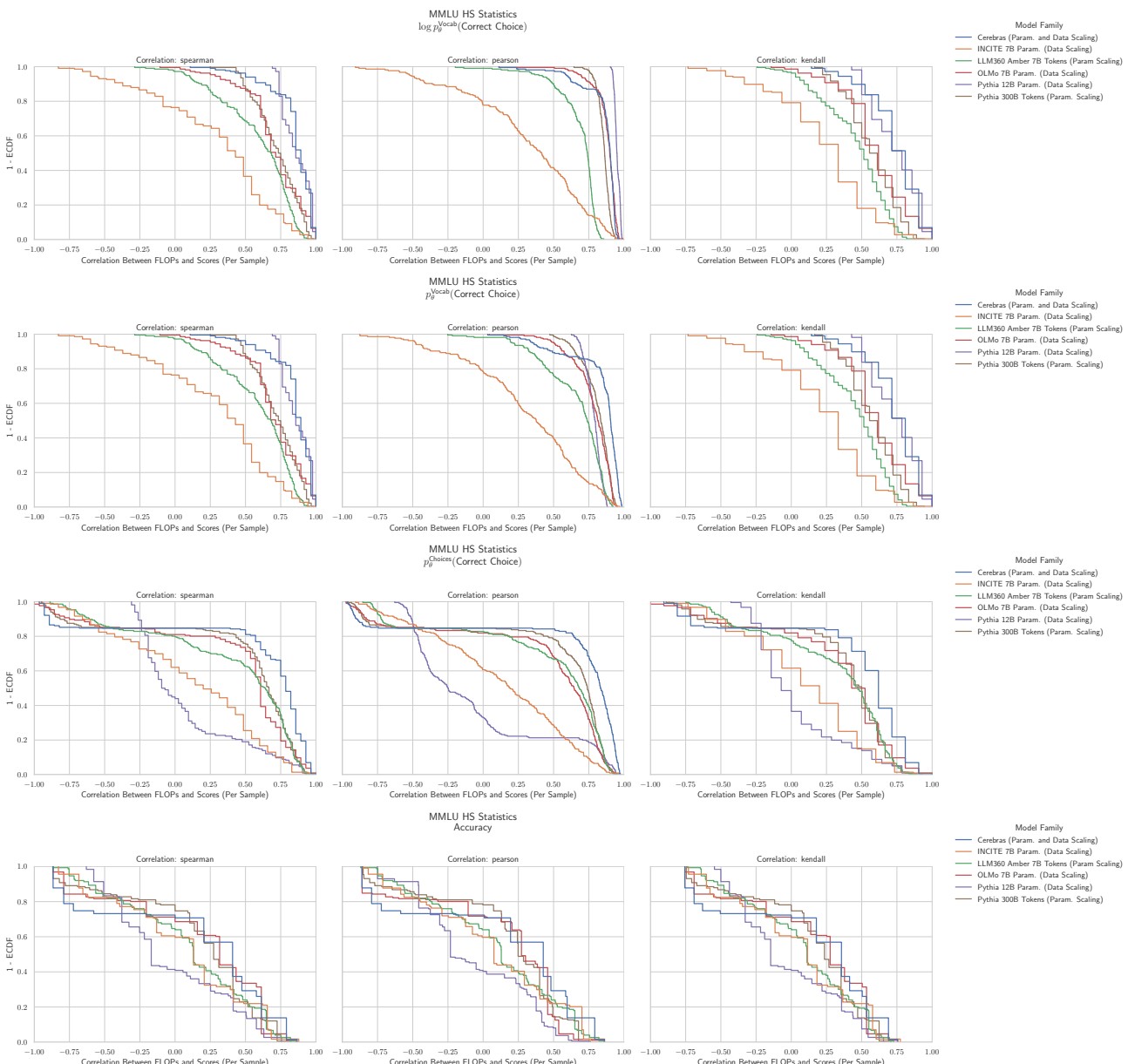

*Figure 45.* **MMLU High School Statistics: Downstream performance is computed via a sequence of transformations that deteriorate correlations between scores and pretraining compute.**

## H.37. NLP Benchmark: MMLU High School US History (Hendrycks et al., 2020)

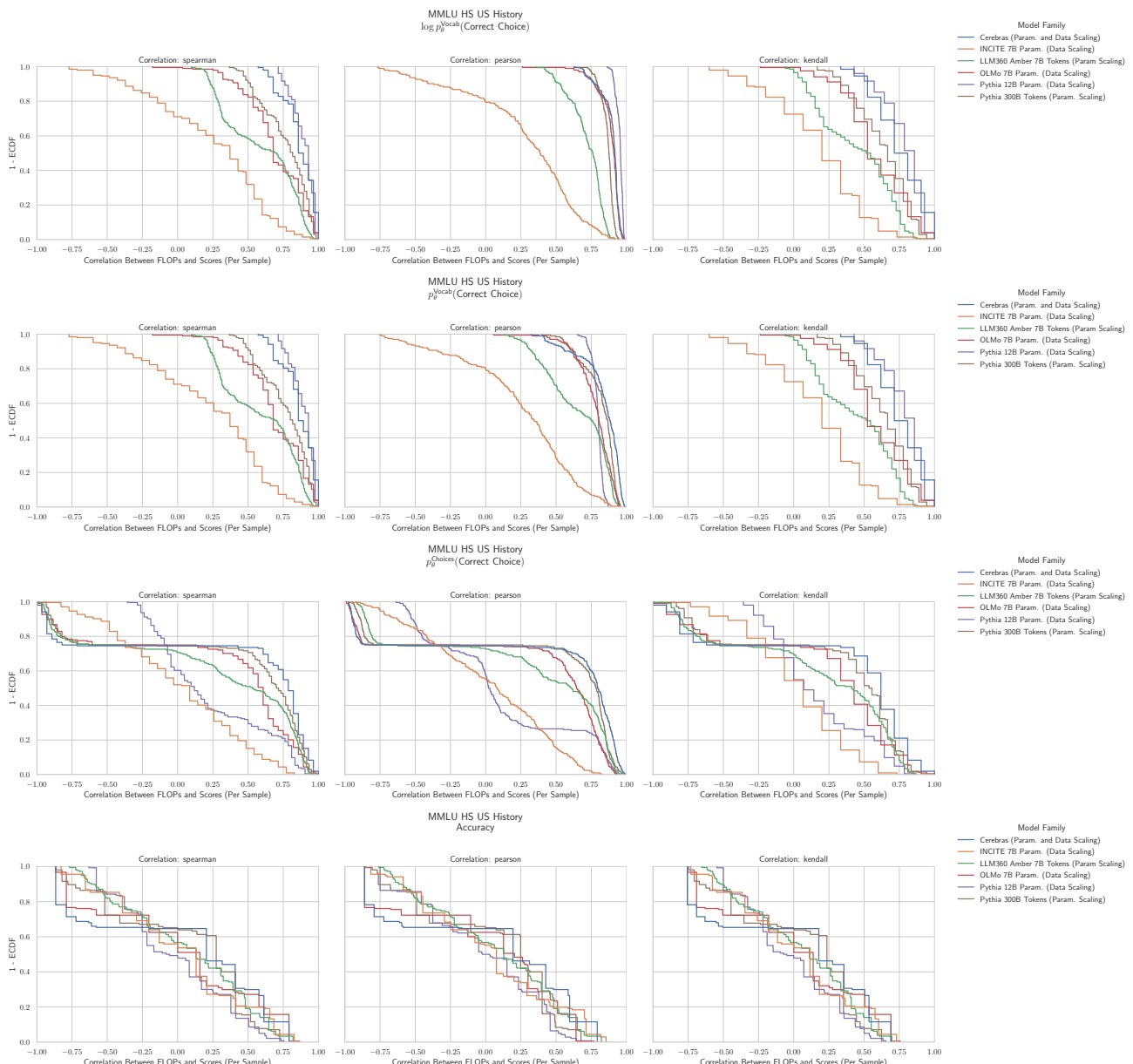

*Figure 46.* **MMLU High School US History: Downstream performance is computed via a sequence of transformations that deteriorate correlations between scores and pretraining compute.**

## H.38. NLP Benchmark: MMLU High School World History (Hendrycks et al., 2020)

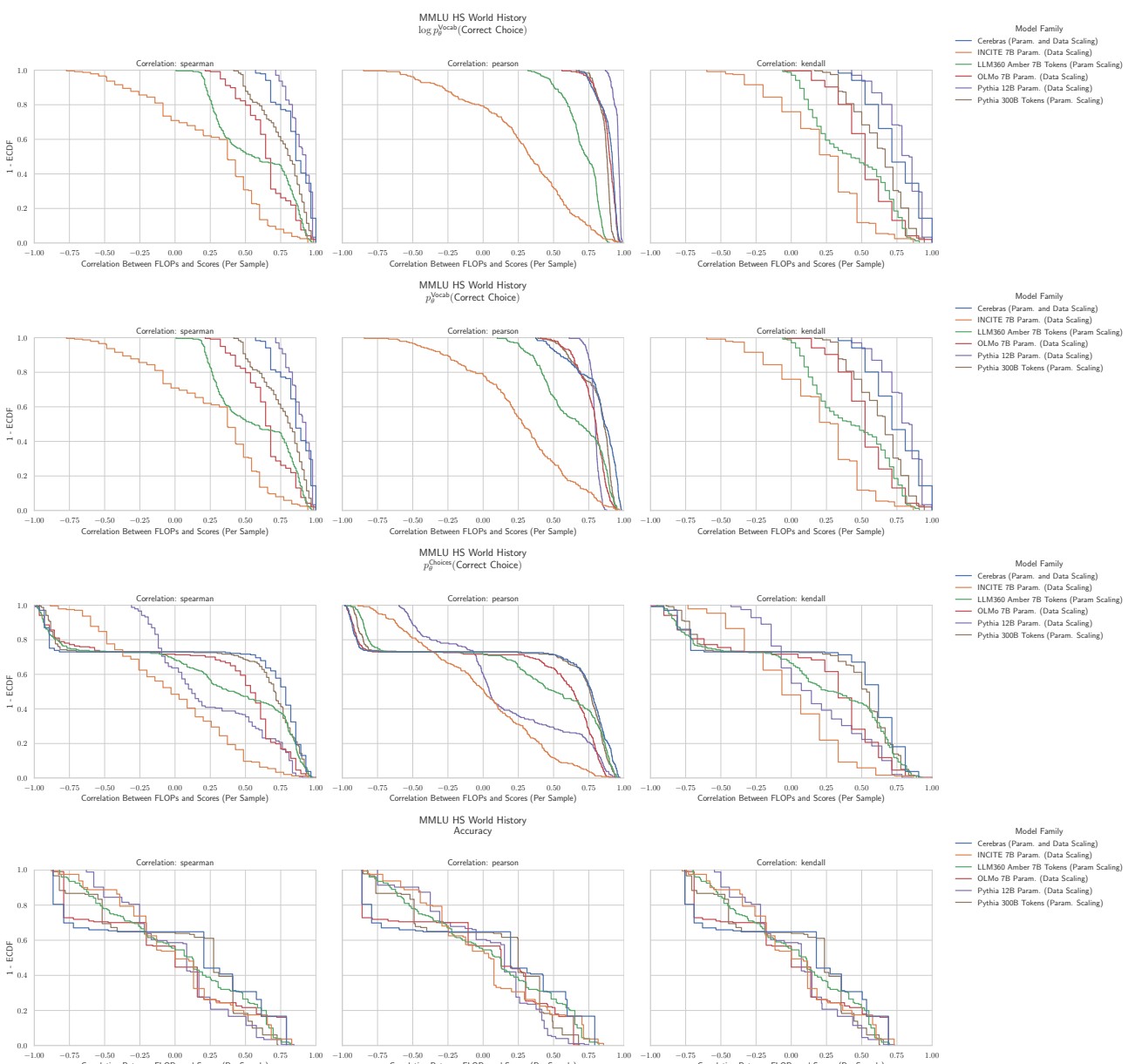

*Figure 47.* **MMLU High School World History: Downstream performance is computed via a sequence of transformations that deteriorate correlations between scores and pretraining compute.**

## H.39. NLP Benchmark: MMLU Human Aging ([Hendrycks et al., 2020](#))

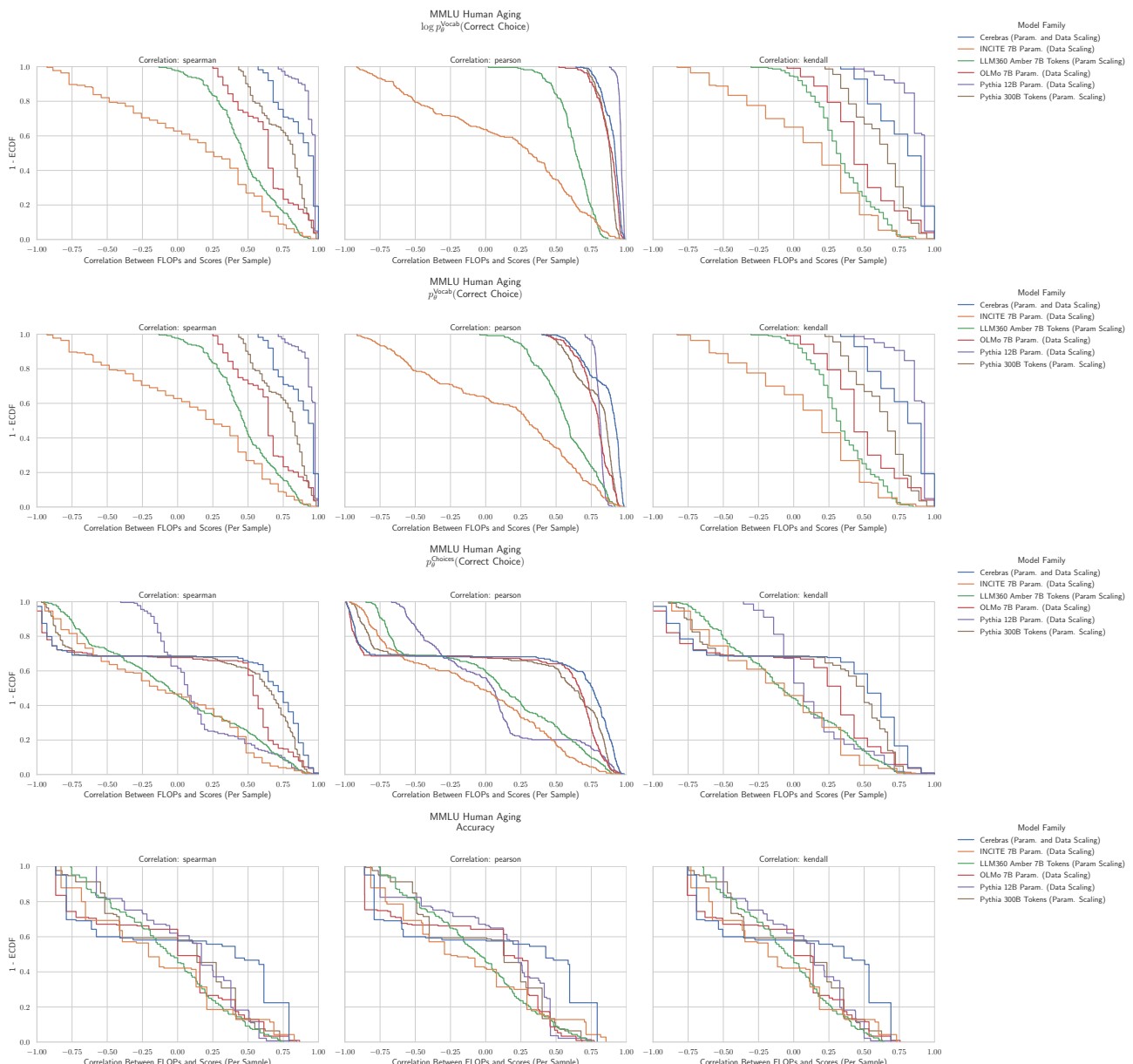

*Figure 48.* **MMLU Human Aging: Downstream performance is computed via a sequence of transformations that deteriorate correlations between scores and pretraining compute.**

## H.40. NLP Benchmark: MMLU Human Sexuality (Hendrycks et al., 2020)

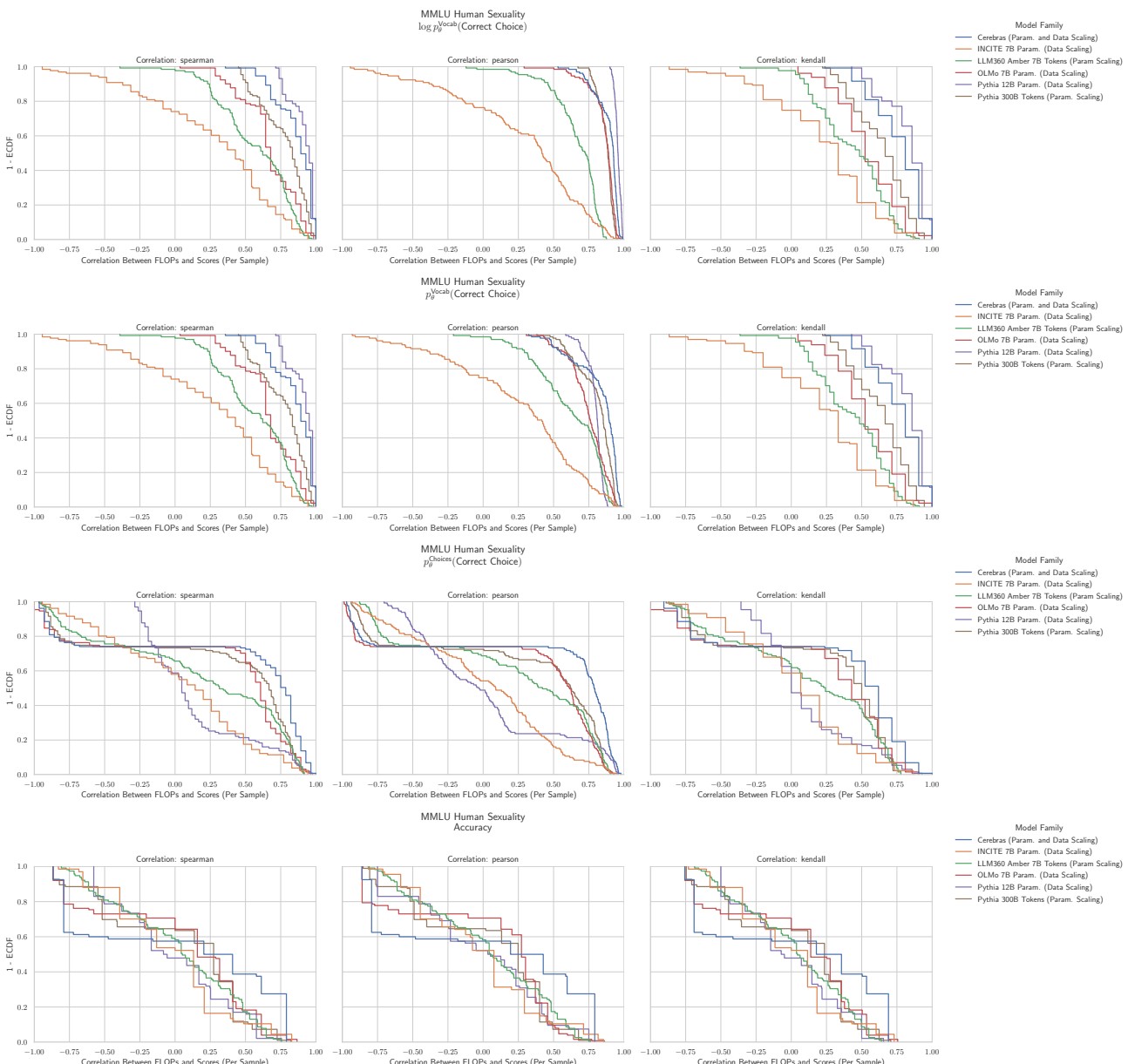

*Figure 49.* **MMLU Human Sexuality: Downstream performance is computed via a sequence of transformations that deteriorate correlations between scores and pretraining compute.**

## H.41. NLP Benchmark: MMLU International Law (Hendrycks et al., 2020)

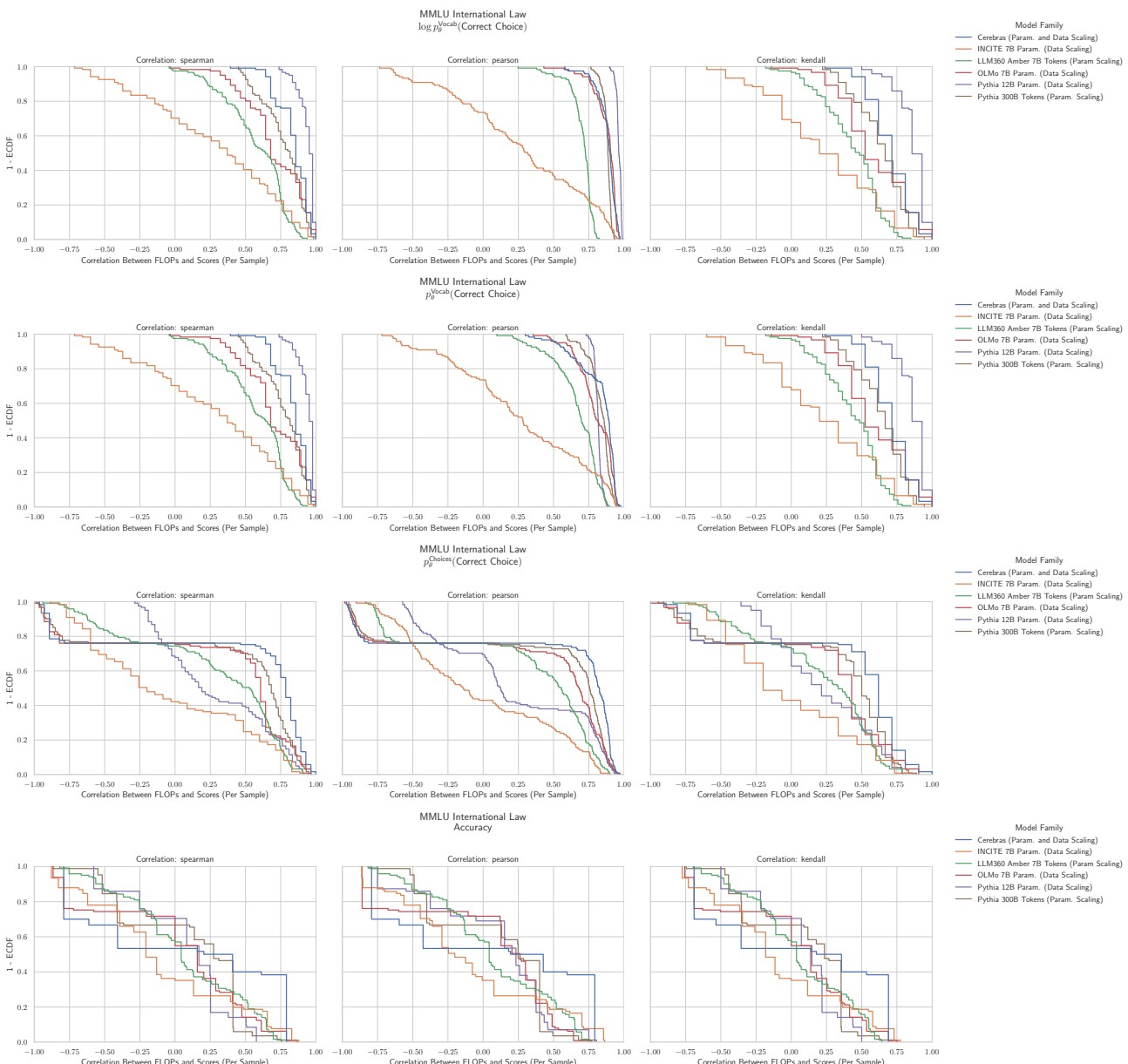

*Figure 50.* **MMLU International Law: Downstream performance is computed via a sequence of transformations that deteriorate correlations between scores and pretraining compute.**

## H.42. NLP Benchmark: MMLU Jurisprudence (Hendrycks et al., 2020)

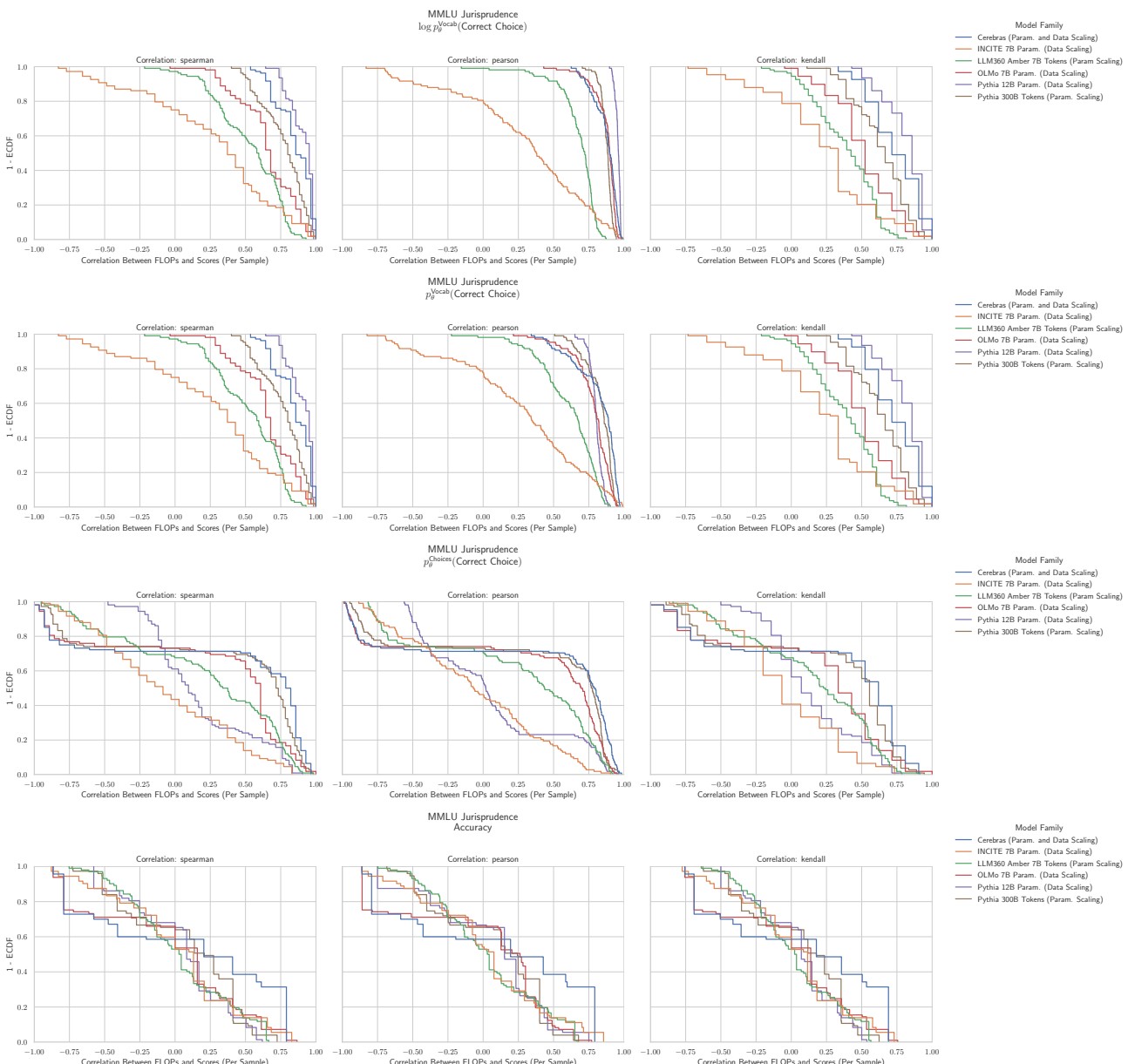

*Figure 51.* **MMLU Jurisprudence: Downstream performance is computed via a sequence of transformations that deteriorate correlations between scores and pretraining compute.**

## H.43. NLP Benchmark: MMLU Logical Fallacies (Hendrycks et al., 2020)

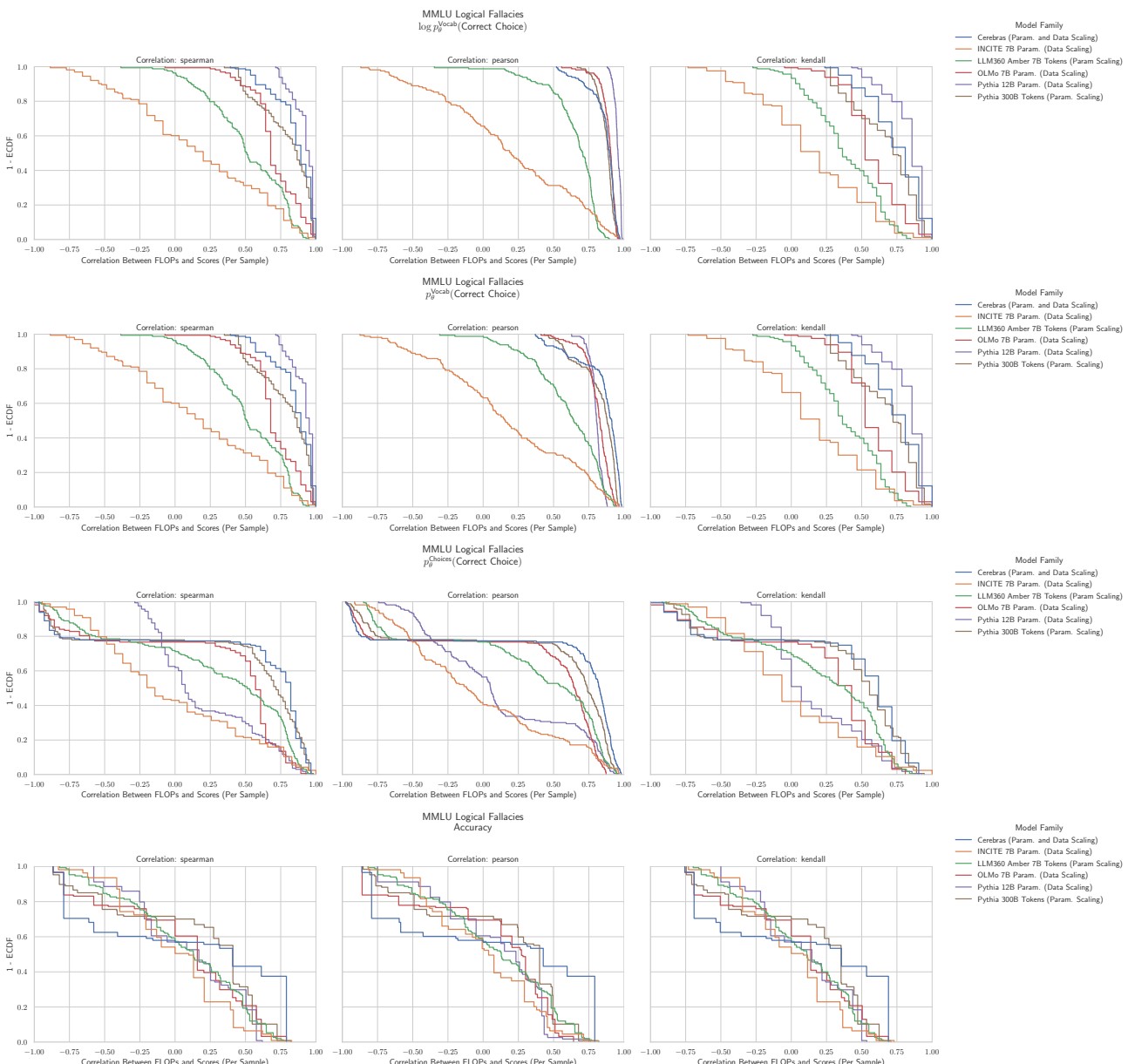

*Figure 52.* **MMLU Logical Fallacies: Downstream performance is computed via a sequence of transformations that deteriorate correlations between scores and pretraining compute.**

## H.44. NLP Benchmark: MMLU Machine Learning (Hendrycks et al., 2020)

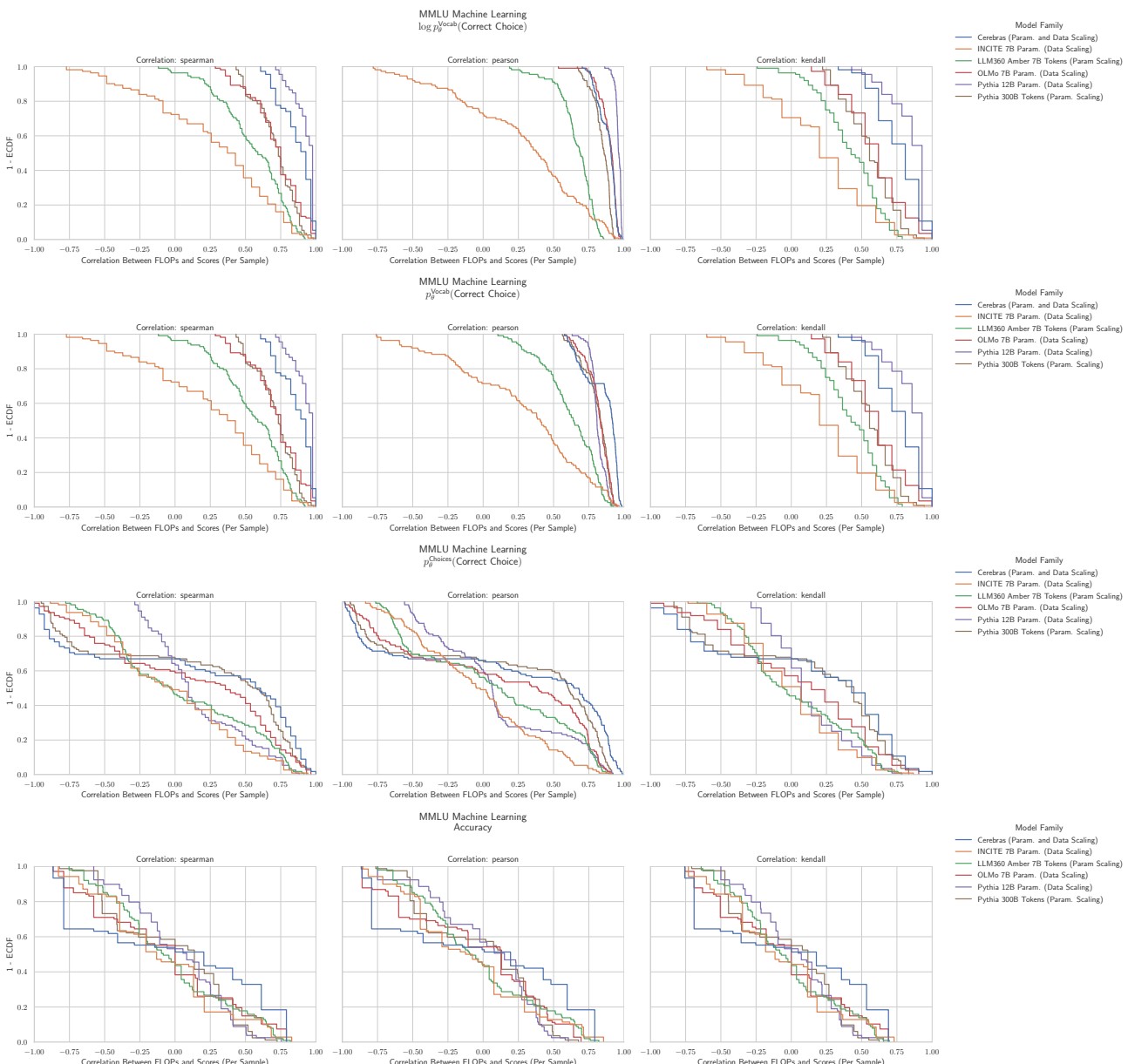

*Figure 53.* **MMLU Machine Learning: Downstream performance is computed via a sequence of transformations that deteriorate correlations between scores and pretraining compute.**

## H.45. NLP Benchmark: MMLU Management (Hendrycks et al., 2020)

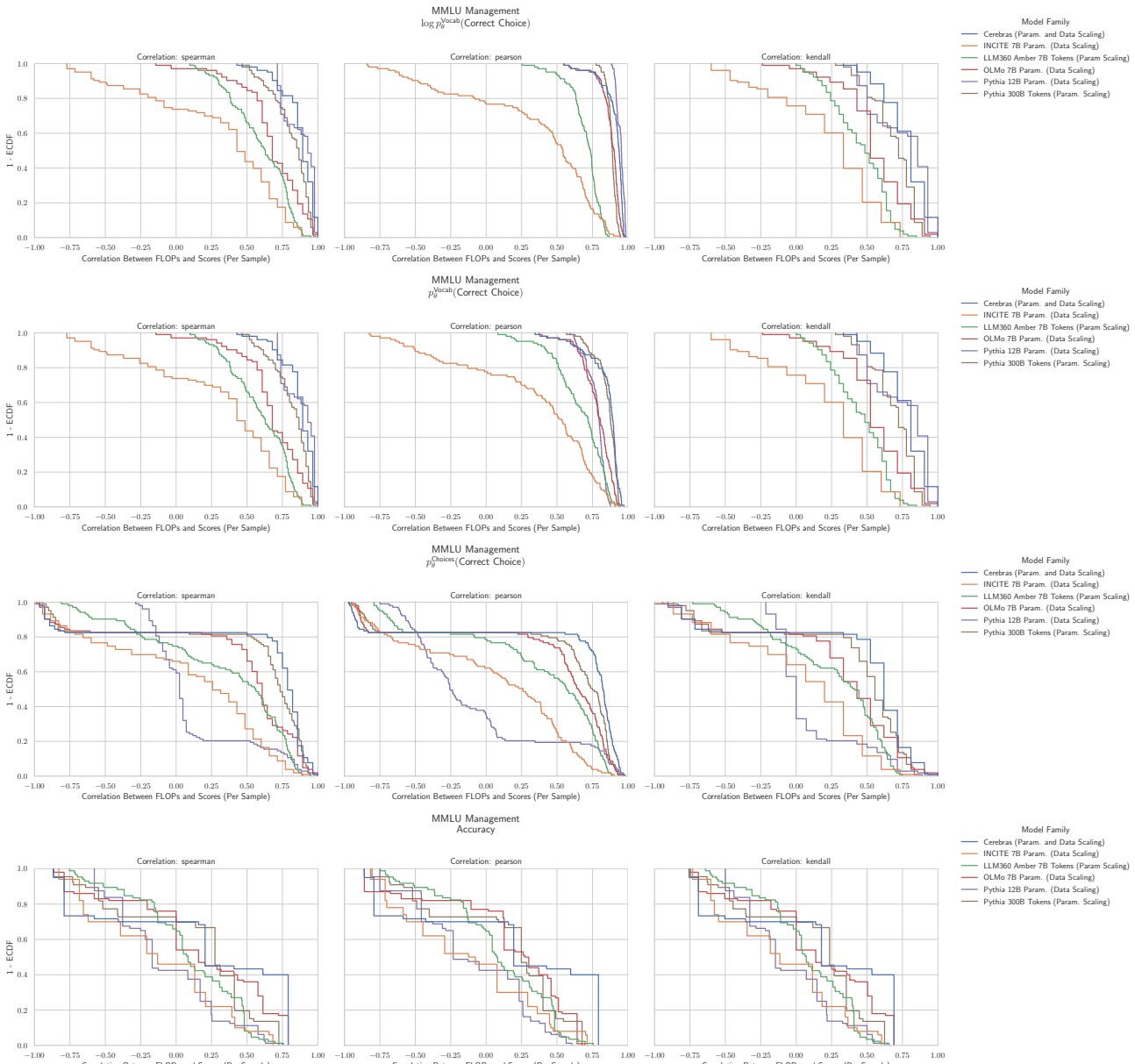

*Figure 54.* **MMLU Management: Downstream performance is computed via a sequence of transformations that deteriorate correlations between scores and pretraining compute.**

## H.46. NLP Benchmark: MMLU Marketing ([Hendrycks et al., 2020](#))

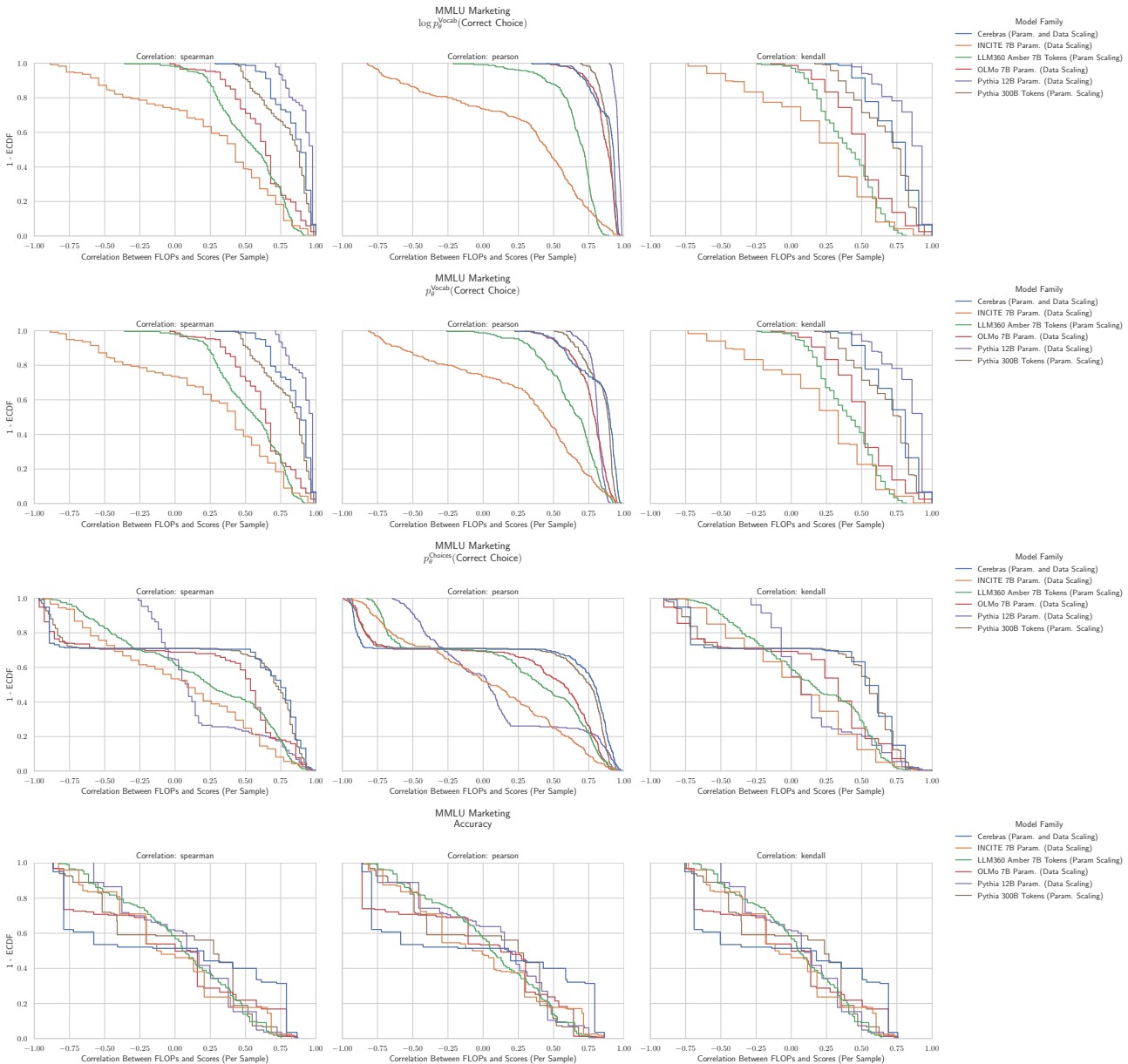

*Figure 55.* **MMLU Marketing: Downstream performance is computed via a sequence of transformations that deteriorate correlations between scores and pretraining compute.**

## H.47. NLP Benchmark: MMLU Medical Genetics (Hendrycks et al., 2020)

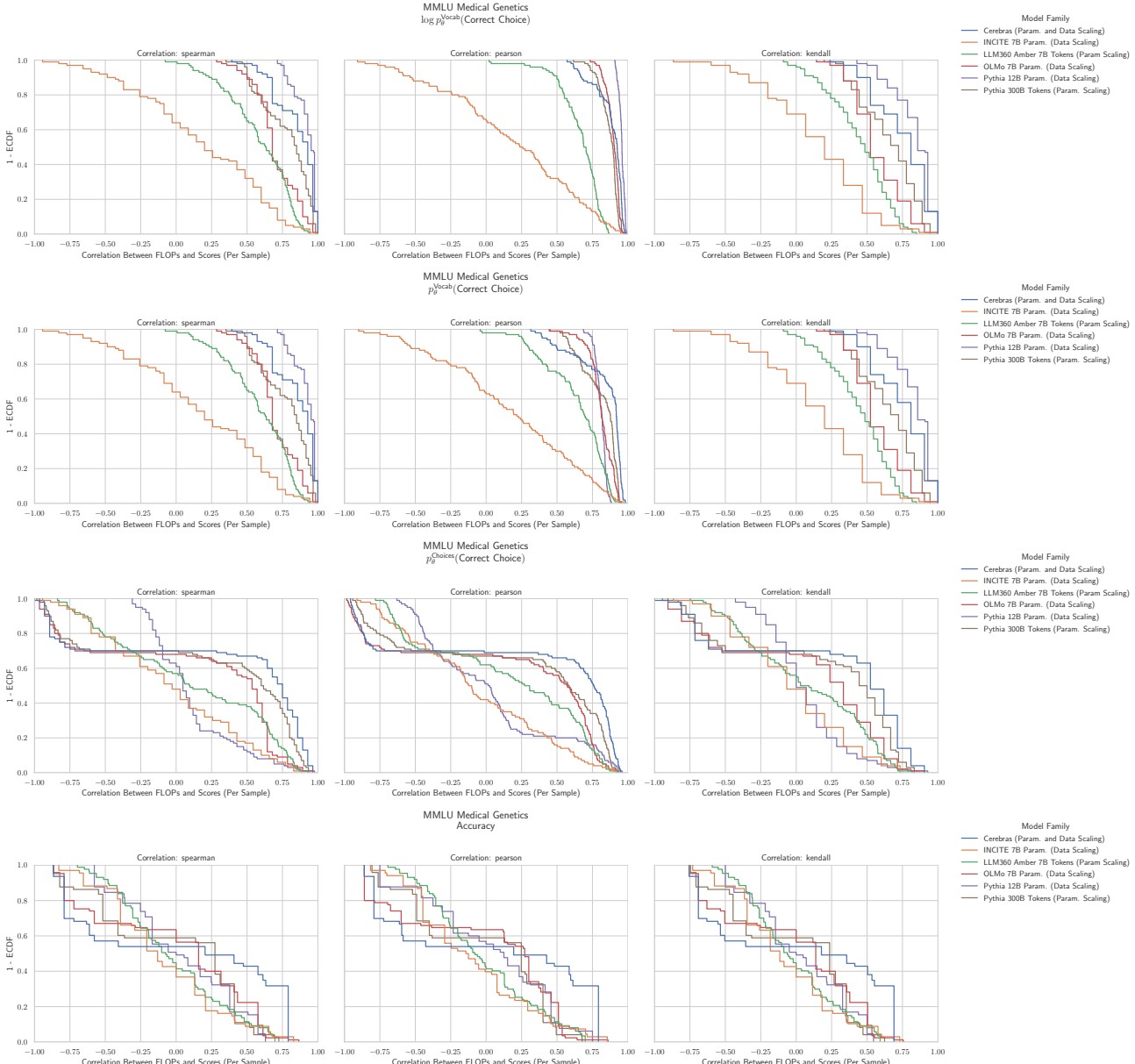

*Figure 56.* **MMLU Medical Genetics: Downstream performance is computed via a sequence of transformations that deteriorate correlations between scores and pretraining compute.**

## H.48. NLP Benchmark: MMLU Miscellaneous (Hendrycks et al., 2020)

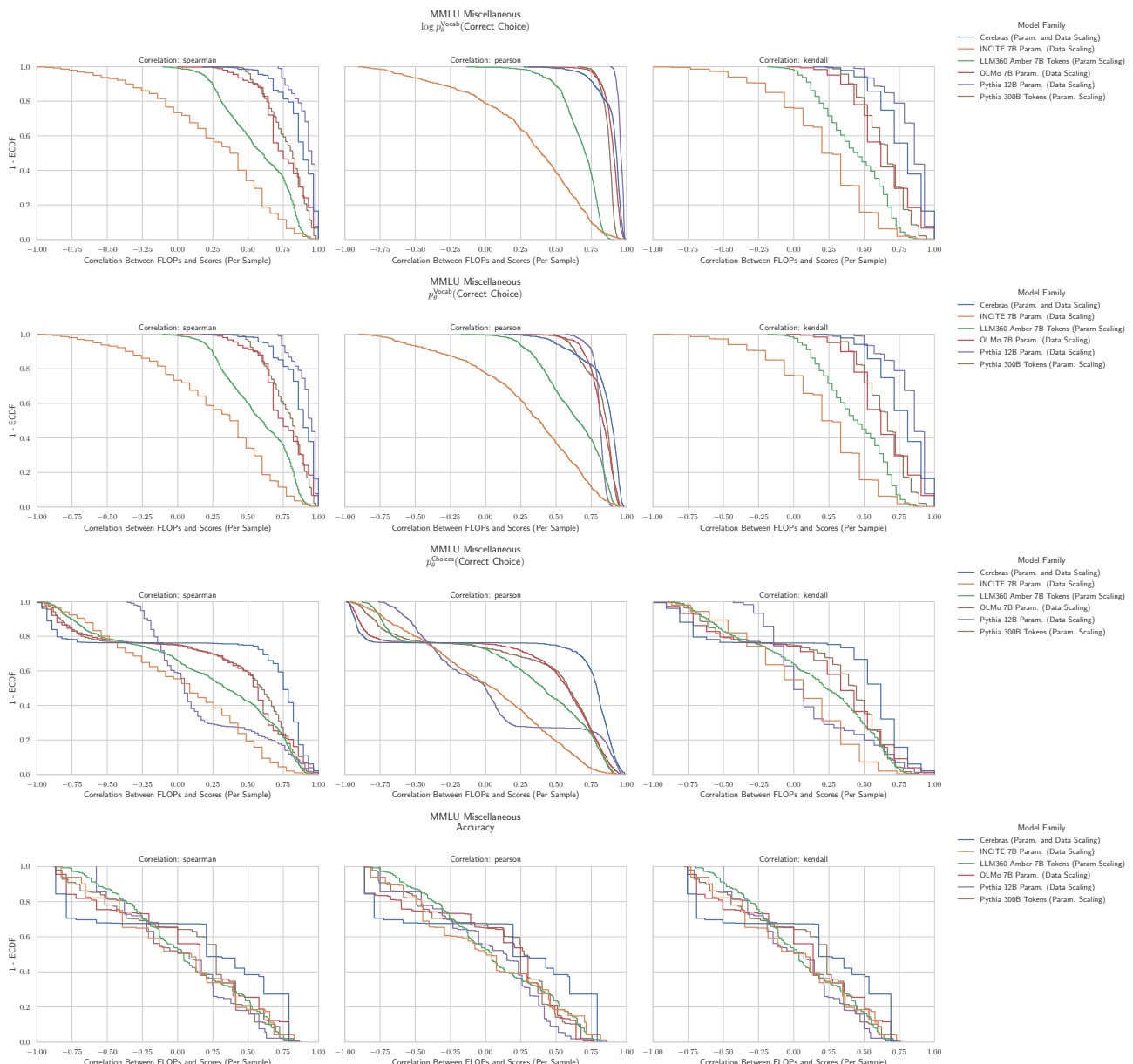

*Figure 57.* **MMLU Miscellaneous: Downstream performance is computed via a sequence of transformations that deteriorate correlations between scores and pretraining compute.**

## H.49. NLP Benchmark: MMLU Moral Disputes (Hendrycks et al., 2020)

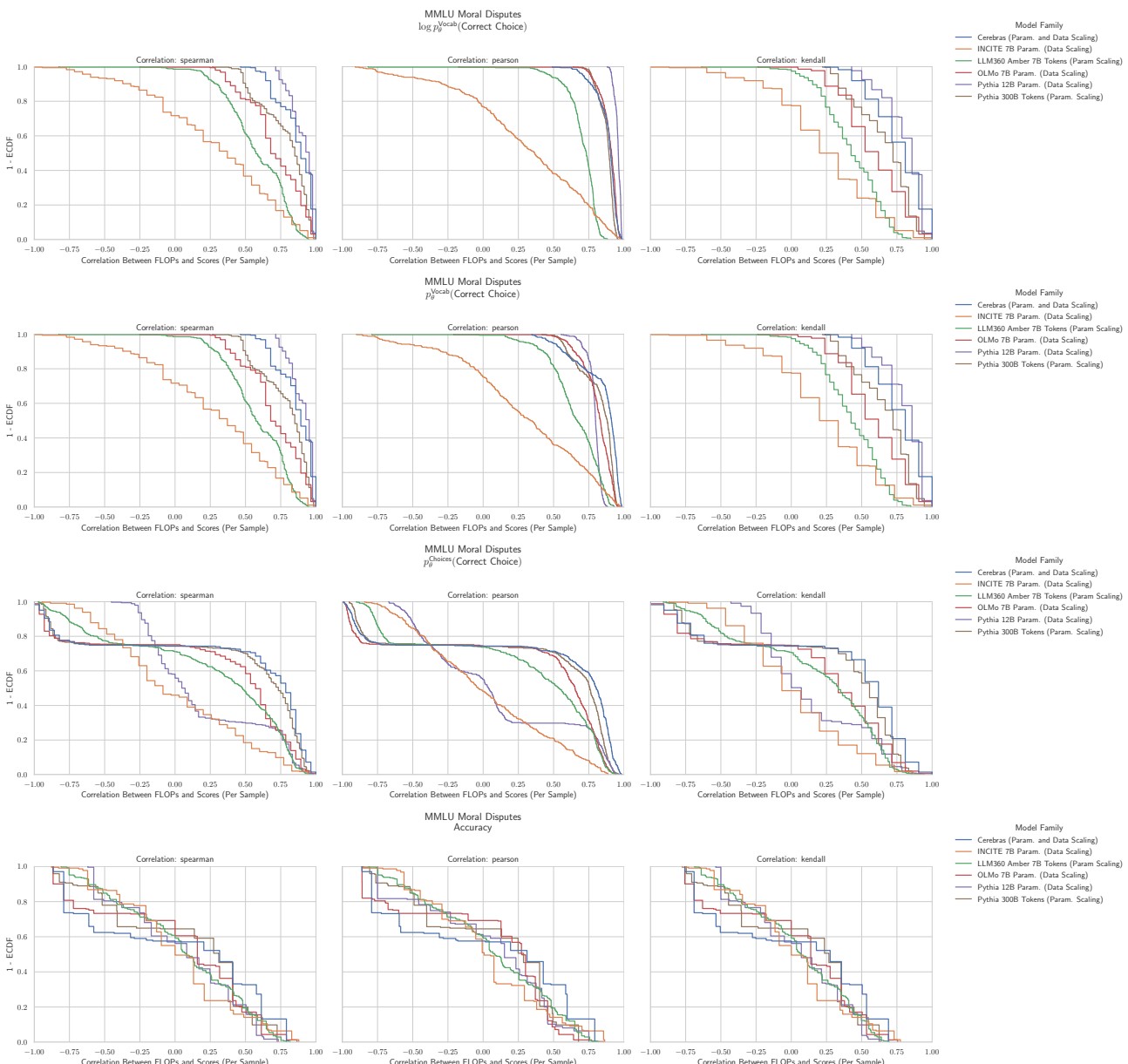

*Figure 58.* **MMLU Moral Disputes: Downstream performance is computed via a sequence of transformations that deteriorate correlations between scores and pretraining compute.**

## H.50. NLP Benchmark: MMLU Moral Scenarios (Hendrycks et al., 2020)

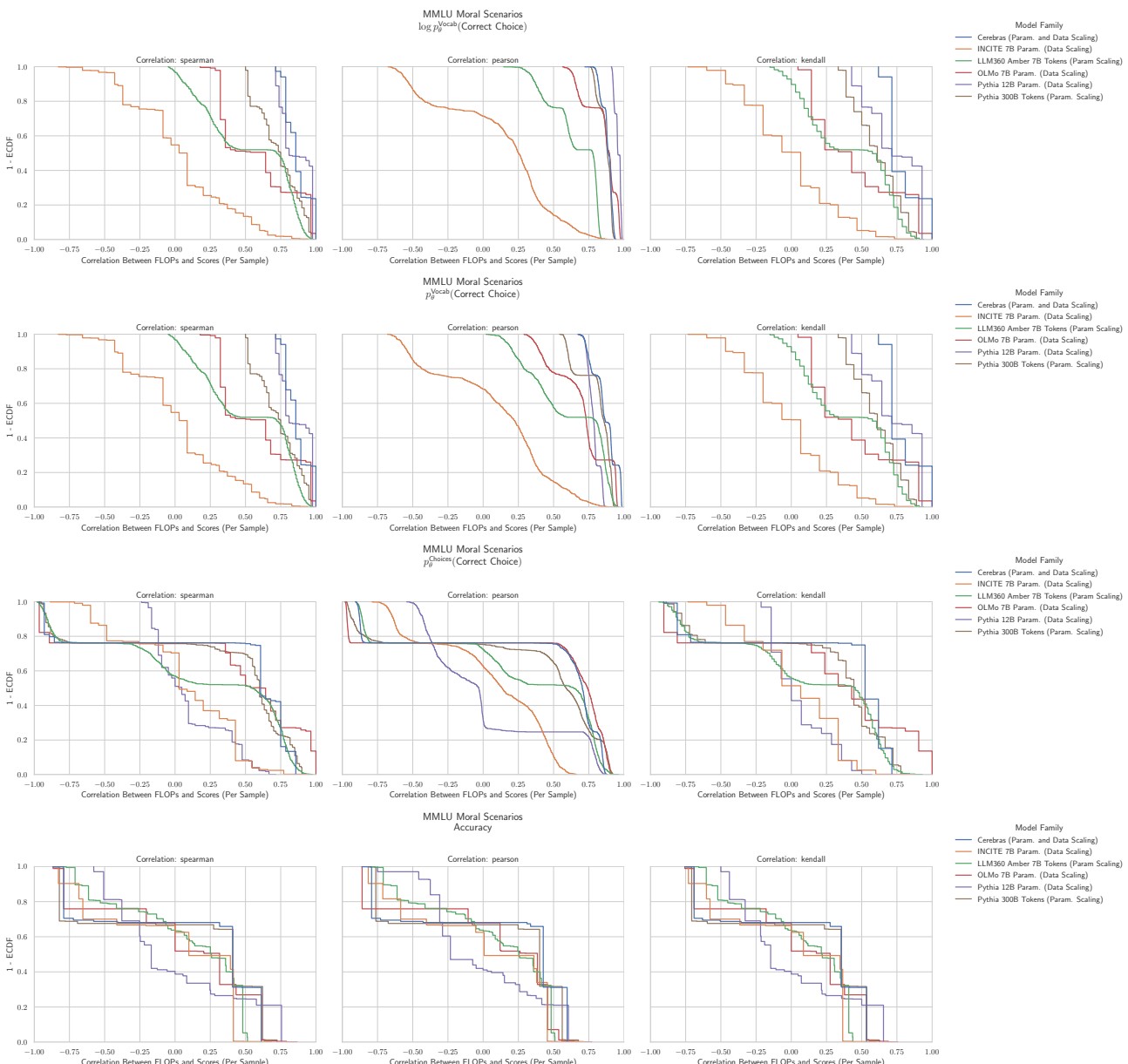

*Figure 59.* **MMLU Moral Scenarios: Downstream performance is computed via a sequence of transformations that deteriorate correlations between scores and pretraining compute.**

## H.51. NLP Benchmark: MMLU Nutrition (Hendrycks et al., 2020)

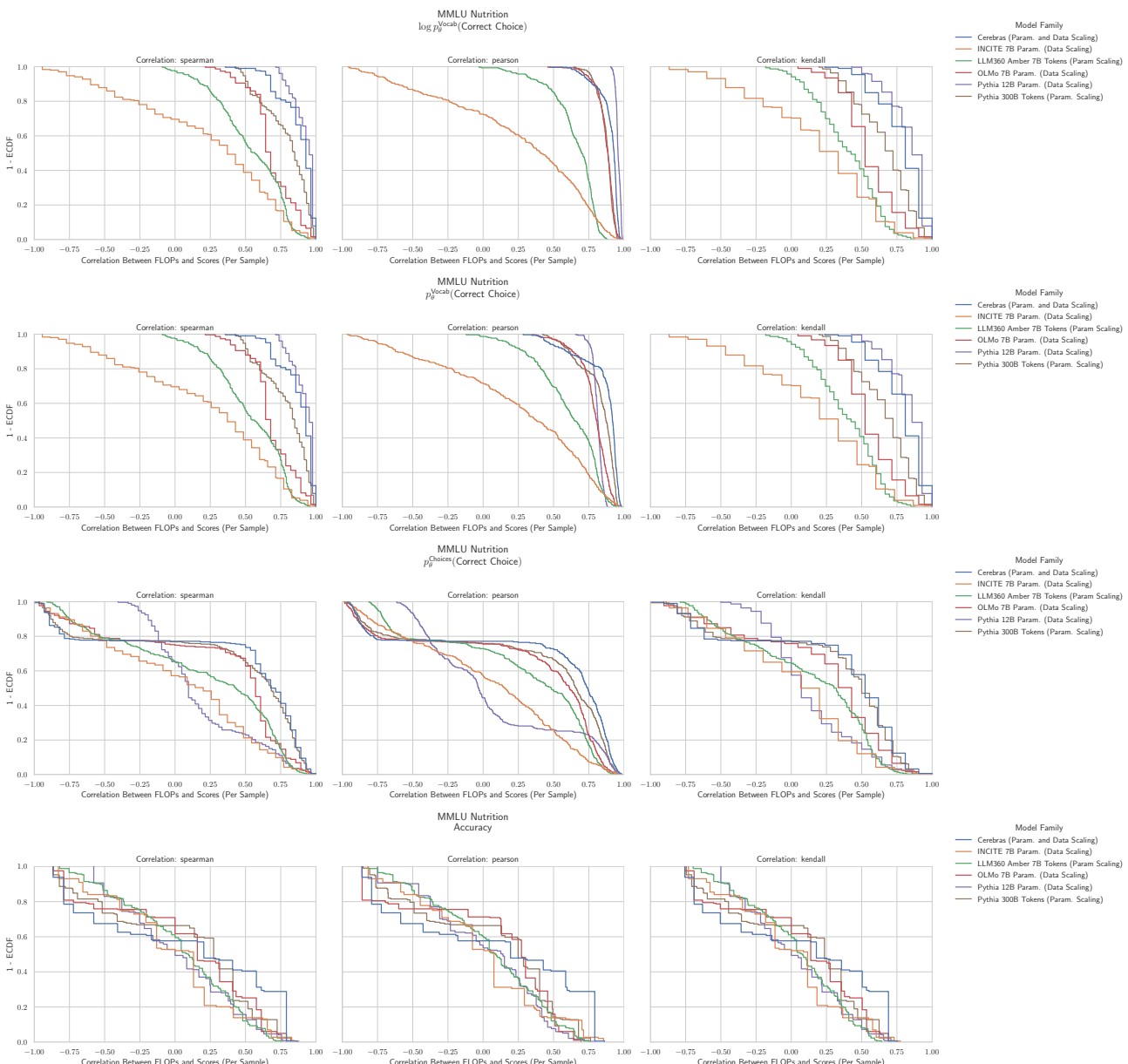

*Figure 60.* **MMLU Nutrition: Downstream performance is computed via a sequence of transformations that deteriorate correlations between scores and pretraining compute.**

## H.52. NLP Benchmark: MMLU Philosophy ([Hendrycks et al., 2020](#))

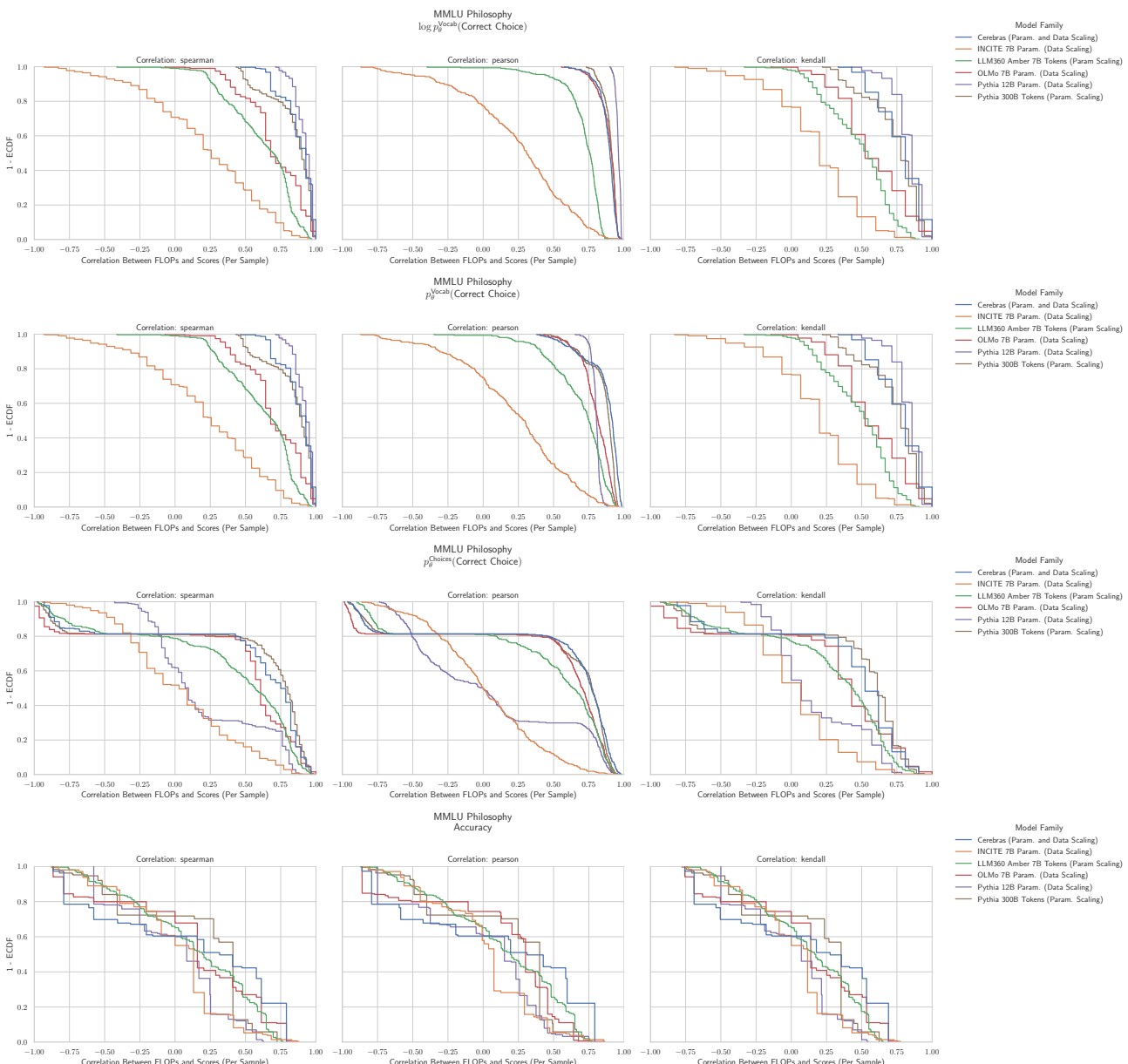

*Figure 61.* **MMLU Philosophy: Downstream performance is computed via a sequence of transformations that deteriorate correlations between scores and pretraining compute.**

## H.53. NLP Benchmark: MMLU Prehistory ([Hendrycks et al., 2020](#))

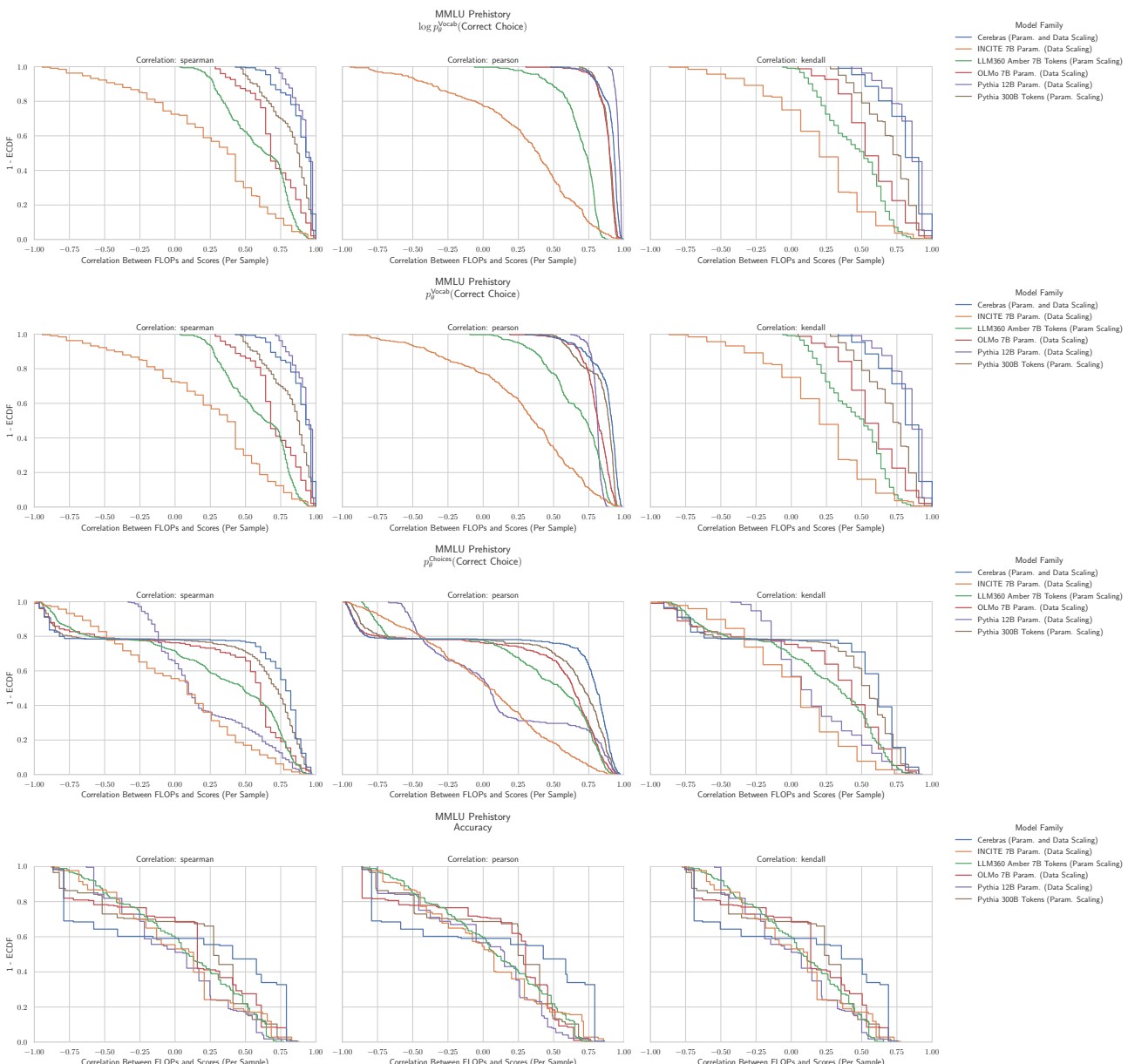

*Figure 62.* **MMLU Prehistory: Downstream performance is computed via a sequence of transformations that deteriorate correlations between scores and pretraining compute.**

## H.54. NLP Benchmark: MMLU Professional Accounting (Hendrycks et al., 2020)

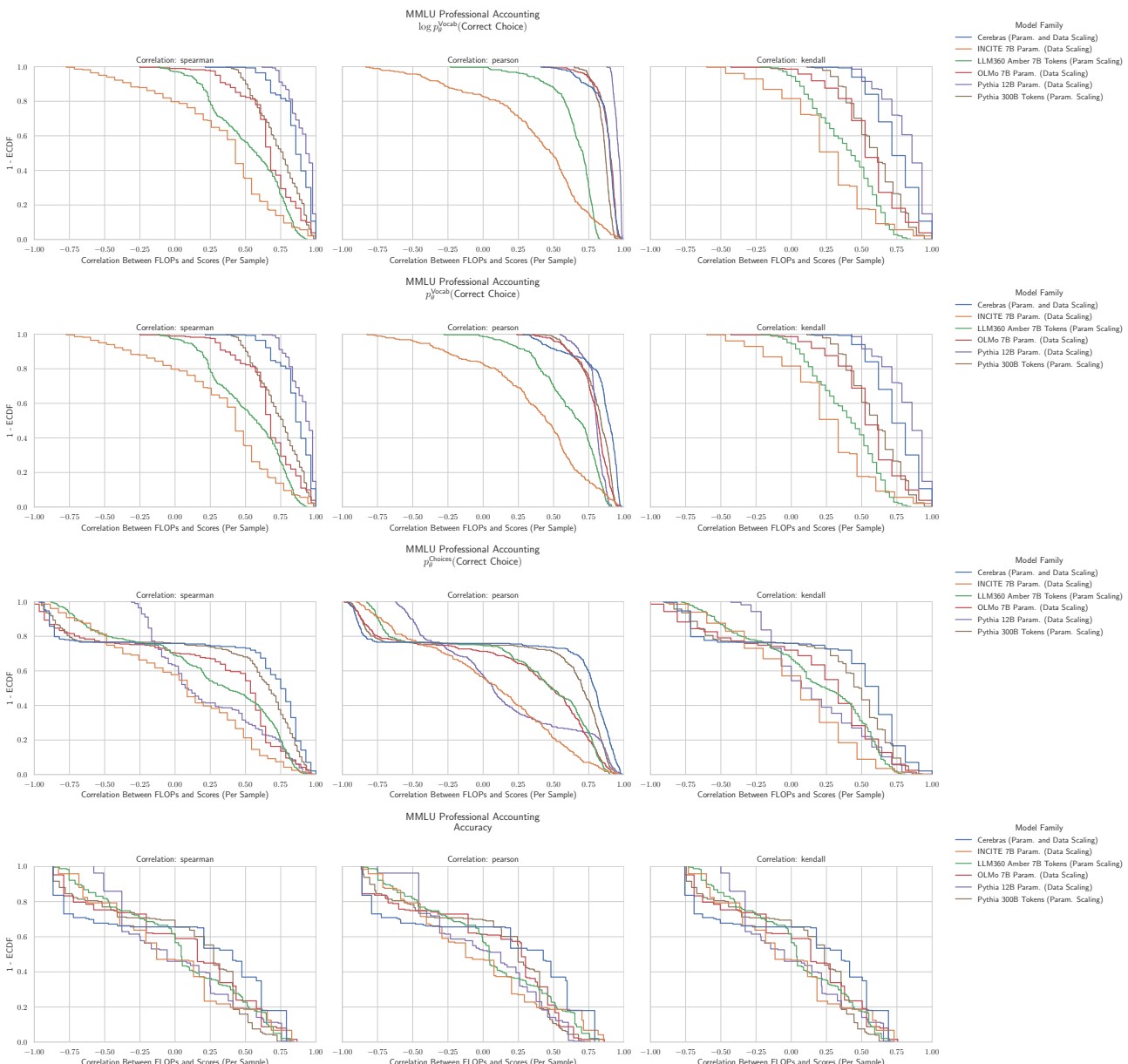

*Figure 63.* **MMLU Professional Accounting: Downstream performance is computed via a sequence of transformations that deteriorate correlations between scores and pretraining compute.**

## H.55. NLP Benchmark: MMLU Professional Law (Hendrycks et al., 2020)

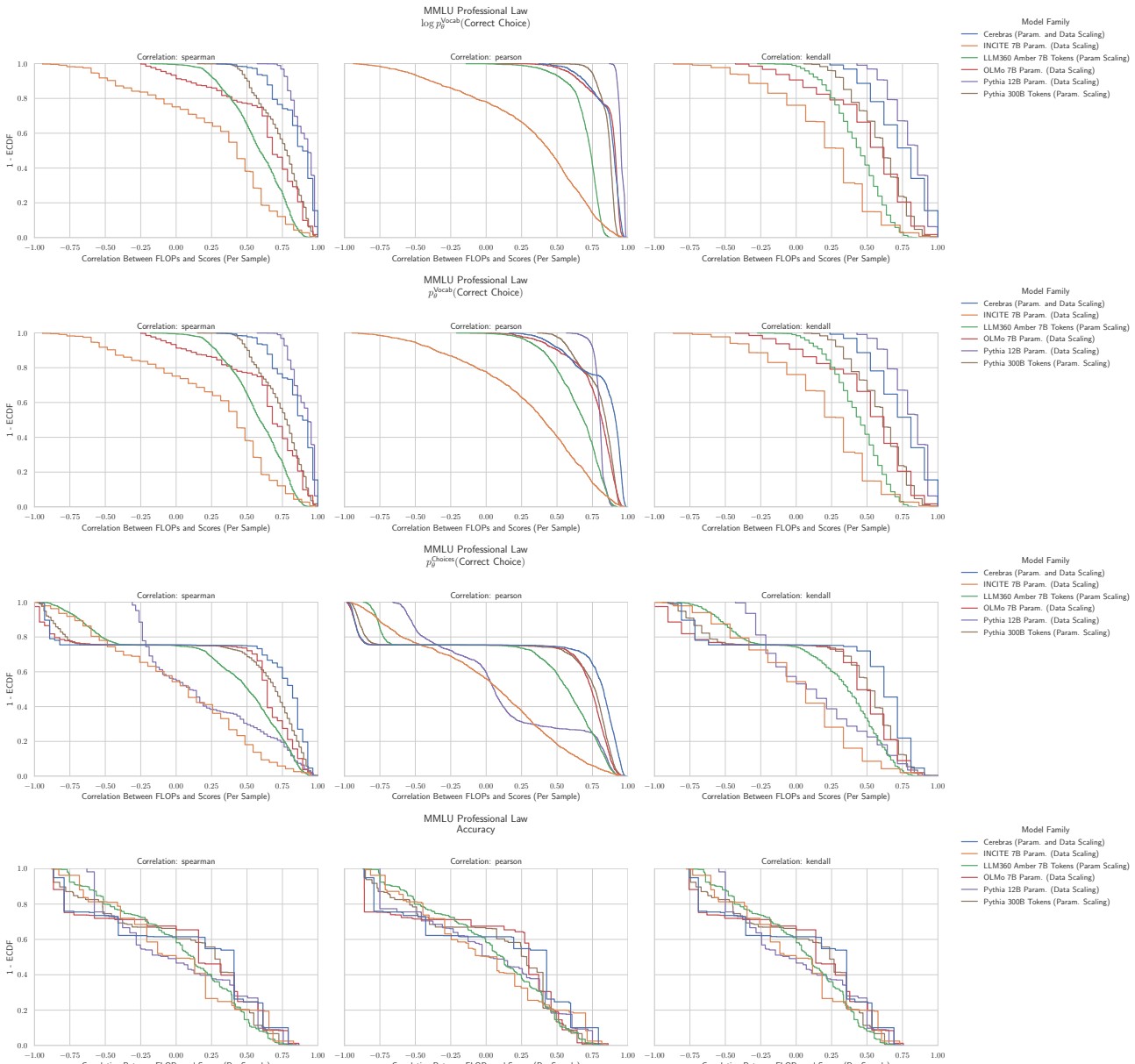

*Figure 64.* **MMLU Professional Law: Downstream performance is computed via a sequence of transformations that deteriorate correlations between scores and pretraining compute.**

## H.56. NLP Benchmark: MMLU Professional Medicine (Hendrycks et al., 2020)

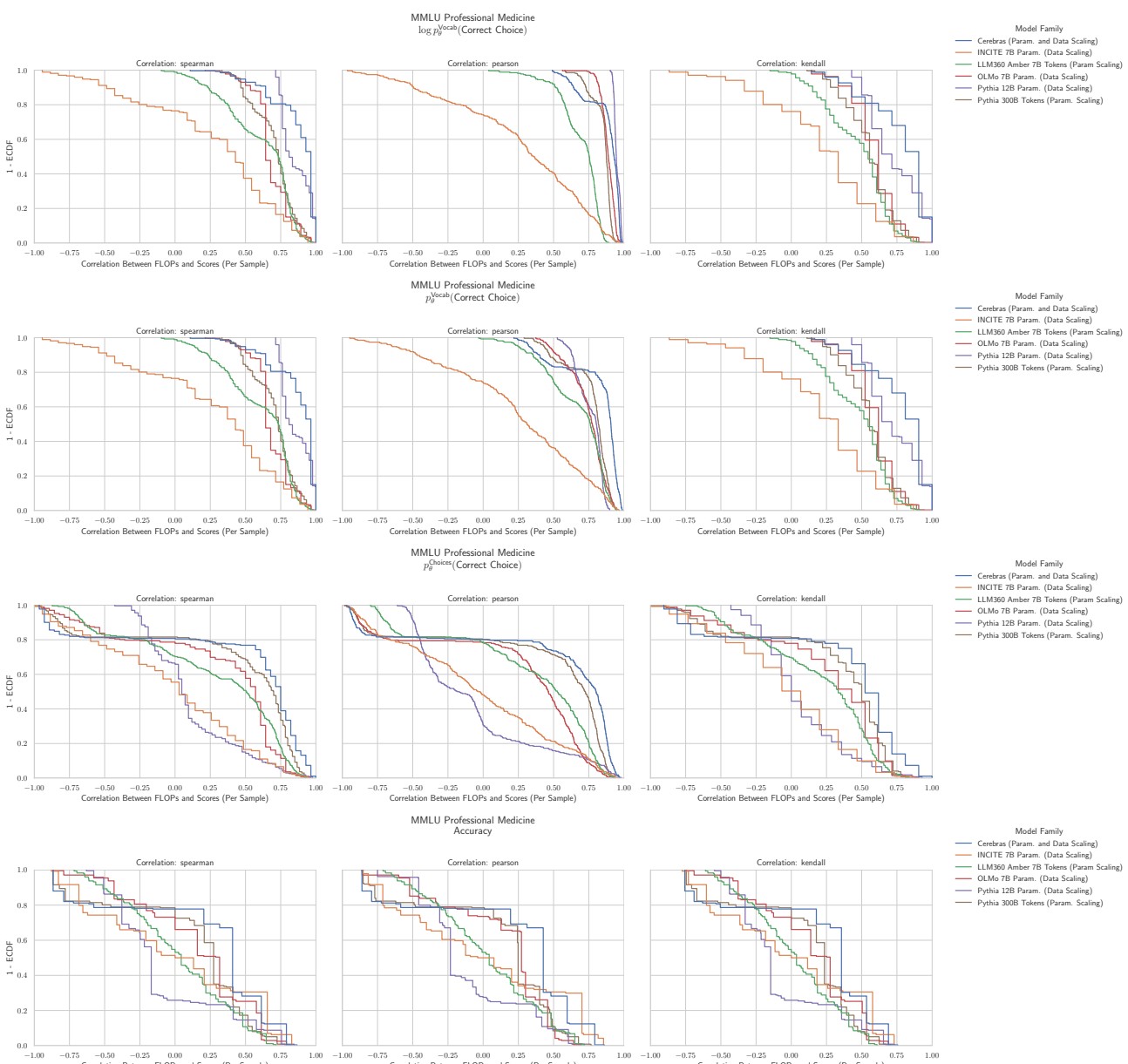

*Figure 65.* **MMLU Professional Medicine: Downstream performance is computed via a sequence of transformations that deteriorate correlations between scores and pretraining compute.**

### H.57. NLP Benchmark: MMLU Professional Psychology (Hendrycks et al., 2020)

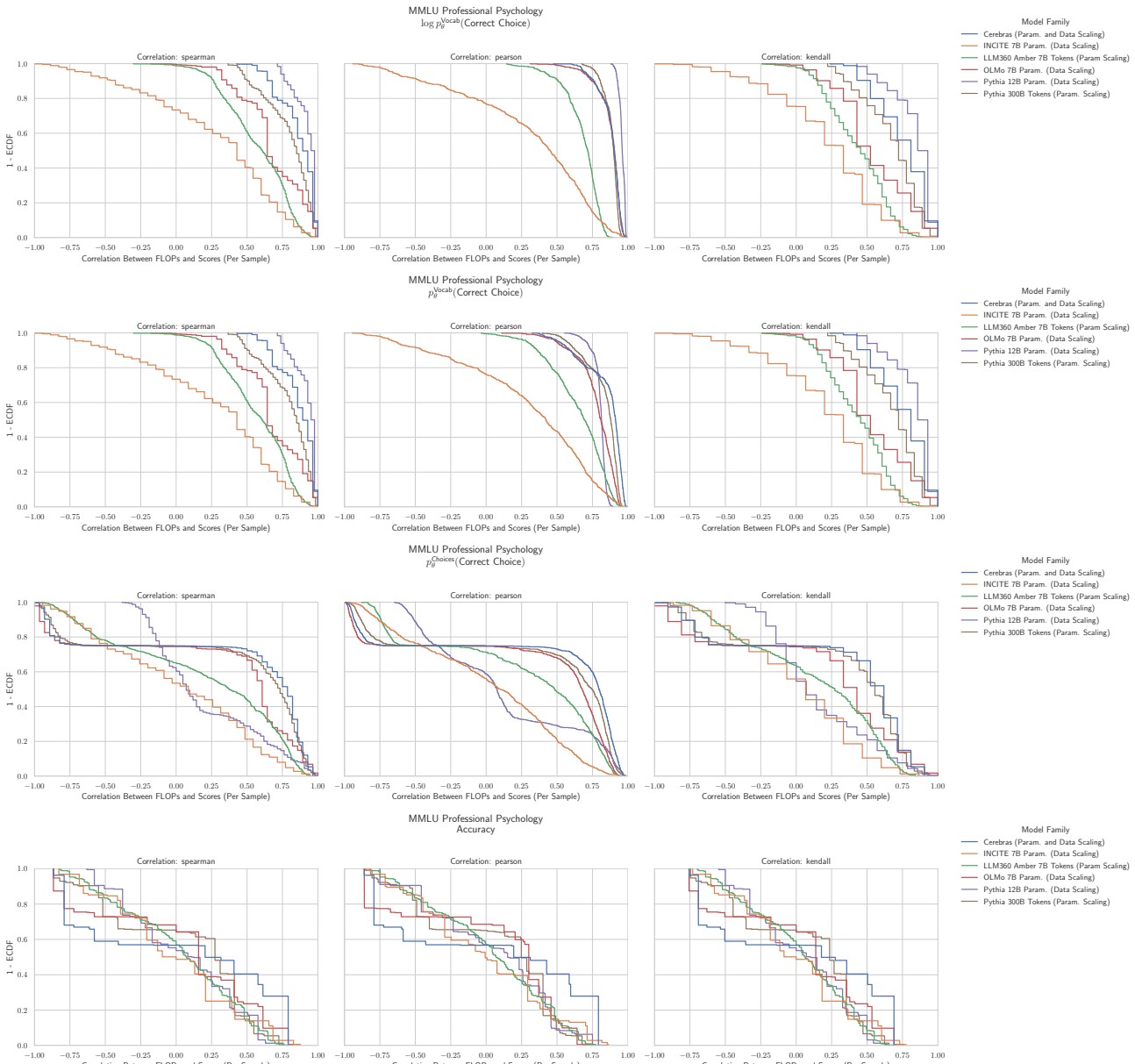

*Figure 66.* **MMLU Professional Psychology: Downstream performance is computed via a sequence of transformations that deteriorate correlations between scores and pretraining compute.**

## H.58. NLP Benchmark: MMLU Public Relations (Hendrycks et al., 2020)

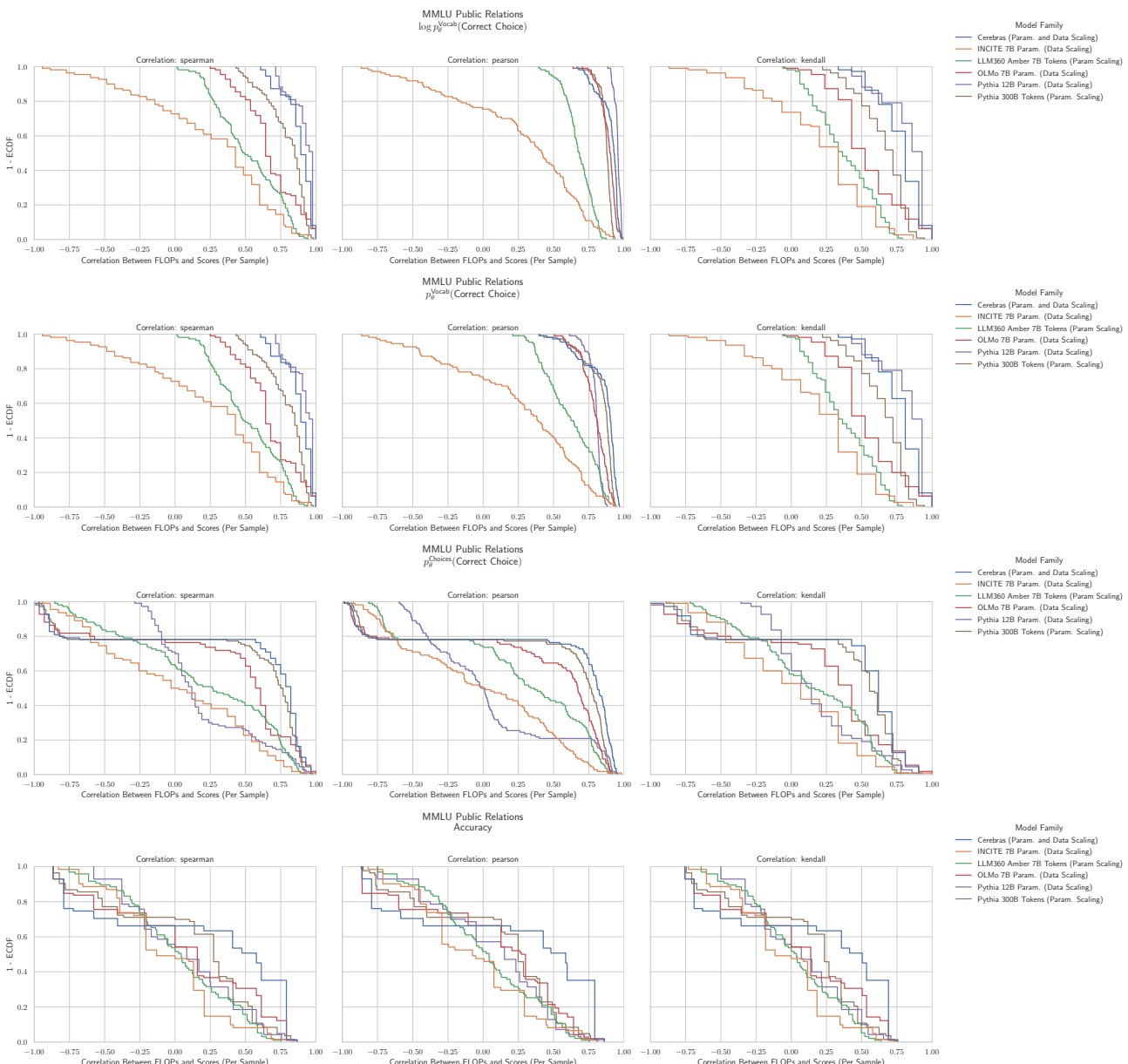

*Figure 67.* **MMLU Public Relations: Downstream performance is computed via a sequence of transformations that deteriorate correlations between scores and pretraining compute.**

## H.59. NLP Benchmark: MMLU Security Studies (Hendrycks et al., 2020)

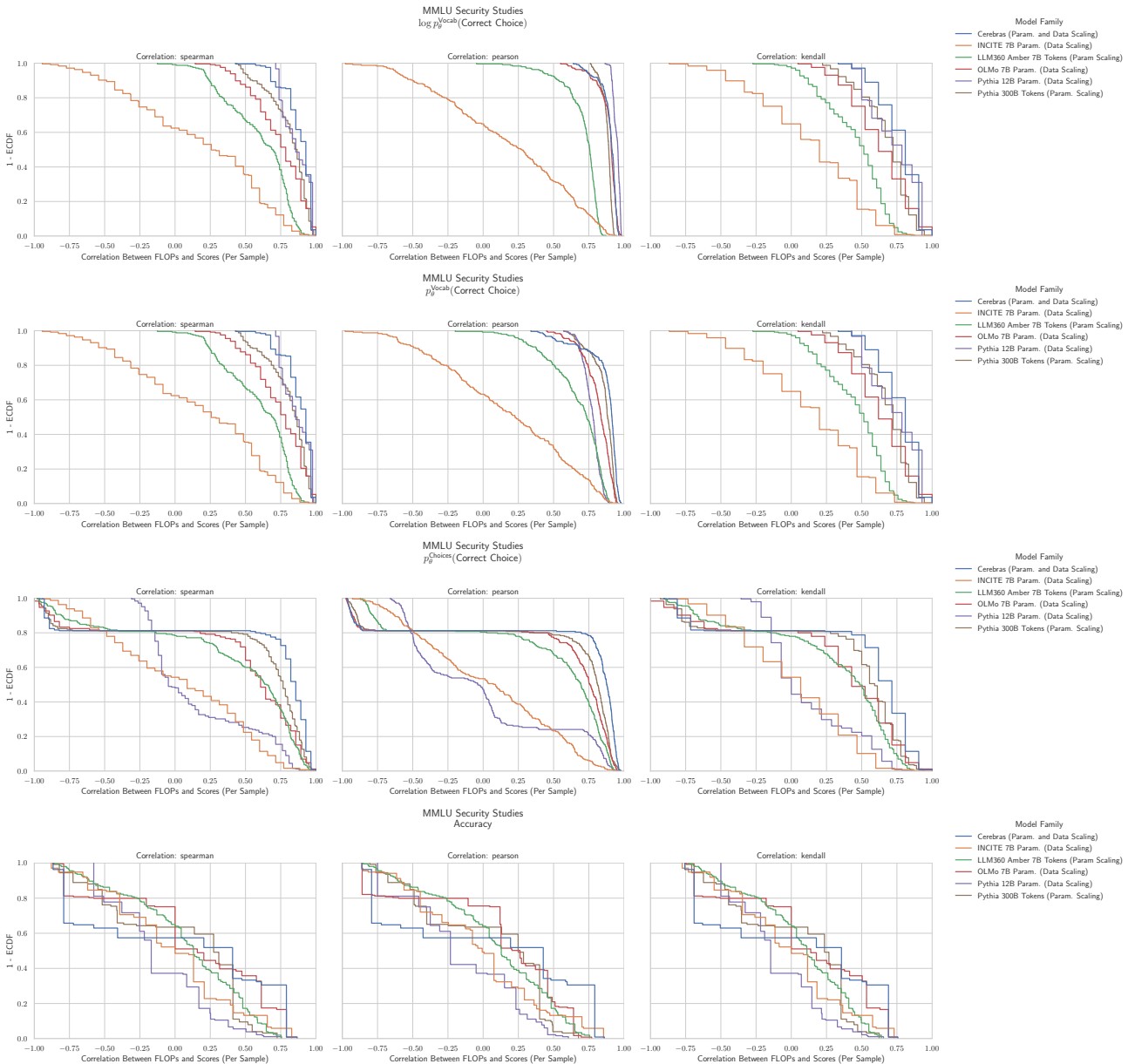

*Figure 68.* **MMLU Security Studies: Downstream performance is computed via a sequence of transformations that deteriorate correlations between scores and pretraining compute.**

## H.60. NLP Benchmark: MMLU Sociology (Hendrycks et al., 2020)

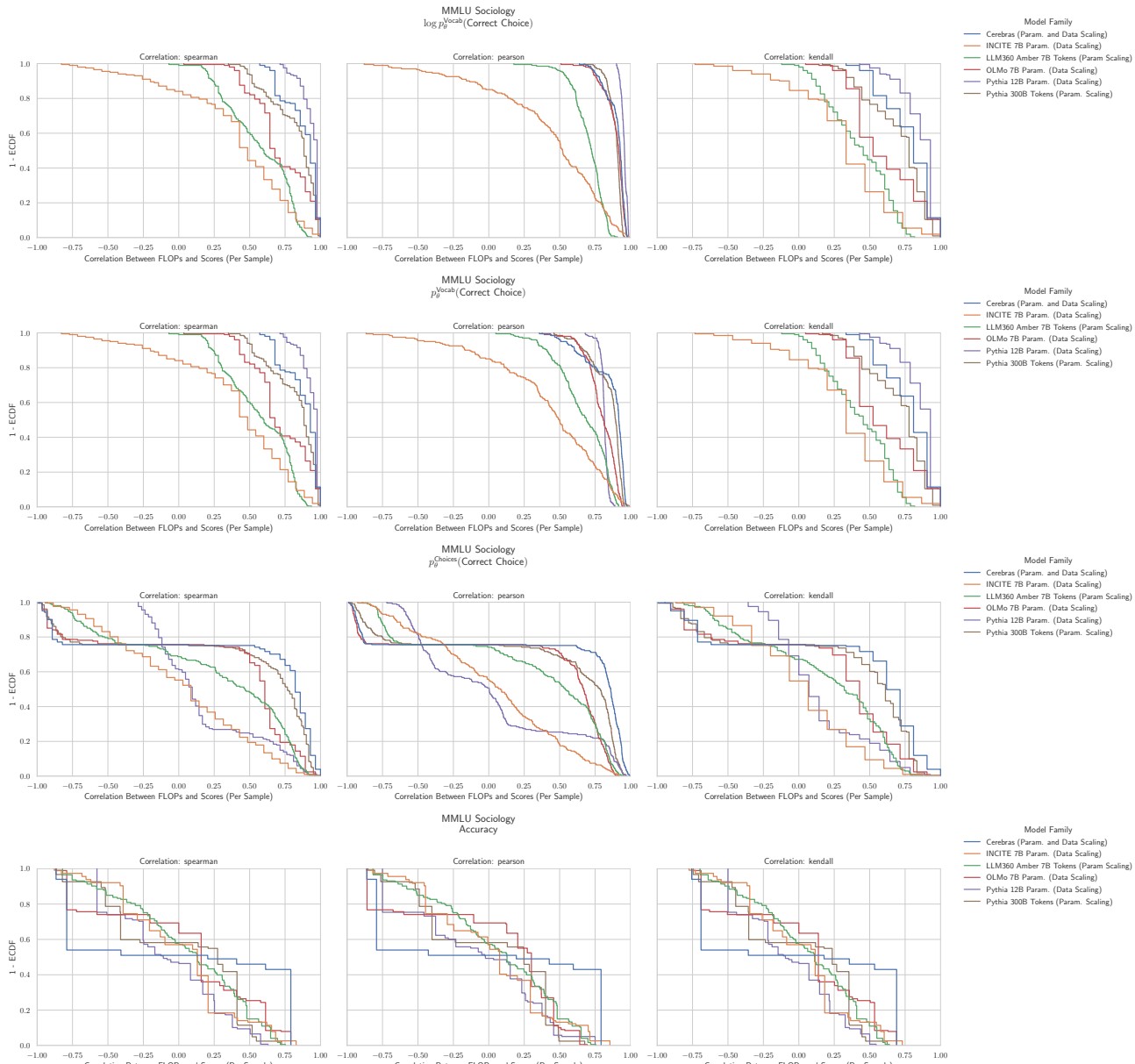

*Figure 69.* **MMLU Sociology: Downstream performance is computed via a sequence of transformations that deteriorate correlations between scores and pretraining compute.**

## H.61. NLP Benchmark: MMLU US Foreign Policy (Hendrycks et al., 2020)

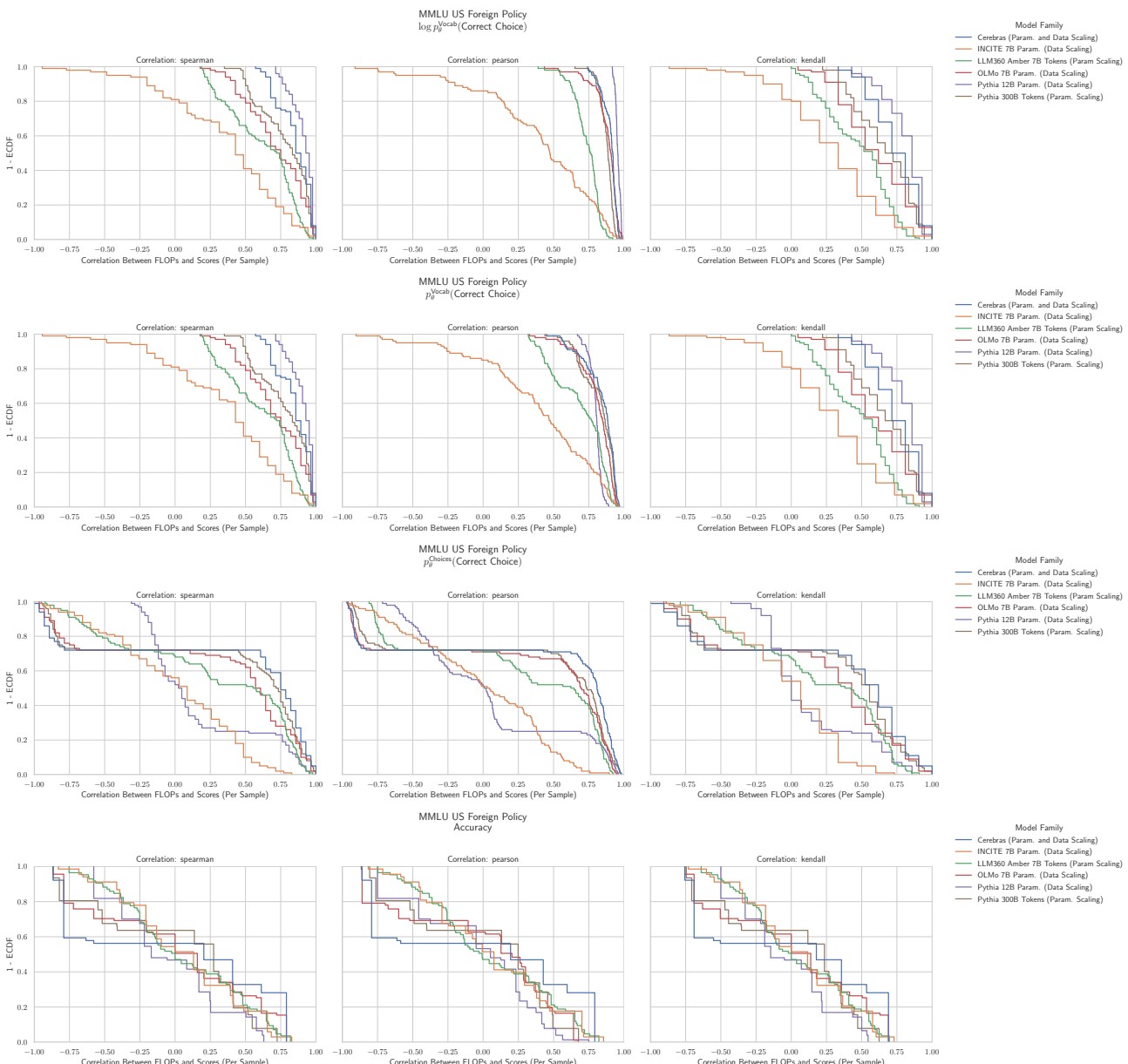

*Figure 70.* **MMLU US Foreign Policy: Downstream performance is computed via a sequence of transformations that deteriorate correlations between scores and pretraining compute.**

## H.62. NLP Benchmark: MMLU Virology ([Hendrycks et al., 2020](#))

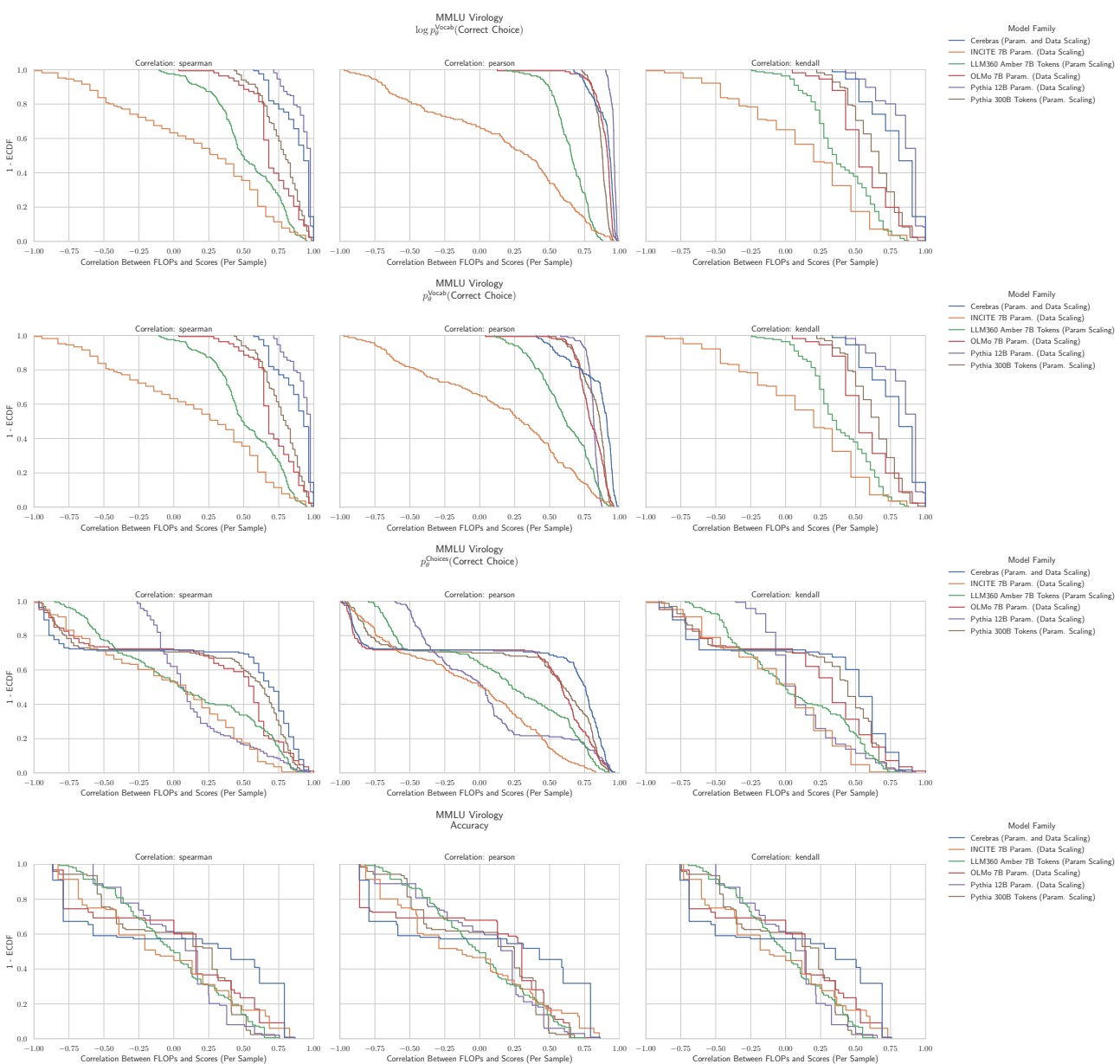

*Figure 71.* **MMLU Virology: Downstream performance is computed via a sequence of transformations that deteriorate correlations between scores and pretraining compute.**

## H.63. NLP Benchmark: MMLU World Religions (Hendrycks et al., 2020)

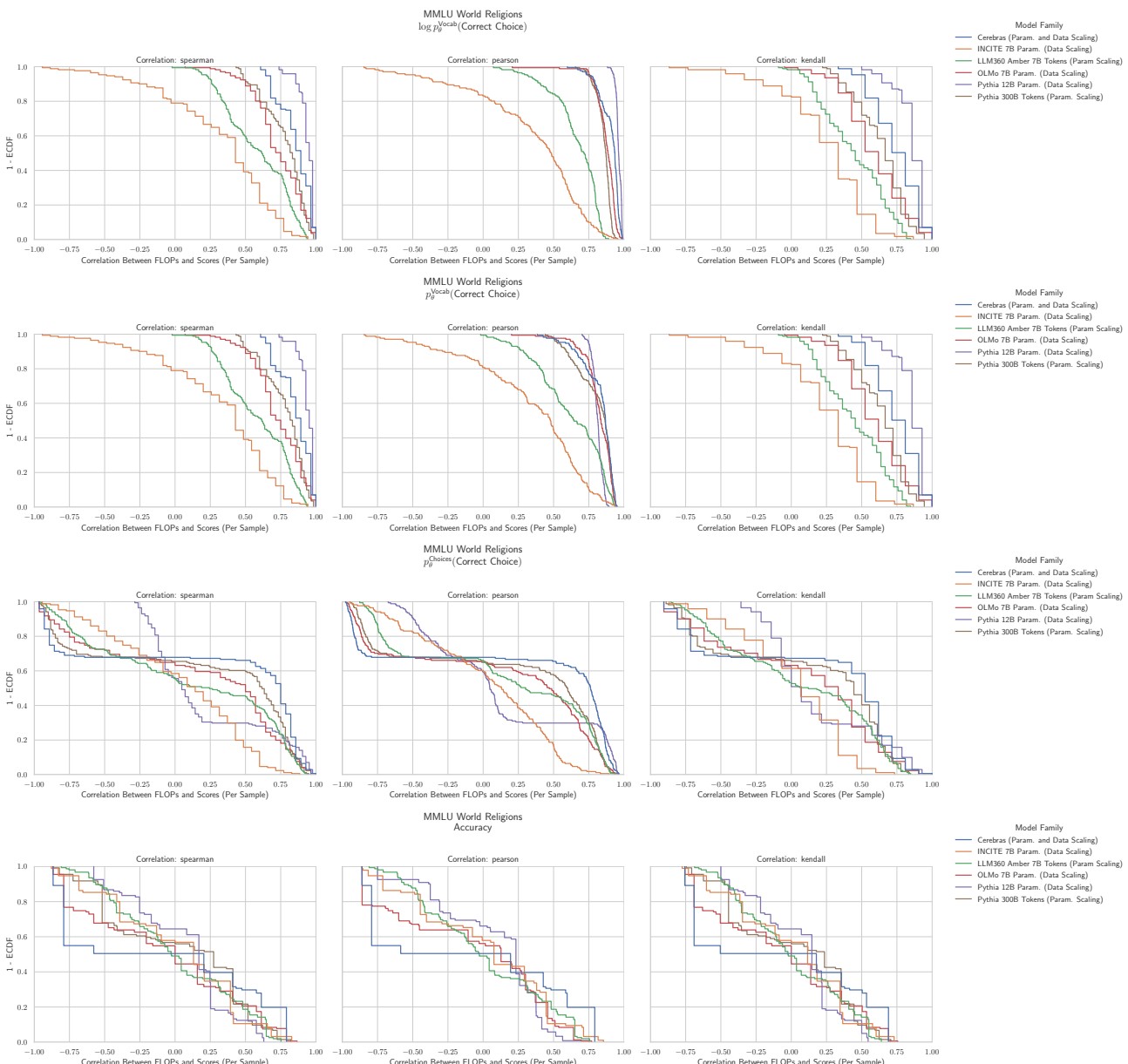

*Figure 72.* **MMLU World Religions: Downstream performance is computed via a sequence of transformations that deteriorate correlations between scores and pretraining compute.**

## H.64. NLP Benchmark: OpenBookQA ([Mihaylov et al., 2018](#))

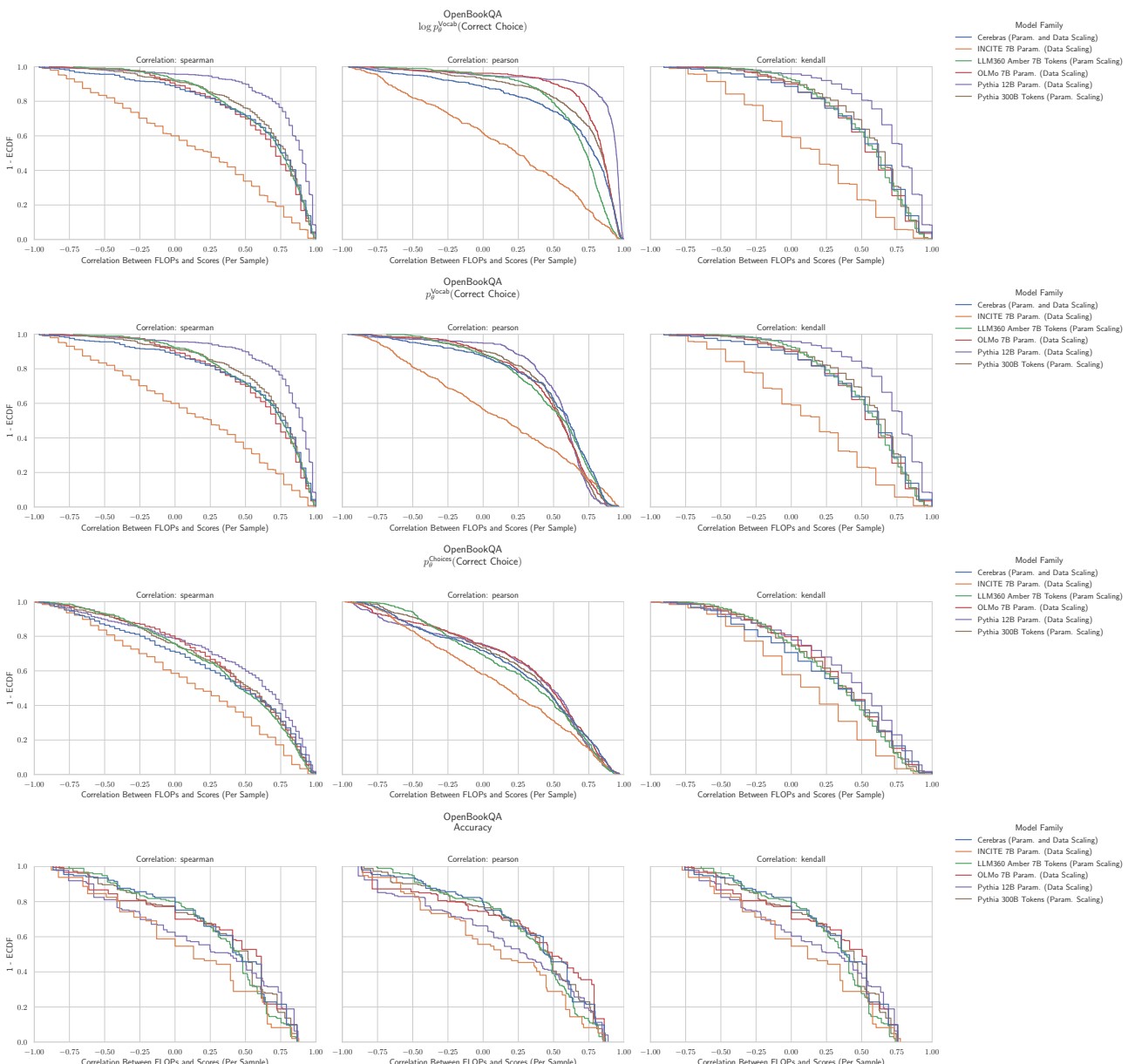

*Figure 73.* **OpenBookQA: Downstream performance is computed via a sequence of transformations that deteriorate correlations between scores and pretraining compute.**

## H.65. NLP Benchmark: PIQA (Bisk et al., 2020)

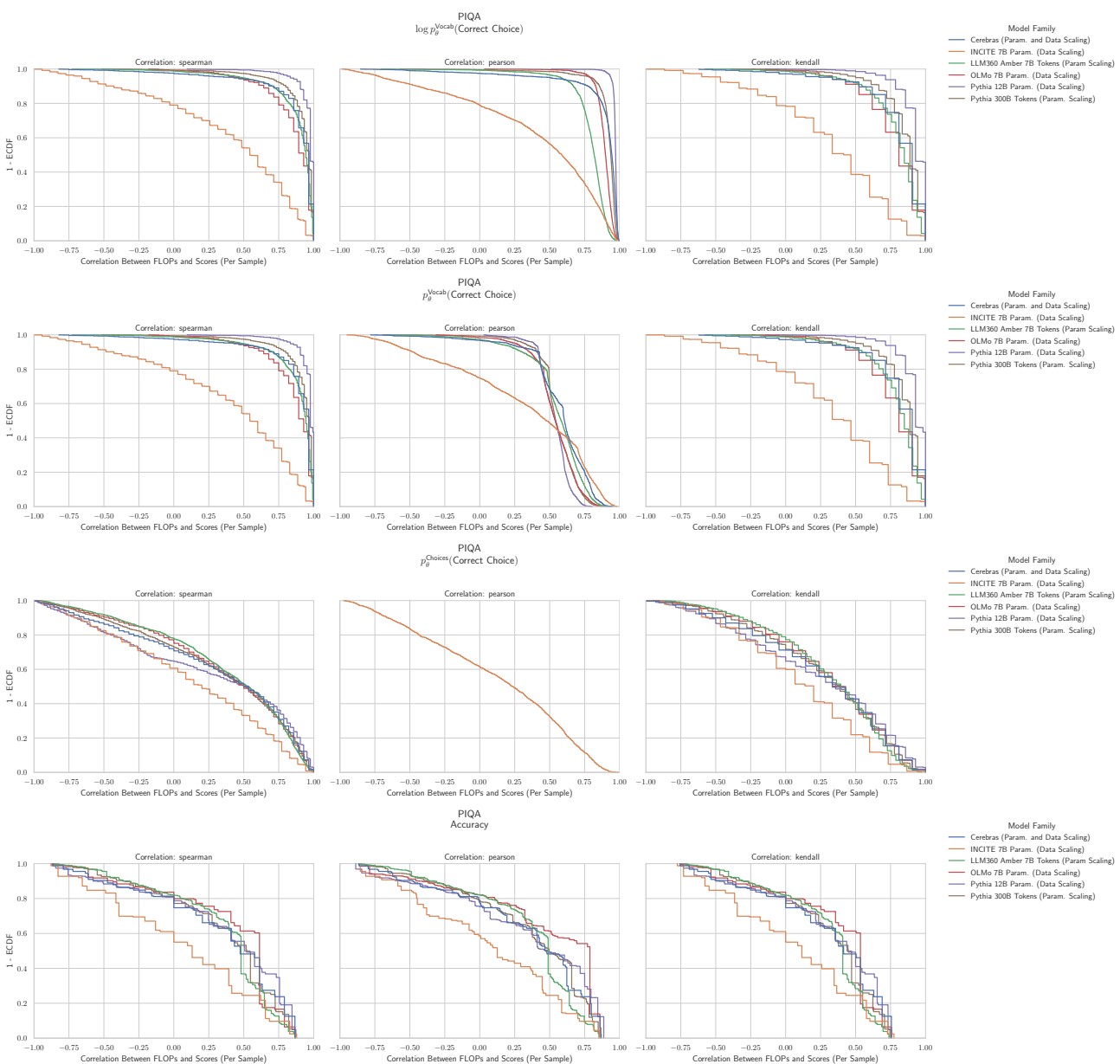

*Figure 74.* **PIQA: Downstream performance is computed via a sequence of transformations that deteriorate correlations between scores and pretraining compute.**

## H.66. NLP Benchmark: RACE (Lai et al., 2017)

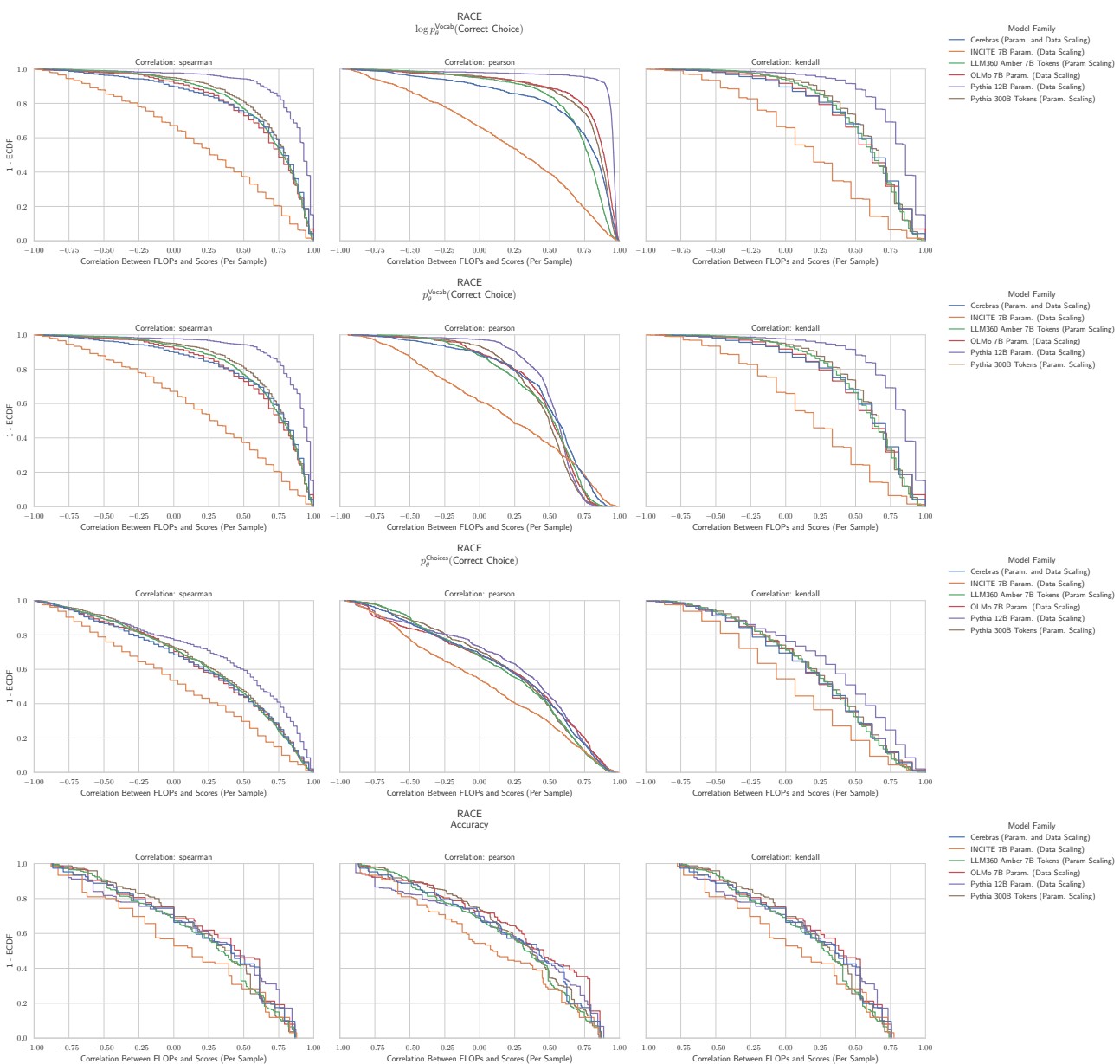

*Figure 75.* **RACE: Downstream performance is computed via a sequence of transformations that deteriorate correlations between scores and pretraining compute.**

## H.67. NLP Benchmark: SciQ (Welbl et al., 2017)

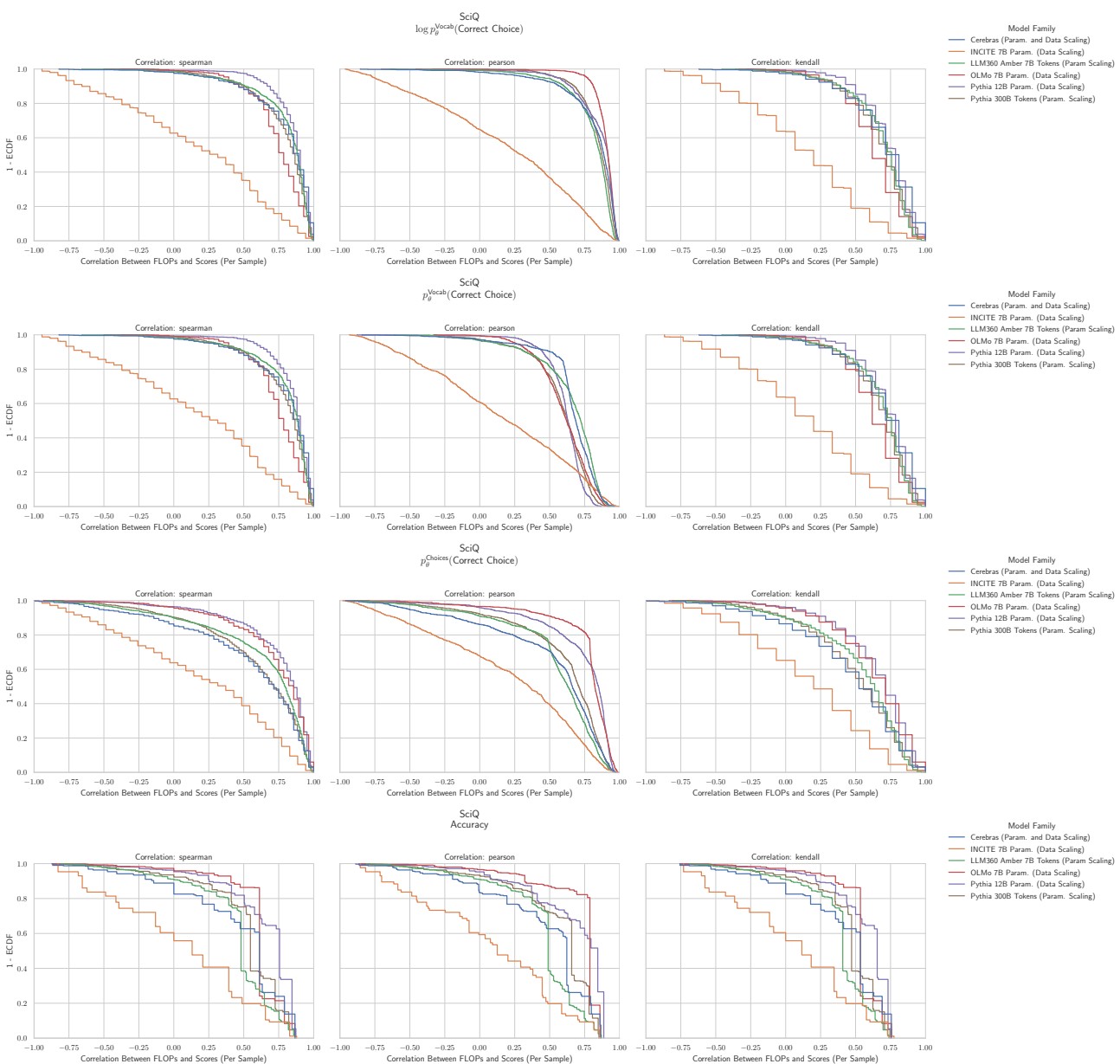

*Figure 76.* **SciQ: Downstream performance is computed via a sequence of transformations that deteriorate correlations between scores and pretraining compute.**

## H.68. NLP Benchmark: Social IQA (Sap et al., 2019b)

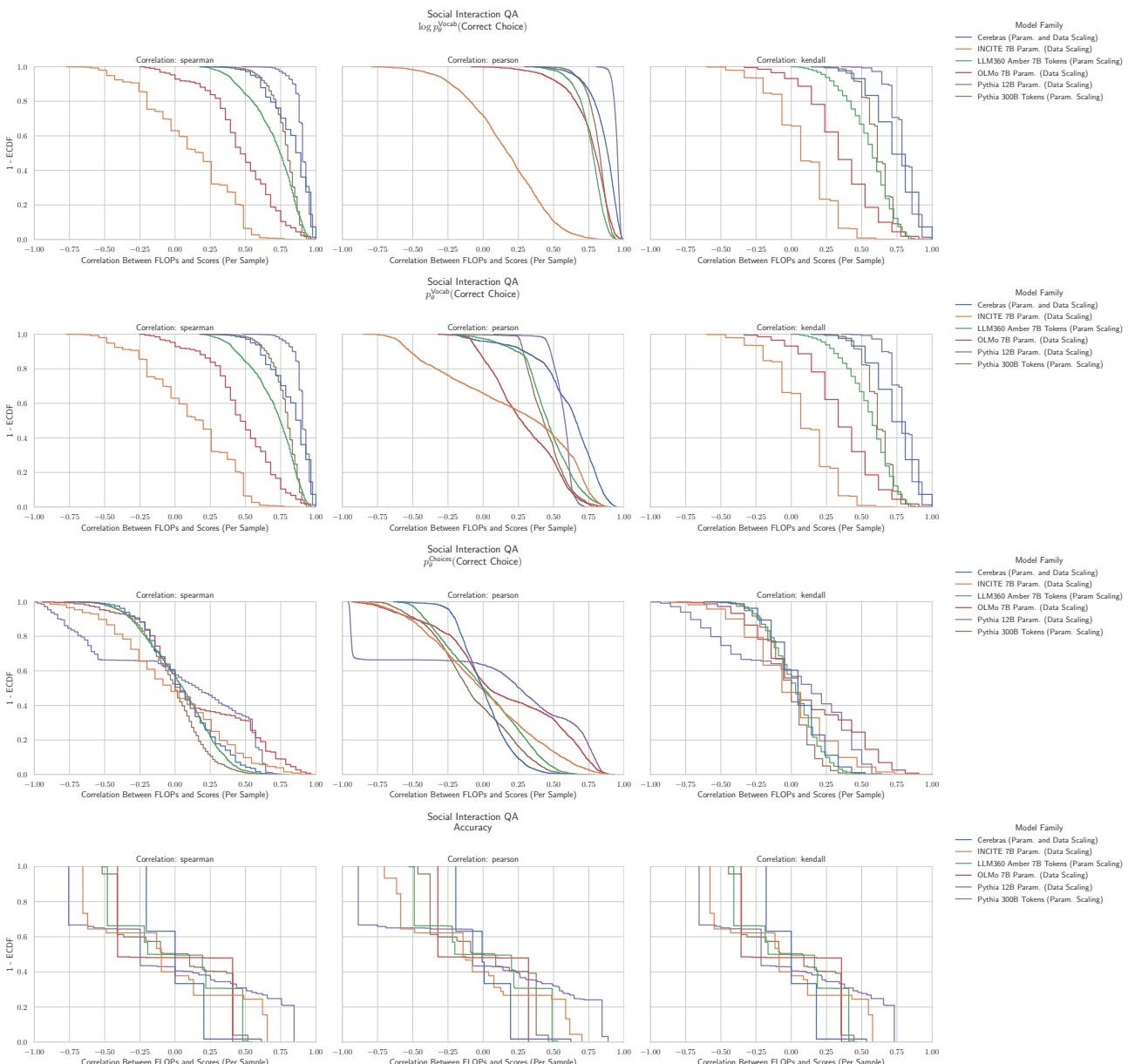

*Figure 77.* **Social IQA: Downstream performance is computed via a sequence of transformations that deteriorate correlations between scores and pretraining compute.**

## H.69. NLP Benchmark: Winogrande ([Keisuke et al., 2019](#))

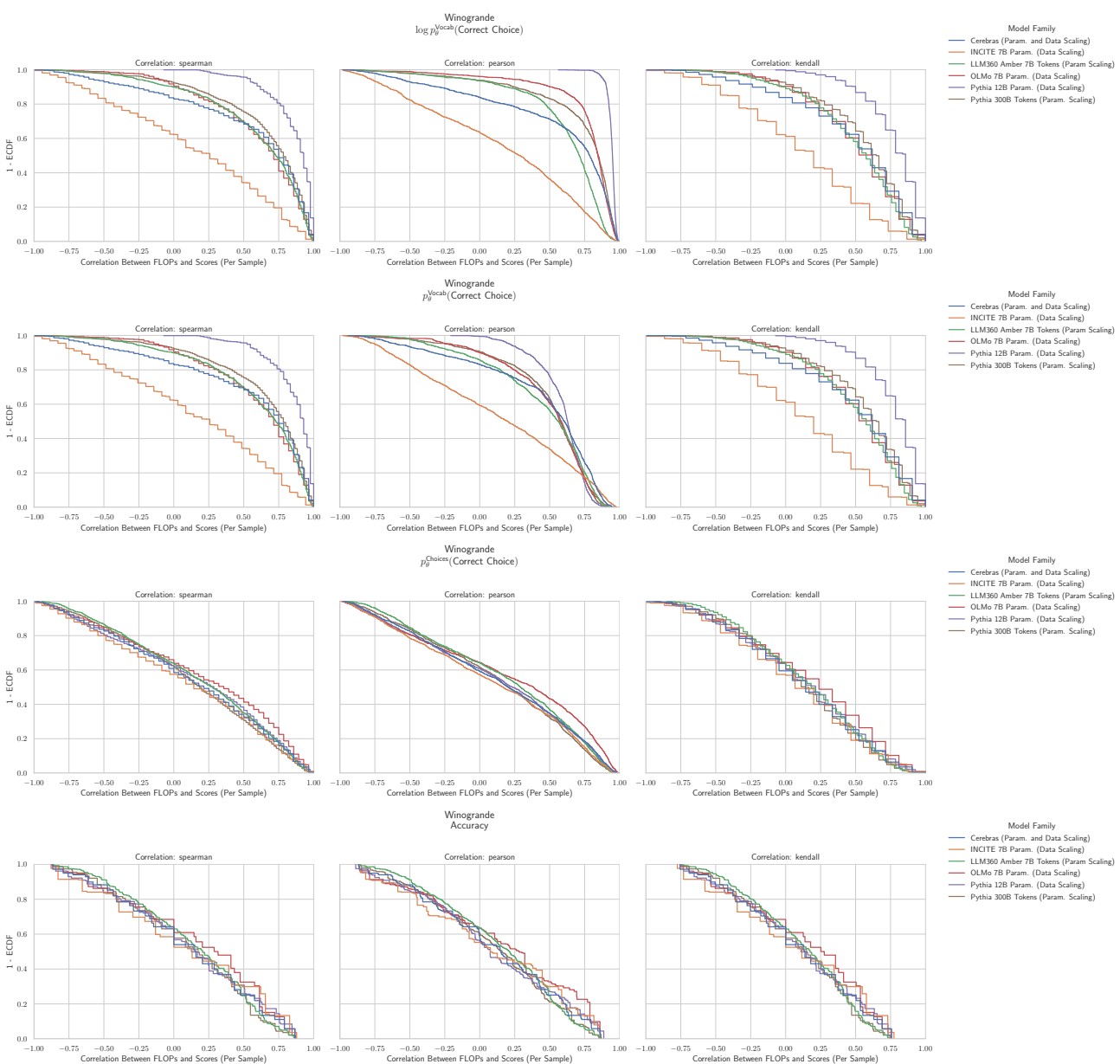

*Figure 78.* **Social IQA: Downstream performance is computed via a sequence of transformations that deteriorate correlations between scores and pretraining compute.**

## H.70. NLP Benchmark: XWinograd English (Muennighoff et al., 2023)

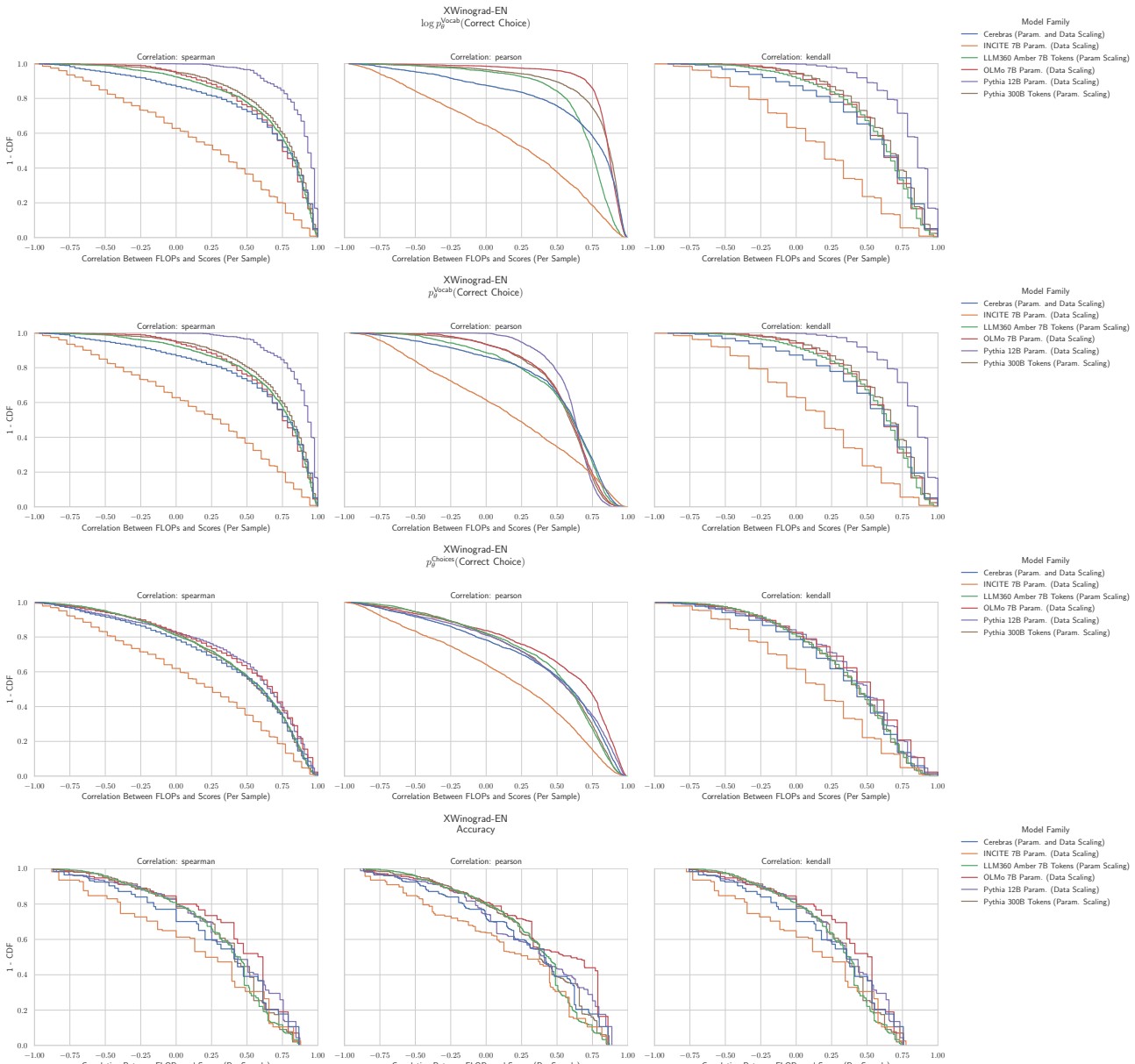

*Figure 79.* **XWinograd English: Downstream performance is computed via a sequence of transformations that deteriorate correlations between scores and pretraining compute.**

