# OpenReview forum: "Why Has Predicting Downstream Capabilities of Frontier AI Models with Scale Remained Elusive?"
_ICML.cc/2025/Conference — ICML 2025 poster_

### Official Review · Reviewer_Zmym · 2025-03-13

**Overall Recommendation:** 3

**Summary:**

The paper investigated why we can not accurately transform the scaling laws from the general negative token likelihood to the downstream accuracy of multi-choice QA.
The main reasons are accountable for the phenomenon are 1. there is a sequence of transformation from NLL to accuracy; 2. the positive correlation of scaling up compute and wrong choice probabilities.

**Claims And Evidence:**

Yes

**Essential References Not Discussed:**

No.

**Experimental Designs Or Analyses:**

Yes, the author runs through the popular LLMs with middle pertaining checkpoints and popular QA benchmarks.
It is a good experimental setting for the question they would like to explore.

**Methods And Evaluation Criteria:**

Yes

**Other Comments Or Suggestions:**

The paper is pretty refined. But I do think the authors complicates things in several ways.
For example, using a pdf is way more straightforward than using a cdf, people can well distinguish where there is of high density.
And also, the using of wording "the fraction of accuracy=1" I believe "accuracy" would be enough.

**Other Strengths And Weaknesses:**

Paper is pretty clear and the question they tried to investigate is important.

**Questions For Authors:**

Since we know that the trasnformations bring the detoriation to the correlation, I would like to how how much each step contribute to this.

**Relation To Broader Scientific Literature:**

I think it is highly related to the literature on how to evaluate the LLM during training.

**Theoretical Claims:**

No

---

> ### Author Rebuttal · Authors · 2025-03-30
>
> We sincerely thank Reviewer Zmym for their thoughtful assessment of our work, particularly noting that our paper is "pretty clear" and addresses an "important" question with "good experimental settings."
>
> ### Quantifying Score-Compute Decorrelation Per Transformation
>
> > Since we know that the trasnformations bring the detoriation to the correlation, I would like to how how much each step contribute to this.
>
> This is an incisive question. While the answer depends on the model family and the benchmark, Figure 4 suggests the answer, but  we can be much more quantitative. We will create a new figure with 4 subfigures, one for each of the statistics of the score-compute correlation distributions (mean, median, AUC of CDF, and negative Wasserstein distance). The x axis will be the sequence of metrics, the y axis will be the statistic’s value, and the hue will be the model family. This will show exactly how much “predictability” is lost under each transformation.
>
> If the paper is accepted, we will include this detailed quantitative breakdown in the final version, including a new table summarizing these contributions across benchmarks.
>
> ### On Visualizing Distributions
>
> > using a pdf is way more straightforward than using a cdf, people can well distinguish where there is of high density.
>
> We appreciate the suggestion about using PDFs instead of CDFs. We actually explored both approaches during our analysis. While PDFs can be more intuitive in some contexts, we found that CDFs (specifically complementary CDFs) offered three advantages for our particular analysis: (1) they avoid bandwidth parameter tuning artifacts that emerged with KDE-based PDF estimation; (2) they directly visualize the tail behavior critical for our analysis; and (3) they more clearly quantify "what fraction of samples have correlations above X" - the key relationship we wanted to highlight. We'd be happy to include a comparison of both visualization approaches in an appendix if you think this would benefit readers.
>
> ### Terminology
>
> Thank you for suggesting clearer wording. We used 'fraction of accuracy=1' to precisely distinguish between two concepts: (1) the proportion of individual samples where the model answered correctly (binary per-sample outcome), versus (2) the overall accuracy score across the dataset (continuous value between 0-1). While mathematically equivalent, maintaining this precision helps readers follow our sample-specific correlation analysis. We'll revise this wording to improve clarity while preserving this distinction.
>
> ### Request for Additional Constructive Criticism
>
> We value your assessment and would appreciate any additional specific suggestions that might strengthen the paper. In particular, are there specific analyses or clarifications about the transformation process that would make our findings more compelling or accessible to the broader ML community?
>
> We appreciate your 'weak accept' assessment and are committed to strengthening the paper to earn a more enthusiastic endorsement. We believe the quantitative breakdown of correlation degradation across transformations will substantially enhance the paper's contributions and clarity.
>
> Thank you again for your constructive engagement with our work.

---

### Official Review · Reviewer_M3MH · 2025-03-24

**Overall Recommendation:** 2

**Summary:**

This paper explores why predicting the downstream performance of advanced AI systems, especially on multiple-choice question-answering benchmarks, is difficult despite well-understood scaling laws during pre-training. The key finding is that downstream performance degrades as it involves comparing the correct choice to a small set of incorrect choices, making it harder to predict how probability mass shifts with scale. They highlight that to predict downstream capabilities accurately, it’s essential to account for how probability mass fluctuates between correct and incorrect choices as models scale. The study suggests that scaling laws for incorrect choices may be achievable.


## update after rebuttal
My main concern still stands. I believe the study of "unpredictable" scaling laws is narrow scoped towards MCQ benchmarks. And even there the finding is rather superficial: probability mass on incorrect samples increases unpredictably. It is unclear why this happens. The authors agree, that eventually if we scale enough then this should not be a concern since pre-training loss is a consistent objective. If so, then is that point (where we enough data to predictably scale performance on MCQ) itself predictable? Many questions like these remain unanswered. Because of this, I am leaning towards a weak reject, **though after reading other reviews, I am willing to raise my score by 0.5 to 2.5, and would not be opposed to accepting this paper if other reviewers are willing to champion it**.

**Claims And Evidence:**

- There are some scaling-predictable quantities (from parameters, data, compute), like log-prob over vocabulary. But, these under go transformations, that make the scaling predictable quantities less predictable.
- Downstream performance depends a lot on the probability mass assigned to the incorrect choice, with scale. Suggests that we need to model external information different from scale-predictable quantities like log prob.
- Continuous metrics like Brier score are insufficient to predict downstream performance.

**Essential References Not Discussed:**

- Language Models (Mostly) Know What They Know (Kadavath et. al.), also conducts an analysis of performance and calibration on multiple choice questions benchmarks, albeit they don't focus on predictable scaling.

**Experimental Designs Or Analyses:**

Their experimental protocol is stated cleanly -- I really like their Figure 3, which presents the degradation of predictive power on the ARC benchmark. In general, they follow the following protocol.

- The authors compute the correlation between score and compute, as we scale compute in a given model family.
- The score metric is one of the following: p(correct choice | vocab), p(correct choice | given other choices), accuracy, brier score.
- Then, they plot the distribution of this correlation across samples.
- Ideally, this distribution should be concentrated at values close to 1.0.
- But, they find that everytime some transformation is applied on the log-probs predicted by the model, the correlation degrades, and the distribution of this correlation shifts away from 1.0.

They find that the main reason for the above degradation in correlation is the probability mass on incorrect choices.

**Methods And Evaluation Criteria:**

Yes, the paper uses some common benchmarks and model families (though it does not cover Llama or Gemma model families which are common open source models).

- The paper conducts scaling analysis on a comprehensive set of model families with multiple models (scaling data, parameters and compute), including Pythia, Cerebras-GPT, OLMo, INCITE, LLM360.
- The work only evaluates downstream performance on NLP benchmarks, for which it uses: ARC, HellaSwag, MathQA, MCTACO, MMLU, OpenbookQA, PIQA, RACE, SciQ, SIQA, WinoGrande and XWinoGrad En.
- The performance is measured in terms of accuracy, Brier score, and probability mass on the correct choice.

**Other Comments Or Suggestions:**

- The plot labels (result plots) are very small and hard to read.

**Other Strengths And Weaknesses:**

Strengths:
- The analyses of downstream prediction capabilities on MCQ benchmarks is quite comprehensive, and it gets at the main cause: the probability mass on incorrect outputs varies in unpredictable ways as we scale compute.
- The story is presented in a clean way, and the paper is easy to read.

Weaknesses:
- The analyses and findings are very heavily focussed on MCQ benchmarks. It is unclear why or how the predictability scales on open-ended QA benchmarks, or reasoning problems.
- It mentions that the problem mainly stems from unpredictable movement of probability mass on the incorrect options, but does not discuss any ways in which this can be fixed, or is this a fundamental limitation?
- Intuitively, one might imagine that the probability mass on all incorrect samples should reduce monotonically with scale, so why does it increase on some, and decrease on others? The paper does not explain this. An investigation of this underlying cause seems needed to really understand if the lack of predictability is fundamental, even for MCQ benchmarks.

In general, I feel that the analysis is interesting. It mechanistically and systematically explains why the correlation degrades but somehow leaves the reader wanting by the end of Section 6, at which point it still remains unclear why the probability on incorrect samples has a high and un-predictable sensitivity with scale. Also, it is unclear how to transfer the analysis or insights to non MCQ benchmarks (I did look at the discussion in Appendix B which mainly defers discussion to future work). Because of these two points, I am leaning towards a weak reject, but if the authors can add additional discussion on these points, I would be happy to re-consider my score.

**Questions For Authors:**

- In Figure 6, do the trends change depending on the number of incorrect options present for each question. For example, if there are more incorrect options, then is the join probability distribution less correlated?
- How is the analysis affected by the relationship between correct and incorrect options? For example, if the incorrect answers are close negatives, vs. being very different negatives.
- For True/False questions, is the scaling more predictable?

**Relation To Broader Scientific Literature:**

The paper talks about predictable downstream performance, as a function of scaling data, parameters and compute for pre-training. Existing analyses in this space has been centered on predicting loss (NLL) on test-data, either in data rich or data constrained settings. From that perspective, this analyses is novel, but the concern is that it is too narrow (only focused on MCQ), and only presents some concerns, as opposed to also presenting practically implementable fixes to make the downstream performance more predictable.

**Theoretical Claims:**

There are no theoretical claims in the paper.

---

> ### Author Rebuttal · Authors · 2025-03-30
>
> We sincerely thank you for your thorough and thoughtful review. We greatly appreciate that you found our analysis "interesting," "quite comprehensive," and that it "gets at the main cause" of unpredictability in downstream capabilities. We're particularly pleased that you found our story "presented in a clean way" and that the "paper is easy to read." Thank you also for highlighting Figure 3 as effectively demonstrating the degradation of predictive power on the ARC benchmark.
>
> ### On Specificity to Multiple Choice Question Answering (MCQA) Benchmarks
>
> We agree that our focus on MCQA benchmarks represents a scope limitation. We are upfront about this but made this decision for several reasons:
>
> 1. MCQA benchmarks remain a widely used evaluation format for frontier models due to their objective scoring and standardization. They're employed by major benchmarking efforts (e.g., MMLU, HELM) and by leading labs to track progress.
> 2. The MCQA format is particularly valuable for investigating scaling predictability because it allows us to isolate specific transformation steps in a controlled manner. This methodological clarity helped us identify the core mechanism behind unpredictability.
> 3. We view our manuscript as contributing to the science of scaling-predictable evaluations. While any given task or type of task will have its quirks, our paper shows how one can dive into the mechanisms of apparent unpredictability For example, new work on studying predictable scaling behavior of inference-time compute (https://arxiv.org/abs/2502.17578) also applied a per-sample analysis, albeit in a different context. We will add a citation to this work and clarify how analyses like ours can be applied to generative evaluations.
>
> ### Fundamental Limitation or Temporary Obstacle
>
> > It mentions that the problem mainly stems from unpredictable movement of probability mass on the incorrect options, but does not discuss any ways in which this can be fixed, or is this a fundamental limitation?
>
> This is certainly not a fundamental limitation and we have subsequent work showing how to overcome this limitation by predicting how probability mass changes on incorrect choices. However, demonstrating and dissecting the problem thoroughly is itself already quite long (as you can tell by our manuscript). We will move our Future Directions section into the main text if accepted.
>
> ### Clarification of How Mass Changes on Incorrect Choices
>
> > Intuitively, one might imagine that the probability mass on all incorrect samples should reduce monotonically with scale...
>
> This was our belief initially, but probability mass on incorrect choices almost always increases with compute, at least in the models we were able to study. For example, suppose we have a question about what pet a child has. As models train, (1) they place more mass on syntactically valid words and phrases, so if an incorrect option is "airplane", its mass will still increase relative to a randomly initialized model, even though airplane isn't a valid pet ; (2) they place more mass on semantically plausible alternatives e.g., "cat" can be a likely pet, even if it isn't the correct option for this question; (3) they learn the nature of MCQA tasks and place more mass on the available options regardless of content.
>
> At some point, probability mass on incorrect choices _must_ decrease, but it seems the models we studied are not close to this tipping point.
>
> ### Responses to Specific Questions
>
> > In Figure 6, do the trends change depending on the number of incorrect options present for each question?
>
> > For True/False questions, is the scaling more predictable?
>
> This is an excellent question. Most widely-used MCQA NLP benchmarks have 4 options. MMLU-Pro (https://github.com/TIGER-AI-Lab/MMLU-Pro) but offers 4-10 choices per question, but was publicly released after our data were collected. We expect that if there are more incorrect options, scaling is less predictable and if there are fewer options (e.g., True/False), scaling would be more predictable. We will add this to our Discussion.
>
> ### Visual Presentation Improvements
>
> We appreciate your feedback about plot label sizes. We will increase font sizes in all figures.
>
> ### Conclusion
>
> We believe addressing these concerns will substantially strengthen the paper and hope they address your "weak reject" assessment. Thank you again for your constructive feedback, which has genuinely helped us improve our work.

---

> > ### Comment · Reviewer_M3MH · 2025-04-05
> >
> > Thank you for the response. I am still unsure about the intuition behind why probability mass increase on the incorrect response is the main cause behind the scaling being hard to predict. Since, log-loss is a consistent objective, as we minimize pre-training  loss for the next-token prediction objective, it should be the case, that the most likely option based on the contexts seen in the pre-training data should be preferred (over the other MCQ options). If the most likely option is incorrect, then there is task mismatch between pre-training and fine-tuning task, and that seems to be a distribution shift problem. While it is plausible (as shown in the paper) that probability mass increases on the negative samples, but if we train long enough, then the mass on the most likely (and correct if task matches pre-training) sample should still be higher. For example, "airplane" may a viable option for a "pet" and may have some mass increase on this option, but this should be smaller than "cat" or "dog" eventually (at least relatively).

---

> > > ### Author Response · Authors · 2025-04-08
> > >
> > > Dear Reviewer M3MH,
> > >
> > > Thank you for your follow-up comment and for engaging deeply with our work.
> > >
> > > Your intuition that minimizing pre-training loss should **eventually** favor the most likely (and hopefully correct) continuation is correct in the context of the next-token prediction task over the full vocabulary. Indeed, as our paper shows, the loss and the log-probability of the correct choice when considered over the entire vocabulary, $p_{\theta}^{Vocab}​(\text{Correct Choice})$, does correlate strongly and predictably with scale.
> > >
> > > However, **this description is only applicable after one scales up enough. Our paper studies: what happens before that point?** That is precisely the realm of scaling studies. In the context of multiple choice question answering (MCQA), performance on these benchmarks depends on the probability of the correct answer and also on the probability on a small, specific set of incorrect distractors provided within the question. Before one scales up enough, probability mass fluctuations on these specific incorrect distractors affect the model family's performance.
> > >
> > > Our empirical results across various model families and scales show that this unpredictability arising from incorrect choice probabilities is a tangible issue within current and near-term scaling regimes, especially for smaller models on scaling ladders.
> > >
> > > We hope this clarifies why the behavior of probability mass on incorrect choices is central to the challenge of predicting downstream MCQA performance, even given the consistency of the pre-training objective.
> > >
> > > Best regards,
> > >
> > > The Authors

---

### Official Review · Reviewer_AAE5 · 2025-04-01

**Overall Recommendation:** 3

**Summary:**

The paper studies the relationship between model scale and downstream task performance, to understand why it has been hard to formulate a "scaling law" to describe the relationship unlike known results for pretraining performance.
The paper conducts extensive empirical analyses on many tasks, benchmark datasets, model families and compute scales, to derive a comprehensive picture about their relationship to task performance.
The paper proposes an interesting explanation when tasks are specialized to multiple choice questions (MCQ) and back it with their analyses, that there are transformations going from pretraining loss to downstream MCQ performance that systematically degrade the statistical relationship between the two.
The paper also conjectures that measuring and accounting for the log likelihoods of incorrect answers in MCQs may yield a more stable relationship between model scale and task performance.

**Claims And Evidence:**

There are 3 main claims in the paper.
1. Downstream task performance does not have a predictable relationship with the scale of the model, whereas pretraining loss does. The claim can be scoped better by emphasizing that the downstream tasks are of multiple-choice question type. The evidence for the claim can be strengthened by including a plot showing task performance vs model scale (and contrasting it with the plots in the related literature for pretraining scaling laws).

2. As we go from the pretraining loss to the MCQ task objective, each transformation step weakens the statistical correlations with model scale. The evidence contributed by the paper for this claim is very comprehensive!

3. The variations in probability of incorrect choices do not "cancel out" when averaging across datapoints in a dataset. And modeling those probabilities (at a per sample granularity) is key to better prediction of downstream performance metrics as a function of model scale. The evidence for this claim is indirect at best. More direct lines of evidence may be to control for the probability of incorrect choices, and check whether there is a more predictable relationship between task performance and model scale after controlling for confounding.

**Essential References Not Discussed:**

N.A.

**Experimental Designs Or Analyses:**

The experiment design is sound, and quite an accomplishment in how thorough the empirical results are reported.

**Methods And Evaluation Criteria:**

The evaluation methods are sound.

**Other Comments Or Suggestions:**

Fig 5 center: The comment about "0 < P_choices(correct choice) < 0.5 contains little information about accuracy" is a little inscrutable. The plot shows not much dispersion at all for "Fraction of Accuracy=1", unlike Fig 5 left. I can see two explanations: one is that to understand the claim we need to look at the composition of Fig 5 left and Fig 5 center, and realize that a large dispersion in Fig 5 left has large downstream effect. The other explanation is that Accuracy is a binary variable (when measured per datum); there are only 2 values for accuracy and each accuracy value maps to a large spread of P_choices(correct choice) values.

**Other Strengths And Weaknesses:**

N.A.

**Questions For Authors:**

N.A.

**Relation To Broader Scientific Literature:**

The related work section in Appendix is good; I wondered if some abridged version of it could be moved to the main paper (perhaps streamlining Sections 1 and 2 in the process).

**Theoretical Claims:**

There are no theoretical claims in the paper. There is a broad claim about how aggregating the incorrect choice probabilities across independent realizations does not lead to a less noisy estimate (unlike in Monte Carlo estimation); this may be good to justify theoretically.

---

### Decision · Program_Chairs · 2025-05-01

**Decision:**

Accept (poster)

**Comment:**

The paper focuses on Multiple Choice Question-Answering benchmarks, and explores why performance on those benchmarks do not predictably improve with scale. The reviewers agreed that the paper studies a well-motivated problem, and also that the paper overclaims its scope ("downstream tasks" as too broad compared to the MCQ tasks actually studied). All of the reviewers also appreciated the thorough empirical analysis. There were lingering questions around the "why" that the authors identified (that probability mass on incorrect options varies unpredictably as the models' scale increases); addressing those questions will substantially strengthen the paper.

(Regarding reviewer scores, one of the reviewers indicated that their "weak reject" score was actually an on-the-fence rating and that they were willing to accept the paper if the other reviewers reached that consensus. Taking that into account, there is broad consensus among the reviewers.)